# New plastids, old proteins: repeated endosymbiotic acquisitions in kareniacean dinoflagellates

Anna MG Novák Vanclová [ID] [1,2,3 ✉], Charlotte Nef [ID] [1,2], Zoltán Füssy [ID] [4], Adél Vancl[5], Fuhai Liu[1,6,7], Chris Bowler [ID] [1,2] & Richard G Dorrell [ID] [1,2,8 ✉]

## Abstract

**Dinoflagellates are a diverse group of ecologically significant micro-eukaryotes that can serve as a model system for plastid symbiogenesis due to their susceptibility to plastid loss and replacement via serial endosymbiosis. Kareniaceae harbor fucoxanthin-pigmented plastids instead of the ancestral peridinin-pigmented ones and support them with a diverse range of nucleus-encoded plastid-targeted proteins originating from the haptophyte endosymbiont, dinoflagellate host, and/or lateral gene transfers (LGT). Here, we present predicted plastid proteomes from seven distantly related kareniaceans in three genera (*Karenia*, *Karlodinium*, and *Takayama*) and analyze their evolutionary patterns using automated tree building and sorting. We project a relatively limited ( ~ 10%) haptophyte signal pointing towards a shared origin in the family Chrysochromulinaceae. Our data establish significant variations in the functional distributions of these signals, emphasizing the importance of micro-evolutionary processes in shaping the chimeric proteomes. Analysis of plastid genome sequences recontextualizes these results by a striking finding the extant kareniacean plastids are in fact not all of the same origin, as two of the studied species (*Karlodinium armiger*, *Takayama helix*) possess plastids from different haptophyte orders than the rest.**

**Keywords** Myzozoa; Protists; Automated Tree Sorting; Post-Endosymbiotic Organelle Evolution; Shopping Bag Model
**Subject Categories** Evolution & Ecology; Microbiology, Virology & Host Pathogen Interaction; Organelles

## Introduction

The origin of oxygenic photosynthesis, and its adoption by eukaryotes, has fundamentally changed the evolutionary landscape of life on Earth. Originating in cyanobacteria more than 3 billion years ago, photosynthesis has spread across the eukaryotes through multiple endosymbiotic acquisitions of plastids. These began with the primary plastids (i.e., chloroplasts) of cyanobacterial origin that represent the defining synapomorphy of Archaeplastida (glaucophytes, red and green algae, and plants) and are surrounded by two membranes. Plastids subsequently spread into other groups (e.g., diatoms, haptophytes, dinoflagellates, and euglenids) through multiple endosymbiotic acquisitions of eukaryotic algae, forming "complex plastids" that possess additional membranes, often continuous with the endoplasmic reticulum. Across the tree of life, plastids provide their hosts with novel metabolic and biosynthetic capabilities and serve as important vectors of horizontal gene transfer (Howe et al, 2008; Ku et al, 2015; Novák Vanclová et al, 2020; Dorrell et al, 2021).

Plastids have their own genomes that typically encode core photosynthetic proteins, alongside some of their genetic house-keeping machinery and primary metabolic enzymes. Their size and coding capacity vary, typically possessing 20–250 genes, driven by gradual and somewhat convergent gene loss (Uthanumallian et al, 2021). Plastids, however, require a much larger inventory of proteins to function (e.g., >2000 in *Arabidopsis* and model eukaryotic algal groups; Kleffmann et al, 2004; Gruber et al, 2015; Novák Vanclová et al, 2020); and most of these are encoded in the nucleus and post-translationally imported based on cleavable N-terminal targeting signals. These plastid-targeted proteins comprise the expression products of plastid-derived genes that were transferred to the host nucleus, host-derived genes newly adapted for plastidial function, and genes originating from lateral gene transfer (LGT) from other sources, including other symbionts that the host has interacted with over its evolutionary history (Moustafa et al, 2008). This makes plastid proteomes innately chimeric entities, or "shopping bags", whose composition may reflect both general principles of organelle evolution and the unique evolutionary histories of individual lineages (Larkum et al, 2007; Suzuki and Miyagishima, 2010; Dorrell et al, 2017; Novák Vanclová et al, 2020). These processes are poorly understood and very problematic to model in systems with very ancient plastids, such as plants, due to weak phylogenetic signal. However, a very pertinent model in this respect can be found in dinoflagellates.

Dinoflagellates are a diverse group of unicellular algae that are ecologically and economically significant; including crucial

[1]Institut de Biologie de l'École Normale Supérieure (IBENS), École Normale Supérieure, CNRS, INSERM, Université PSL, Paris, France. [2]CNRS Research Federation for the study of Global Ocean Systems Ecology and Evolution, FR2022/Tara Oceans GOSEE, Paris, France. [3]Institute Jacques Monod, Paris, France. [4]Faculty of Science, Charles University, BIOCEV, Vestec, Czechia. [5]Faculty of Mathematics and Physics, Charles University, Prague, Czechia. [6]Centre de Recherches Interdisciplinaires, Paris, France. [7]Tsinghua–UC Berkeley Shenzhen Institute, Shenzhen, China. [8]CNRS, IBPS, Laboratoire de Biologie Computationnelle et Quantitative - UMR 7238, Sorbonne Université, Paris, France. ✉E-mail: anna.novak-vanclova@ijm.fr; richard.dorrell@sorbonne-universite.fr

symbionts of corals, and toxin-producing species that can cause major losses in fisheries and aquaculture industries as well as endangering public health (Hackett et al, 2004; Wang, 2008). Within the eukaryotic tree of life, they fall among the supergroup Alveolata, specifically Myzozoa, along with chrompodellids and apicomplexans (Ševčíková et al, 2015). Dinoflagellates are of further evolutionary interest given their unique and idiosyncratic cell biology. Their nuclear genomes are generally extremely large (up to several hundred Gbp) with permanently condensed chromosomes that lack histones, while their mitochondrial genomes are very streamlined (Lin, 2011; Dorrell and Howe, 2015). The evolutionarily ancestral dinoflagellate plastids are descended from red algae and bound by three membranes (Moog and Maier, 2017). The ancestral dinoflagellate plastid is further marked by multiple functional oddities. These include a unique carotenoid pigment, peridinin (Haxo et al, 1976), and peridinin-chlorophyll a-proteins (PCP) that together form membrane-extrinsic light-harvesting antennae (Ogata et al, 1994), while the chlorophyll-binding proteins are massively paralogized and synthesized as polyproteins (Boldt et al, 2012). Another striking characteristic of these plastids is a bacterial-like nucleus-encoded form II RuBisCO (Morse et al, 1995) in contrast to the partially plastid-encoded form I RuBisCO (consisting of homo-octamers of large and small subunits) found in all other plastid lineages except chromerids (Janouškovec et al, 2010). Finally, dinoflagellate mitochondrial and plastid genomes have generally extremely reduced coding contents (three and twelve protein-coding genes, respectively), with the plastid genome further fragmented into small unigenic elements termed "minicircles" (Zhang et al, 1999; Nash et al, 2007). The transcripts from dinoflagellate plastid minicircles undergo distinctive maturation events, including extensive substitutional editing and the addition of a 3' poly(U) tail not found in most other plastid lineages (Zauner et al, 2004; Wang and Morse, 2006; Dorrell and Howe, 2015).

Dinoflagellates as a group are ecologically opportunistic: most plastid-bearing representatives are, in fact, mixotrophic (Cohen et al, 2021; Jeong et al, 2021), and in global meta-genomic analyses (e.g., *Tara* Oceans) they appear mostly as predators and parasites (Pierella Karlusich et al, 2022). Their plastids are frequently secondarily reduced to non-photosynthetic organelles, lost completely (Saldarriaga et al, 2001; Sanchez-Puerta et al, 2007; Gornik et al, 2015; Cooney et al, 2022), or replaced in new symbiogenetic events. These may result in fully integrated plastids (Tengs et al, 2000; Matsumoto et al, 2011), complex endosymbionts retaining plastids, mitochondria, and nuclei (Imanian and Keeling, 2007; Sarai et al, 2020), and kleptoplastids of various degrees of stability (Koike et al, 2005; Myung et al, 2006; Gast et al, 2007). In some cases, the coexistence of a new organelle or endosymbiont with a remnant of the ancestral plastid has been proposed (Hehenberger et al, 2019), and in one lineage, a complete repurposing of parts of the plastid and mitochondrion into an enigmatic, photosynthesis-unrelated optical structure termed an "ocelloid" has taken place (Gavelis et al, 2015).

Kareniaceae, typified by the genera *Karenia, Karlodinium*, and *Takayama*, are mixotrophic dinoflagellates whose plastids are related to those of haptophytes. Kareniaceae are the most abundant dinoflagellate lineage with non-peridinin plastids in *Tara* Oceans (Dataset EV1, based on de Vargas et al, 2015), and both *Karenia* and *Karlodinium* have important impacts as toxic components of harmful algal blooms. Instead of peridinin, plastids of the

Kareniaceae contain the accessory pigment fucoxanthin typical of their haptophyte ancestors (Zapata et al, 2004, 2012). Kareniacean plastids are further assumed to be bound by four membranes, the outermost of which is continuous with the endoplasmic reticulum as per haptophytes, distinct from the three membrane-bound plastids found in peridinin dinoflagellates. Further ecological and evolutionary complexity is found within the Kareniaceae in the Antarctic native Ross Sea dinoflagellate (RSD) (Gast et al, 2007; Hehenberger et al, 2019), which uses a kleptoplastid from the haptophyte *Phaeocystis antarctica* acquired independently to that of other Kareniaceae, *Shimiella gracilenta* with cryptophyte kleptoplastids (Ok et al, 2021), and *Gertia stigmatica* which contains peridinin instead of fucoxanthin (Takahashi et al, 2019). The well-characterized identity of both the ancestral and current kareniacean plastids, alongside their ecological abundance, diversity, and relevance, renders them a particularly appealing system for understanding the fundamental events associated with serial endosymbiosis and plastid replacement. Previous studies have suggested that similarly complex endosymbiotic interactions occurred more broadly across the eukaryotic tree of life and may have been key to the resulting symbiogenetic events (Hannaert et al, 2003; Huang and Gogarten, 2007; Dorrell and Smith, 2011; once-Toledo et al, 2019).

Partial kareniacean plastid genomes and transcriptomes have been sequenced for the species *Karlodinium micrum* (syn. *veneficum*) and *Karenia mikimotoi*, indicating the retention of around 100 genes on a circular or linear chromosome (Gabrielsen et al, 2011; Dorrell et al, 2016), which is somewhat fewer than the ca. 140 genes associated with haptophyte plastids, alongside the presence of episomal minicircles (Richardson et al, 2014). Further studies, based on expressed sequence tags (ESTs) (Ishida and Green, 2002; Nosenko et al, 2006; Patron et al, 2006) and, more recently, transcriptomes of *Karenia brevis* and *Karlodinium micrum* realized through the Marine Microbial Eukaryote Transcriptome Sequencing Project (MMETSP) (Burki et al, 2014; Keeling et al, 2014; Hehenberger et al, 2019), have provided foundational insights into the nucleus-encoded proteome of the kareniacean plastid. These are defined by a bipartite targeting sequence, consisting of an N-terminal signal peptide followed by a hydrophilic transit peptide, similar to those of haptophytes, although with apparent differences in composition (Patron and Waller, 2007; Yokoyama et al, 2011). Phylogenetic analysis of the genes encoding plastid-targeted kareniacean proteins (Nosenko et al, 2006; Patron et al, 2006; Burki et al, 2014; Bentlage et al, 2016; Matsuo and Inagaki, 2018) indicate that some resolve with haptophytes (i.e., the endosymbiont ancestor), while others show alternative origins. These include key elements of the plastid gene expression machinery (e.g., RNA editing and poly(U) tail addition), which are likely to have been derived from the dinoflagellate host and potentially associated with the ancestral peridinin plastid (Dorrell and Howe, 2012; Jackson et al, 2013); and cofactor biosynthetic pathways (e.g., isopentenyl pyrophosphate and proto-porphyrin/heme), which resolve principally either with dinoflagellates or sources suggesting horizontal acquisition (Matsuo and Inagaki, 2018). To our knowledge, a reconstruction of the fucoxanthin plastid proteome as a whole, either in silico or proteomics-based, has yet to be attempted, and its overall metabolic composition and specific evolutionary origin within the

**Table 1.  Assembly statistics and assessment of completeness for the five new transcriptomes.**

|  | Karenia mikimotoi | Karenia papilionacea | Karlodinium armiger | Karlodinium micrum RCC3446 | Takayama helix |
|---|---|---|---|---|---|
| Raw transcripts | 373,333 | 241,077 | 265,372 | 202,000 | 662,715 |
| N50 | 1221 bp | 1168 bp | 1231 bp | 1332 bp | 1326 bp |
| Avg contig length | 752 bp | 764.5 bp | 791 bp | 856 bp | 799 bp |
| Total size | 208.9 Mbp | 155.4 Mbp | 167.9 Mbp | 145.1 Mbp | 256.8 Mbp |
| Proteins in final dataset | 167,296 | 134,380 | 139,785 | 112,599 | 196,617 |
| Complete BUSCOs | 71.4% | 68.2% | 71.0% | 70.2% | 77.3% |
| Fragmented BUSCOs | 10.2% | 13.7% | 11.0% | 11.0% | 6.3% |
| Missing BUSCOs | 18.4% | 18.1% | 18.0% | 18.8% | 16.4% |

haptophytes remain unresolved (Yoon et al, 2002; Takahashi et al, 2019; Leblond et al, 2022).

Here, we explore kareniacean plastid evolution as a model for the endosymbiotic establishment of plastids and the spread of photosynthesis across eukaryotes. We leverage newly sequenced transcriptomic datasets for two additional strains of *Karenia* and *Karlodinium* and the first transcriptome for the genus *Takayama*, alongside high-throughput phylogenetic and environmental data to understand the diverse evolutionary trajectories of the kareniacean plastid following its acquisition. Our data point to a complex origin of the kareniacean plastid, whose probable nucleus-encoded plastid-targeted proteins appear to be of predominantly dino-flagellate origin, but whose composition is influenced by between-pathway, between-protein, and even between-species variations in phylogenetic origin. Considering the phylogenetic origins of both the nucleus-encoded and plastid-encoded proteins, we further reveal at least three independent acquisitions of the kareniacean plastid from different haptophyte lineages. This, however, does not reflect on the nucleus-encoded plastid protein inventory which exhibits a very similar and largely shared Chrysochromulinaceae-like phylogenetic signal suggesting monophyly of this part of the plastid proteome. Finally, we evaluate the different distributions of key lineages in the contemporary ocean. The repeated acquisition of the kareniacean plastid with different evolutionary consequences across different species can provide insights into the functional principles that constrain post-endosymbiotic organelle evolution, and the symbiogenetic establishment of photosynthesis across eukaryotes.

# Results

## Pan-transcriptomics reveals the core fucoxanthin plastid proteome

Five new transcriptomic datasets were produced for fucoxanthin dinoflagellates: *Karenia mikimotoi* and *Karenia papilionacea*, the RCC3446 strain of *Karlodinium micrum* and its distant relative *Karlodinium armiger*, and for *Takayama helix*, a predator of other dinoflagellates more closely related to *Karlodinium* than *Karenia* (Jeong et al, 2016). To our knowledge, this latter transcriptome represents the first one reported for the genus *Takayama*. These libraries, alongside four previously sequenced MMETSP libraries for *Karenia brevis*, one for *Karlodinium micrum* strain CCMP2283, and an independently sequenced transcriptome for the RSD (Data

refs: Keeling et al, 2014; Ryan et al, 2014; Hehenberger et al, 2019), represent the pan-kareniacean library used for all downstream analyses.

The raw dataset sizes range from around 200 to 660 Mbp but contain large amounts of redundant or partially redundant transcripts, reflecting the extensive paralogization and pseudogen-ization associated with dinoflagellate nuclear genomes (Stern et al, 2010). To reduce the size and redundancy of our datasets for subsequent analysis, we selected the most complete paralogs of each studied protein by cd-hit. The completeness of the datasets as assessed by BUSCO is around 70% which is rather high in comparison to other dinoflagellate datasets (Table 1; Dataset EV2).

## In silico prediction reveals within-lineage divergence in fucoxanthin plastid proteome contents

A modified version of ASAFind (Gruber et al, 2015), which has previously been used to reconstruct the haptophyte plastid proteome *in silico* (Dorrell et al, 2017), was constructed with a custom scoring matrix for fucoxanthin dinoflagellate plastids (see Materials and Methods) and used in combination with SignalP 5.0 in the final prediction pipeline. After preliminary automatic annotation of the retrieved *in silico* plastidial proteomes of all studied organisms by KAAS (https://www.genome.jp/kegg/kaas/; Moriya et al, 2007), we noticed that some of the integral plastidial proteins missing from these datasets could be captured by an alternative prediction approach combining PrediSI (Hiller et al, 2004) and ChloroP (Emanuelsson et al, 1999) employed previously for *Euglena gracilis* data (Ebenezer et al, 2019), and so we included these as well. The proportion of redundancy-treated translated transcripts predicted as plastid-targeted ranged from 7.5–14.5% in *Karlodinium micrum* and *Karenia brevis*, respectively (~9.5% being both the average and the modal value). Of these, most have a targeting sequence predictable by the modified ASAFind, while between 16–22% of the predicted plastid-targeted sequences represent additions based solely on the abovementioned alternative signal.

The putative phylogenetic origin of all predicted plastid proteins was investigated via a custom pipeline, integrating homology mining, single-gene tree construction and sorting based on topology and its statistical support (Appendix Fig. S1). Between 40 and 55% of each protein dataset had some homologs in our reference database containing representatives for most major eukaryotic groups, particularly enriched in haptophyte and dinoflagellate transcriptomes (see Materials and Methods). In about one-third of these, the identified

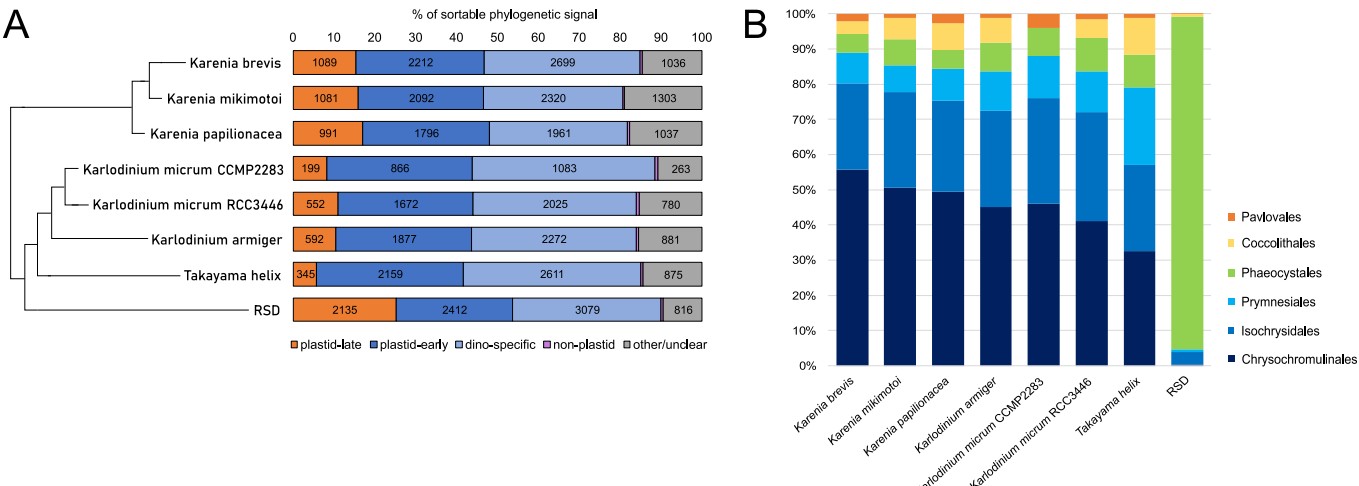

**Figure 1. Ratios of phylogenetic signal of plastid proteins with detectable homologs in Kareniacean transcriptomes.**

This plot shows:plastid-late (haptophyte-like; orange), plastid-early (dinoflagellate-like or dinoflagellate-specific; blue), ancestrally alveolate-like (purple), and other or unclear (gray) (**A**), and further breakdown of the plastid-late signal traceable to a concrete haptophyte family (Chrysochromulinaceae (dark blue), Isochrysidales (blue), Prymnesiales (light blue), Phaeocystales (green), Coccolithales (yellow), and Pavlovales (orange)) (**B**).

homologs were shared solely with other dinoflagellates and were therefore considered dinoflagellate-specific proteins without any further phylogenetic analysis. For the rest, single-gene trees (29,329 in total) were constructed and automatically sorted based on which evolutionary origin they support. The main two investigated categories were "plastid-early" (i.e., vertically inherited from the dinoflagellate host; color-coded as blue and purple dependent on whether the protein retrieved only dinoflagellate homologs or showed deeper homology to other alveolates or myzozoan lineages) and "plastid-late" (i.e., immediately clustering with the haptophytes; color-coded as orange in all graphical materials). In addition to this, potential cases of lateral gene transfer (LGT) from prokaryotes, green algae, and ochrophytes (as previously identified major sources of LGT within eukaryotic algal lineages, e.g., Moustafa et al, 2008; Matsuo and Inagaki, 2018; Novák Vanclová et al, 2020; Dorrell et al, 2021) were identified (examples shown in Appendix Figs. S6–S12).

Of all the proteins with homologs in the database, 65–80% were categorized as "plastid-early" in all studied organisms (Fig. 1A). The proportion of proteins categorized as "plastid-late" is generally in accord with the previously published estimates (Burki et al, 2014) but there was a noticeable difference between the genera, with ~15% noted in all three species of *Karenia*, 10% in *Karlodinium*, and 6% in *Takayama*. The proportion of proteins shared and clustering with ciliates, i.e., primarily heterotrophic alveolates (the "non-plastid signal"), is almost negligible in all studied organisms, as is the contribution of prokaryotic or "green" LGT (cases of which do not exceed 20 per organism). The amount of "brown" LGT is slightly higher than the amount of non-plastid signal (varying from 12–107 cases per organism). Between 10 and 20% of the signal in each organism was not resolvable into any of the investigated categories.

## Phylogenomic evidence for different evolutionary histories of kareniacean plastidial and non-plastidial proteins

We used the whole transcriptomes as input for the PhyloFisher pipeline (Tice et al, 2021) to construct a matrix of 241 pan-

eukaryotic nuclear-encoded genes for a reference phylogeny of the studied kareniacean nuclear genomes in the context of the eukaryotic tree of life. The topology reconstructed from the PhyloFisher matrix is consistent with recently published rRNA trees (Takahashi et al, 2019; Ok et al, 2021) with *Karenia* as a sister to both *Karlodinium* and *Takayama* (Fig. EV1, simplified in Fig. 1A).

Previous studies of the fast-evolving fucoxanthin plastid genome revealed a deep-branching position sister to all prymnesiophytes but no affinity towards individual orders (Choi et al, 2017; Klinger et al, 2018; Kawachi et al, 2021). To recover a more specific affinity, we enumerated the predicted plastid-targeted proteins clustering specifically with one of six haptophyte subgroups (Chrysochromulinaceae, Isochrysidales, Phaeocystales, Coccolithales, Pavlovales, and Prymnesiales; Fig. 1B), constituting approximately one-quarter of the total plastid-late signal. We also recorded how many trees showing these phylogenetic affiliations also recover Kareniaceae as monophyletic (1418 in total, 220 for Chrysochromulinaceae, 93 for Isochrysidales, 33 for Prymnesiales, and <20 for the rest; Appendix Fig. S2). For each of the seven studied fucoxanthin-containing kareniaceans the most abundant category (up to >50%) was Chrysochromulinaceae. In RSD, which possesses a kleptoplastid derived from *Phaeocystis antarctica*, the main source was Phaeocystales (>70%) and only a very small proportion (~2.5%) was shared with the other kareniaceans. We noted a strong secondary signal to the order Isochrysidales in all kareniacean species, including the RSD, even normalizing for dataset size (Appendix Fig. S3), although many were specific to individual genera (Appendix Figs. S4, S5).

We also sorted the single-gene trees based on the inner kareniacean topology they support (Fig. 2) and note that while the organismal topology (Fig. EV1) is predominantly recovered in trees defined as plastid-early (1505 out of 1922, 78%), the plastid-late trees often show different topologies, especially the one in which *Karenia* and *Karlodinium* are sisters to the exclusion of *Takayama* which forms a sister or

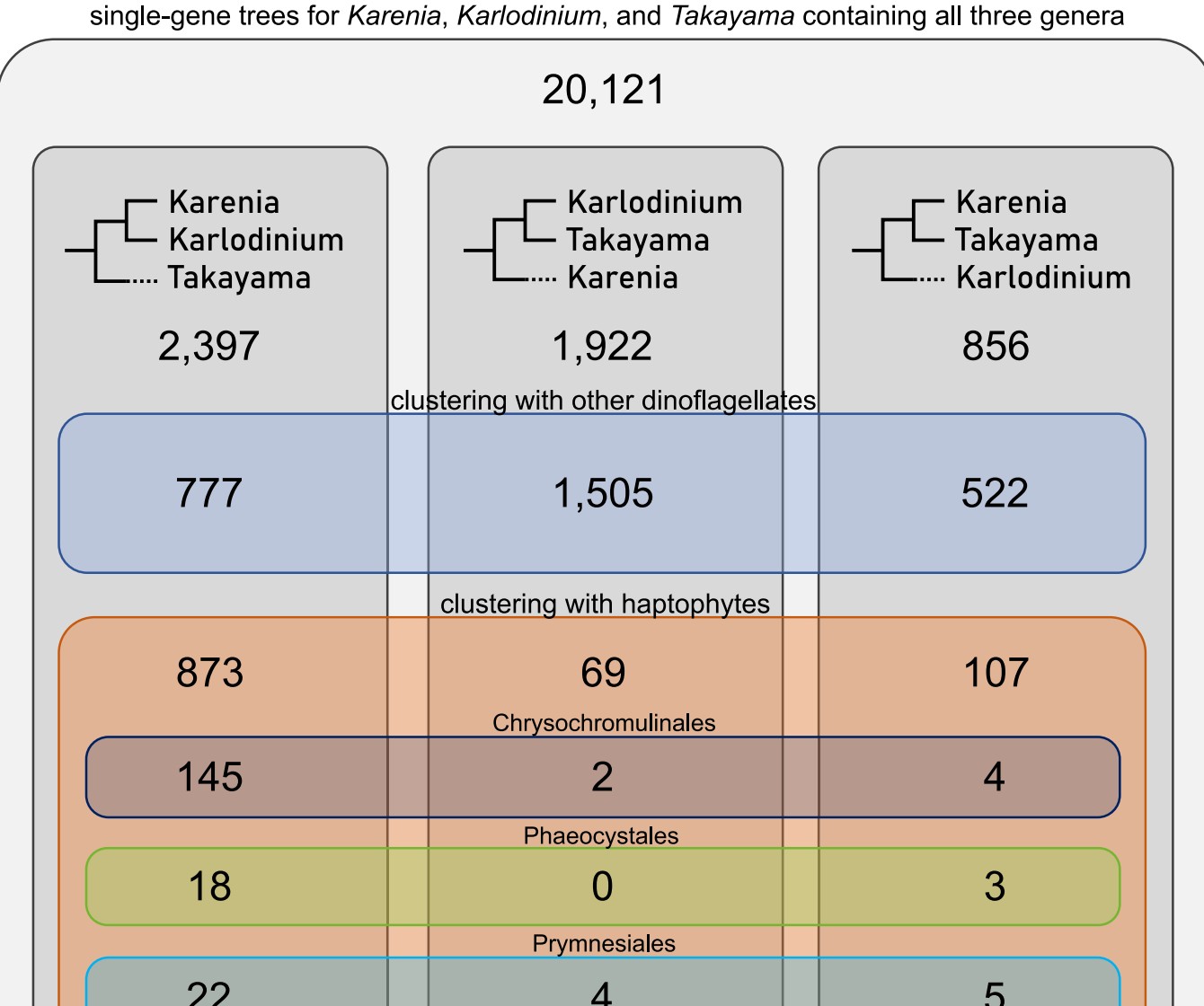

**Figure 2. Proportions of single-gene trees supporting three different internal topologies of the three studied Kareniacean genera, grouped by broader phylogenetic origin.**

The plastid-early genes clustering with other dinoflagellates primarily support the topology obtained from 18S trees and PhyloFisher trees (i.e., with *Karenia* sister to *Takayama/Karlodinium*). In contrast, the plastid-late genes clustering with haptophytes predominantly support *Takayama* as an outgroup of *Karenia* and *Karlodinium*. *Karlodinium* is less frequently recovered as an outgroup to a monophyletic clade of *Takayama/Karenia*.

separate branch (873 trees compared to 176 that support the two alternative topologies). We further investigated these relationships using a selected set of 23 nucleus-encoded plastid-late genes to build manually curated single-gene and partial-concatenation trees (Fig. EV2; Dataset EV10). Consistent with the above results, they largely supported the monophyly of *Karenia* and *Karlodinium* and exclusion of *Takayama*, and a close relationship to Chrysochromulinaceae.

## A hybrid plastid proteome with species-specific innovations

The plastid proteins of each kareniacean were automatically annotated with enzymatic functions based mainly on KEGG, presented as a pivot table (Dataset EV3). Considering $\chi^2$ enrichments of individual KEGG annotations (KO IDs) across all seven fucoxanthin-containing species, we noted a statistically

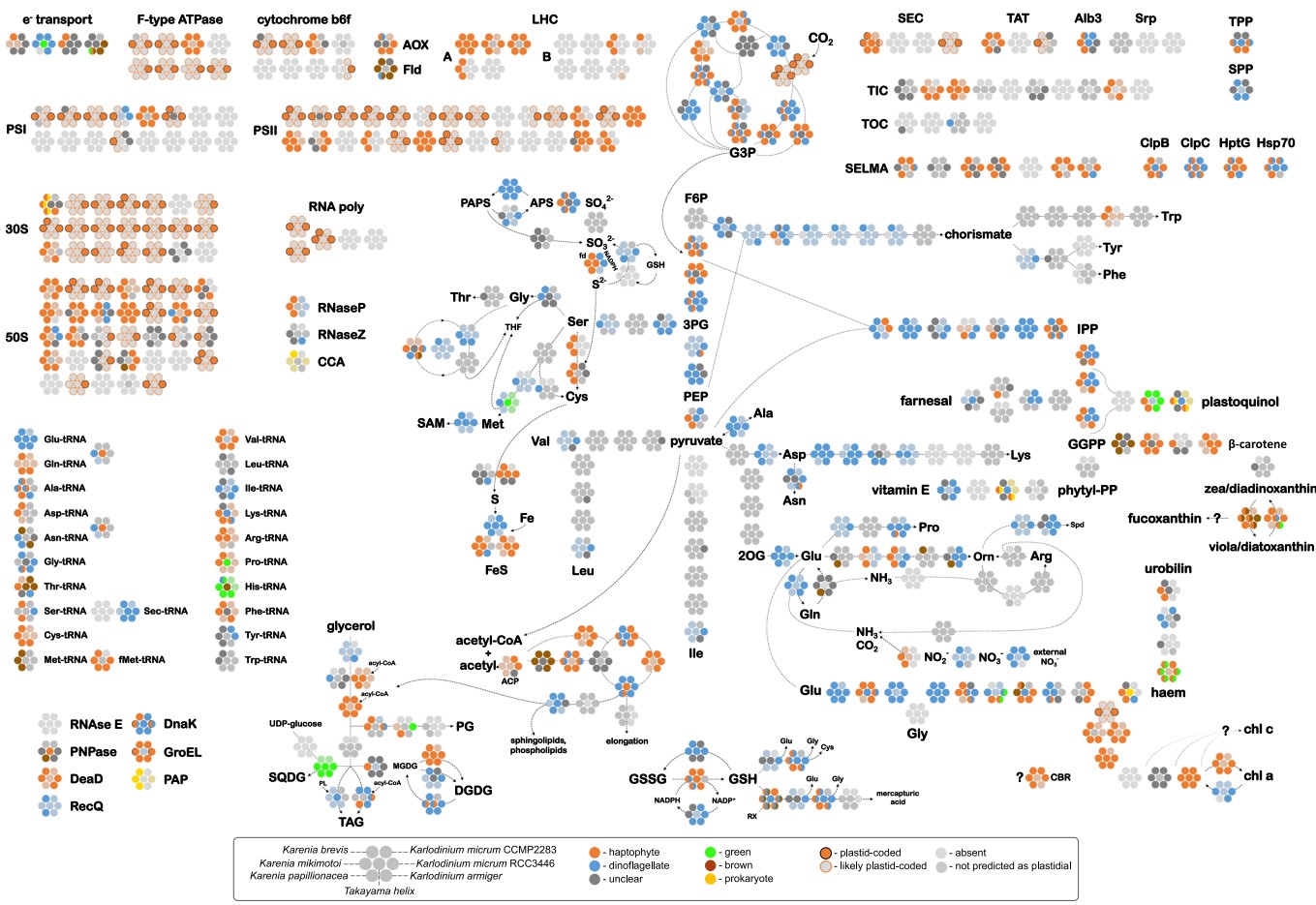

**Figure 3. Reconstruction of major metabolic pathways of plastids of the seven kareniaceans, adapted from KEGG (Kanehisa and Goto, 2000).**

Plastid proteins are arranged by major metabolic pathway or biological process, with each protein shown as rosettes. Each rosette (described in the legend) relates to the homologs of a specific nucleus-encoded and plastid-targeted protein, with each circle within each rosette corresponding to a different species, and different colors corresponding to different evolutionary affiliations. Proteins of plastid-late (haptophyte) origin, such as are concentrated in photosystem and ribosomal processes, are colored red; and proteins of plastid-early (dinoflagellate) origin, such as are concentrated in carbon and amino acid metabolism are colored blue. Additional phylogenetic origins are denoted by further examples in the legend, e.g., LGT shown in orange, green, and brown. Proteins encoded in plastid genomes (red with black border) are provided for the incomplete genome assembly for *Karlodinium micrum*, and partial plastid transcriptomes are identified for *Karenia mikimotoi* and *Karenia brevis*. In certain cases (shown as rosettes with multiple colors), homologs from different species have different evolutionary origins, e.g., *Karenia* possessing plastid-late and *Karlodinium/Takayama* plastid-early. A detailed breakdown of the individual sequences with their predicted origins used for this reconstruction are available in Dataset EV3, and an alternative graphic, including the RSD, is available as Appendix Fig. S13.

significant enrichment ($p < 0.01$) in the plastid-late signal in the BRITE category "Photosynthesis", the combined category "Translation", and the KEGG Pathway "Porphyrin and chlorophyll metabolism", while the plastid-early signal was enriched in the BRITE categories "Membrane trafficking", "Protein kinases, phosphatases, and associated proteins", and "Ion channels". Reflecting the probable plastid-early origin of the fucoxanthin plastid RNA processing machinery, categories relating to transcription and mRNA do not appear to be biased in their origin.

When analysed manually and qualitatively, additional evolutionary patterns were apparent in the overall metabolic map (Fig. 3), some of which were previously described in *Karenia brevis* and *Karlodinium micrum* (Matsuo and Inagaki, 2018). These include notable differences between the genera and species in some pathways or enzymes, and several general particularities such as unexpectedly missing, duplicated, or laterally gained proteins.

### Tetrapyrrole metabolism

The tetrapyrrole pathway is clearly divided into three parts: a backbone leading from glutamine to protoporphyrin, the heme and bilin pathway, and the chlorophyll-synthesizing branch (Tanaka and Tanaka, 2006). While the former two are generally of mixed origin (including isolated cases of putative LGT), perhaps slightly skewed towards a plastid-early signal, the latter comprises almost exclusively plastid-late proteins. The only plastid-early enzyme associated specifically with chlorophyll is chlorophyllase, which has additional roles in phytol recycling to chlorophyll synthesis. Two evolutionarily distinct versions of the biliverdin-producing heme oxygenase seem to be present in the plastid of representatives of both *Karenia* and *Karlodinium*: one of plastid-late and one of green-like origin (each in one or two copies, Fig. EV3). This enzyme is often present in multiple copies that have been proposed to play slightly different roles in other reactions of bilin

metabolism, especially in photosynthetic organisms where these molecules serve as chromophores of phytochromes (Dammeyer and Frankenberg-Dinkel, 2008).

## Terpenoid metabolism

While the origin of the central MEP-DOXP pathway is mixed (slightly skewed towards plastid-early) and often not consistent between the genera or even species, this is not the case for other pathways branching from it. Carotenoid metabolism comprises mostly plastid-late proteins and does not contain any plastid-early proteins. Two enzymes (15-cis-phytoene synthase (Appendix Fig. S10) and zeaxanthin epoxidase) represent LGT from brown algae. MPBQ/MSBQ methyltransferase (VTE3) and homogentisate solanesyltransferase (HST; Appendix Fig. S11) likely represent LGTs from bacteria and green algae, respectively. However, neither the plastoquinol nor tocopherol synthesis pathway seems to be complete.

## Fatty acid and plastid lipid biosynthesis

Retention of a plastid-derived type II FAS pathway has already been implied in fucoxanthin and in peridinin dinoflagellates (Janousko-vec et al, 2017) and our data supports this with all steps of this pathway identified in at least some of the species. Most of these enzymes are of plastid-late origin or have both plastid-early and plastid-late copies. The origins are not always consistent between the genera (compare FabG and FabI in Fig. 3 or Dataset EV3). While one of the two enzymes for sulfoquinovosyldiacylglycerol (SQDG) synthesis, SQD2 is of green algal-like origin in all genera, representing a reliable case of conserved LGT (Appendix Fig. S9), the other, SQD1, was not identified in any of the seven transcriptomes. That said, SQD1 (and sometimes SQD2 as well) were not detectable in various algae in previous studies (Novák Vanclová et al, 2020, Riccio et al, 2020), possibly due to very low expression or presence of as yet undiscovered alternative enzyme.

## Photosynthesis

The majority of photosystem and light-harvesting complex subunits and parts of the photosynthetic electron transport chain are of plastid-late origin. We detected no homologs of peridinin-chlorophyll-binding proteins (PCP) in any kareniacean transcriptome. PsaD is the only photosystem subunit that was vertically inherited from the ancestral plastid in Karlodinium and likely Takayama (where the protein was not predicted as plastid-targeted, likely due to N-terminal truncation), but it remains plastid-coded in Karenia (Dorrell et al, 2016). Another part of the photosynthetic apparatus that is not of plastid-late origin is plastidial ferredoxin (PetF) which shows a plastid-early origin in Karenia and Karlodinium, and is replaced by a green-like homolog in Takayama. PetF is also associated with non-photosynthetic metabolism (e.g., in leucoplasts of non-photosynthesizing plant tissues), and is plastid-encoded in the non-photosynthetic chryso-phyte Spumella NIES-1486 (Dorrell et al, 2019), and its specific metabolic functions in Kareniaceae remain to be determined.

## ATP synthase

Most subunits of the plastidial F-type ATP synthase are plastid-encoded in Kareniaceae and their sequences were not recovered in the transcriptomes. The gamma subunit (atpG, K02115) is encoded in the nucleus and is of plastid-late origin in all species.

Surprisingly, the delta subunit (atpH, K02113) was not detected in any of the transcriptomes. AtpH is a conserved and essential subunit that is usually plastid-encoded (including in haptophytes) but was not retrieved as such in Karlodinium micrum and Karenia mikimotoi (Gabrielsen et al, 2011; Dorrell et al, 2016). Interestingly, an unrelated and structurally dissimilar but functionally analogous subunit of mitochondrial ATPase (ATP5D, K02134) seems to be duplicated, but it is unclear whether these two phenomena are linked.

## Calvin cycle

The Calvin cycle in kareniaceans is a true evolutionary mixture, with most enzymes having multiple copies of both early and late origins. Sedoheptulose-bisphosphatase is the only protein that seems to be of purely plastid-late origin in all species, which may reflect that it is redox-regulated and has exclusive functions in photosynthesis (Gütle et al, 2016). In contrast, triosephosphate isomerase and transketolase are exclusively of plastid-early origin, while further enzymes that function reversibly in both glycolysis and pentose phosphate pathway (Kroth et al, 2008) possess both plastid-late and plastid-early copies. Notably, phosphoribulokinase is an exception from this overall trend as it is a photosynthetic carbon fixation-specific enzyme that possesses uniquely plastid-early origins.

## Amino acid metabolism

While enzymes for the synthesis of most amino acids are present in the transcriptomes, only a small portion of them were predicted as plastid-targeted: the synthesis of alanine, aspartate, and asparagine from pyruvate and several following enzymes in the pathway towards lysine (up to 4-hydroxy-tetrahydrodipicolinate reductase), some of the enzymes converting glutamine to ornithine, the final step of proline synthesis (pyrroline-5-carboxylate reductase), some enzymes for serine and glycine synthesis from 3-phosphoglycerate, and the GS/GOGAT pathway. The shikimate pathway does not seem to be plastid-localized. The only exceptions to this are the AROM polypeptide of Karenia brevis; and 3-dehydroquinate synthase, which is plastid-targeted in five of the species and shows a plastid-early origin in Karlodinium and Takayama and a plastid-late origins in Karenia. Interestingly, the same evolutionary division can be observed for two of the enzymes of ornithine synthesis from glutamate, acetylglutamate kinase and N-acetyl-gamma-glutamyl-phosphate reductase.

## Sulfur and nitrogen metabolism

Sulfur metabolism, including enzymes converting serine to cysteine, is evolutionarily mixed but comparatively richer in plastid-late signals. The iron-sulfur cluster assembly (SUF) is not uniform in its origin and comprises plastid-early SufS and SufB but plastid-late SufC, D, and E. Most of the nitrogen assimilation enzymes were not predicted as plastid-targeted, but the ones present are also evolutionarily non-uniform: the putative plastidial nitrate/nitrite transporter is pan-alveolate (shared with ciliates); nitrate reductase is plastid-early; while nitrite reductase shows a plastid-late origin.

## Protein import

The kareniacean protein import machinery shows a predominant plastid-late origin. This is expected as the SELMA (Symbiont-specific ERAD-Like MAchinery for protein import) is present in haptophytes

and kareniaceans but not peridinin dinoflagellates. Nonetheless, there are isolated cases of plastid-early SELMA components targeted to the plastid, although these may equally relate to host ERAD components with false positive predictions. Alb3 represents an interesting case of a protein strongly implicated in plastid biogenesis and photosynthesis with both plastid-early and plastid-late homologs in *Karenia brevis* and only a plastid-early homolog in *Karlodinium micrum* and *T. helix*.

### Ribosomal proteins

Plastid-targeted ribosomal proteins show mostly plastid-late origins with only a few exceptions. These include L7/L12 in *Karlodinium* and *Takayama*, L17 of *Karlodinium*, and L15 in *Karenia mikimotoi* and *Karlodinium armiger*, which are of plastid-early origin, and might represent either genes retained from the ancestral plastid, or retargeted, originally mitochondrial proteins. All three species of *Karenia* contain an additional copy of ribosomal protein S1 that likely represents bacterial origin and is shared with only a few eukaryotic algae. Single gene trees of these proteins indicate the presence of further, non-plastid-targeted homologs in all *Karenia* transcriptomes, of as yet unclear function.

### Aminoacyl-tRNA synthesis

The plastidial inventory of aminoacyl-tRNA synthetases is evolutionarily mixed, with relatively high concentrations of LGT (Figs. EV4 and EV5; Appendix Figs. S6–S8). There are two cases of LGT from green algae: one in *Takayama* only (PARS, proline-tRNA synthetase) and a second (HARS, histidine-tRNA synthetase) in all species except *Takayama* where, in turn, the protein seems to be brown in origin. Three other synthetases (TARS, threonyl-, NARS, asparaginyl-, and MARS, methionyl-) are of brown origin in at least some representatives. We did not detect clear evidence for plastid/mitochondrion dual-targeting in the identified plastid-targeted tRNA synthetases.

### Targeting sequences of kareniacean plastid proteins vary with their evolutionary origin

Next, we wished to test if there are differences in the plastid-targeting pre-sequences of plastid-late and plastid-early proteins, by comparison to analogous protein regions from peridinin dinoflagellate and haptophyte references. The signal peptides of the predicted kareniacean plastid-targeted proteins were found to typically contain a central LACLAC motif and a terminal GHG motif directly preceding the cleavage site regardless of whether they were of plastid-late or plastid-early origin (Fig. 4A,B), whereas these motifs do not occur in haptophytes nor in peridinin dinoflagellates (Fig. 4C,D). Comparisons of the unique three-letter motifs associated with the plastid-targeting sequences of each group revealed particularly strong enrichments in the LACLAC and GHG motif (and its variants) in kareniacean proteins of plastid-late origin, compared to proteins of plastid-early origin and haptophyte and dinoflagellate equivalents (Fig. 4E), despite similar overall amino acid composition of signal peptides across all datasets (Appendix Fig. S14). In contrast, limited similarities were found in the transit peptides of the predicted kareniacean plastid-targeted proteins, apart from the double-arginine motif immediately following the cleavage site and conserved proline at +15 and arginine at +22 after the cleavage

sites (Fig. 4F). No notable differences in the investigated pre-sequences were observed between proteins whose functions are consistent with either thylakoid or stromal localizations (Appendix Fig. S15).

### Plastid phylogenetics reveals different origins of kareniacean plastid genomes and proteomes

To further investigate the aforementioned incongruence between plastidial and non-plastidial phylogenetic signals, we sequenced 6 plastid-encoded genes (*psbA*, *psbC*, *psbD*, *psaA*, *rbcL*, 16S) previously used as markers for the origins of the fucoxanthin plastid genome (Takishita et al, 1999; Dorrell and Howe, 2012). We chose to use m/rRNA sequences, sequences amplified by RT-PCR from *Karenia papillionacea*, *Karlodinium armiger*, and *T. helix* mRNA, and retrieved by reciprocal tBLASTn/ BLASTx searches of *Karenia brevis* MMETSP libraries. Considering a concatenated alignment of the five protein-coding genes, alongside equivalent sequences from 19 haptophyte plastid genomes and transcriptomes (Dorrell et al, 2017; Strassert et al, 2021), we obtained a tree topology in which *Karlodinium micrum* and all three species of *Karenia* form a monophylum sister to all haptophytes except Pavlovales and rappemonads, while *Takayama* and *Karlodinium armiger* fall outside this clade (Fig. 5). *Takayama* resolves inside Phaeocystales with 95% bootstrap support. The position of *Karlodinium armiger* is likewise distinct from *Karlodinium micrum* and *Karenia* (90% bootstrap support) and inside the Prymnesiales (86% bootstrap support). Much more densely-sampled 16S rDNA sequence trees broadly support these placements, albeit with weaker bootstrap support, and ambiguity over the monophyly versus polyphyly of the *Karenia*/ *Karlodinium veneficum* grouping (Dataset EV10).

### Global distribution of fucoxanthin dinoflagellates reveals negative co-occurrence between Kareniaceae and relatives of their plastid donors

To provide biogeographical and ecological context for the interpretation of our transcriptomic and phylogenetic results, we investigated the distribution of *Karenia*, *Karlodinium*, and *Takayama* genera in the *Tara* Oceans database using the V9 18S metabarcoding data (Dataset EV4). Our results highlight different patterns: *Karenia* seems to contribute the most to dinoflagellate populations, although this may be biased by the different ribosomal DNA copy numbers in different genera (Galluzzi et al, 2010). Concerning distributions, *Karlodinium* clearly differs from the other two genera in its higher overall station occupancy and more cosmopolitan presence, which is noticeable especially in the Indian and Arctic Oceans (Fig. 6). Partial least square (PLS) analysis showed a negative correlation of phosphate, nitrate, and iron to the abundances of both *Karlodinium* and *Takayama*, whereas *Karenia* showed positive correlations to these variables, alongside temperature and oxygen (Appendix Fig. S16).

A second series of PLS analysis of the abundances of the three kareniacean genera to one another revealed a positive correlation of *Takayama* with both *Karenia* and *Karlodinium*, although no internal correlations between the latter two genera (Appendix Fig. S17). Strikingly, all three genera had strong negative correlations to the relative abundances of six haptophyte subgroups (Chrysochromulinaceae, Phaeocystales, Isochrysidales,

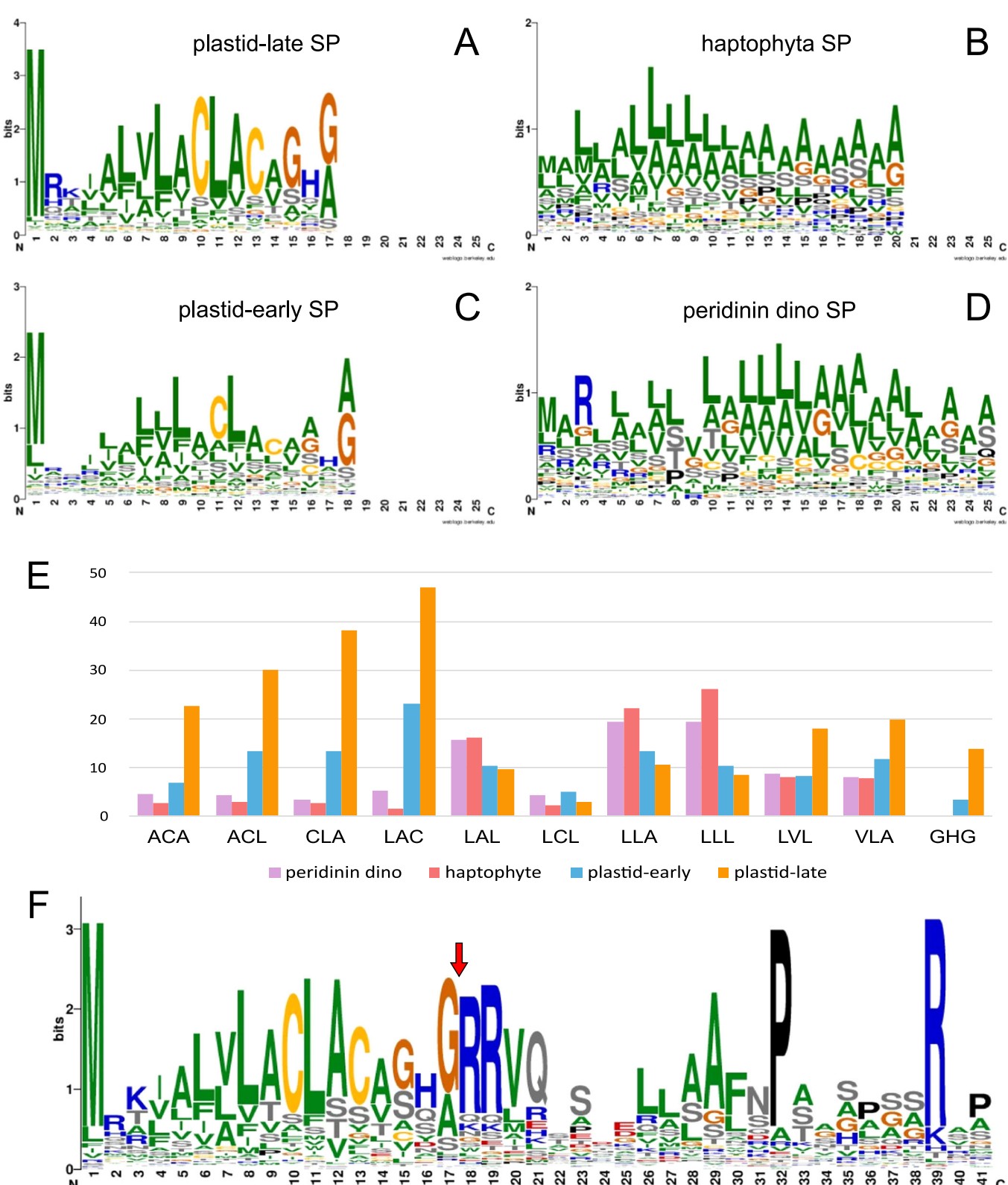

**Figure 4. Sequence logos for aligned signal peptides associated with kareniacean plastid proteomes.**

These plots show comparable signal peptides for four datasets of different evolutionary identity: haptophyte, peridinin dinoflagellate, plastid-late and plastid-early kareniacean signal peptides (**A–D**), occurrences of three-letter sequence motifs in signal peptides of each dataset, represented as a percentage of signal peptides in which at least one such motif occurs (**E**), and the sequence logo of the signal peptide and partial transit peptide for kareniacean plastidial proteins regardless of their phylogenetic origin with a red arrow denoting the signal peptide cleavage site (**F**).

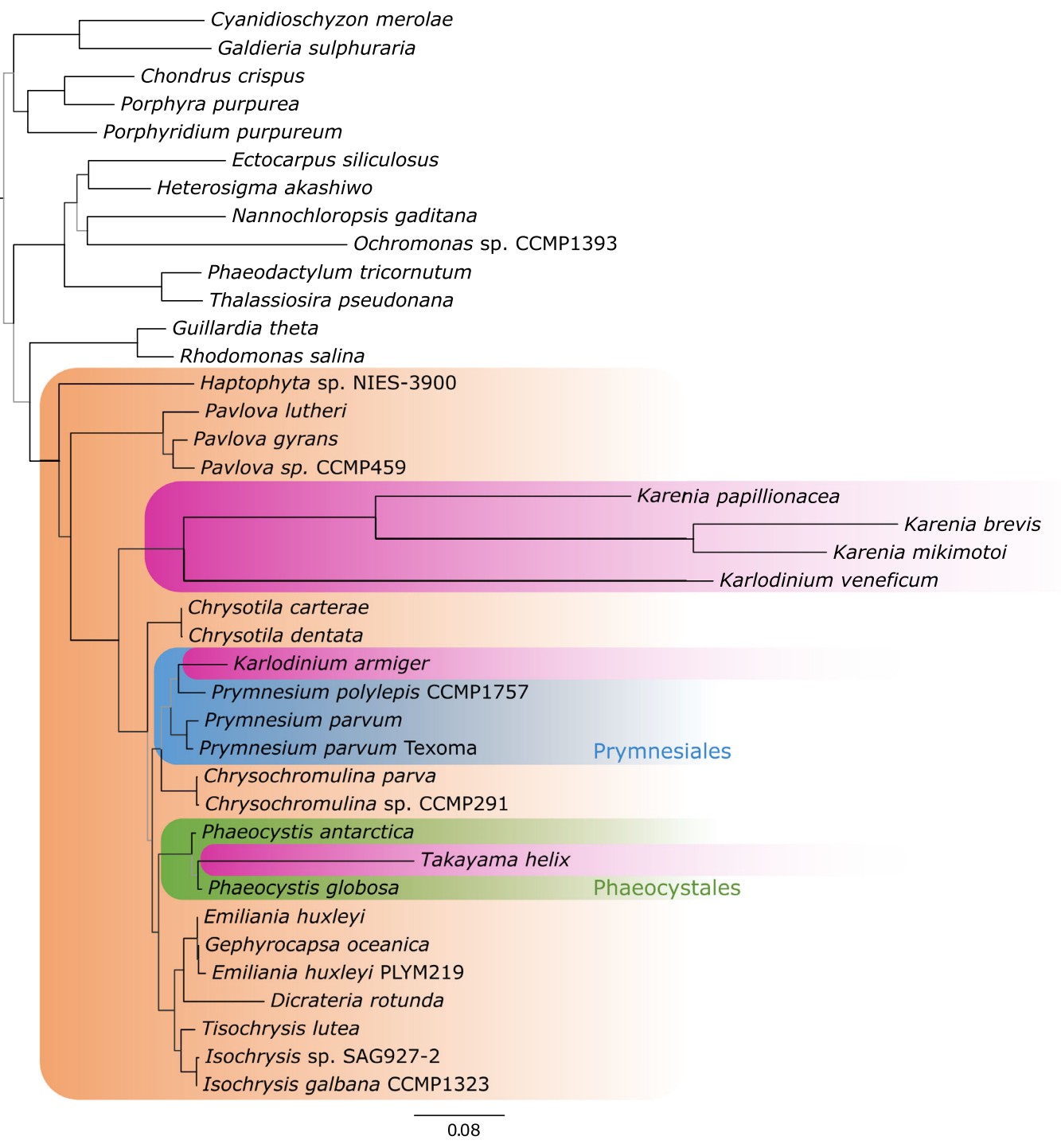

**Figure 5. Polyphyletic origins of kareniacean plastid genomes.**

Phylogenetic tree reconstructed by IQ-TREE based on five plastid-encoded proteins (PsaA, PsbA, PsbC, PsbD, and RbcL) of six kareniaceans and 20 haptophytes for which partial or complete plastid genomic datasets are available, alongside 13 further algae with primary and complex red plastids; bootstrap support is expressed by the branch color (black for ≥90%, dark gray for ≥75%, and light gray for <75%). Phylogeny based on plastid genome sequences does not support the monophyly of kareniacean plastids and their close relationship to Chrysochromulinaceae retrieved based on nucleus-encoded plastid proteins. Instead, *Takayama helix* and *Karlodinium armiger* split from the rest, placing within Phaeocystales and Prymnesiales, respectively, with high support.

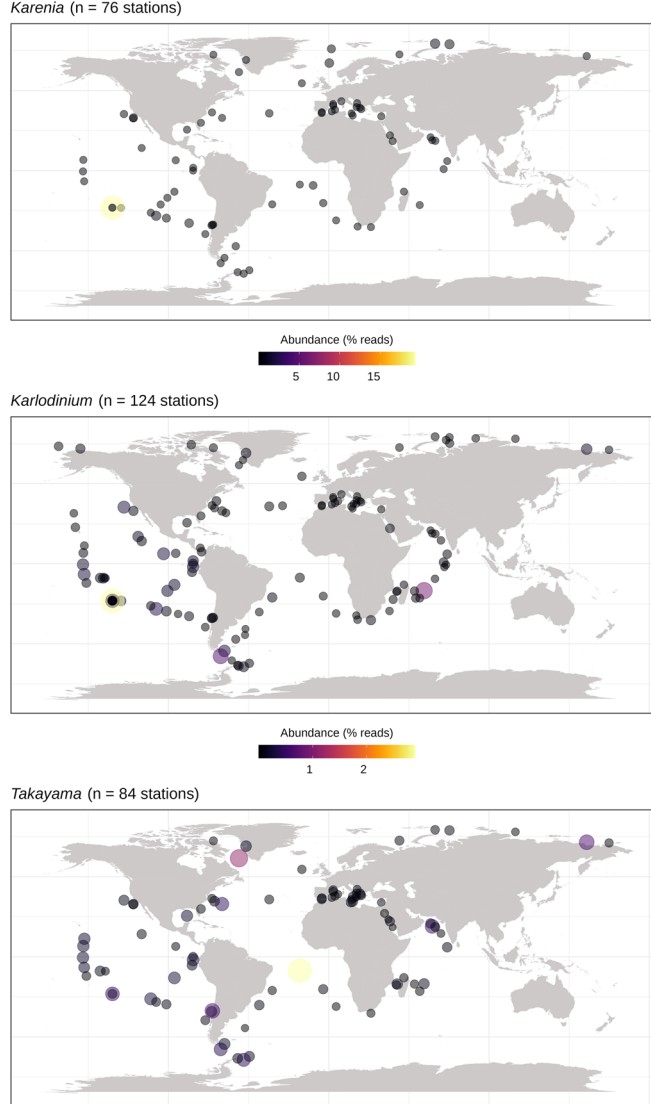

*Karenia* (n = 76 stations)

Abundance (% reads)
5   10   15

*Karlodinium* (n = 124 stations)

Abundance (% reads)
1   2

*Takayama* (n = 84 stations)

Abundance (% reads)
0.02   0.04   0.06

**Figure 6.   Relative abundances of the three studied kareniacean genera in *Tara* Oceans stations based on ribosomal 18S SSU V9 metabarcoding— *Karenia* exhibits higher abundance compared to both *Karlodinium* and *Takayama*, the latter displaying the lowest.**

*Karlodinium* exhibits more cosmopolitan distributions in comparison to the others. A Correlation heatmap of each species against one another, showing a significant positive correlation between the relative abundances of *Karlodinium* and *Takayama* is provided in Appendix Fig. S17, while correlation circles against individual environmental variables are provided in Appendix Fig. S16.

Coccolithales, and Pavlovales), with the strongest negative correlations observed between *Karlodinium*, Chrysochromulinaceae, and Phaeocystales (Fig. 7).

## Discussion

In this study, we harness novel transcriptomes, customized signal prediction, automated phylogenetic pipelines, and metabarcoding

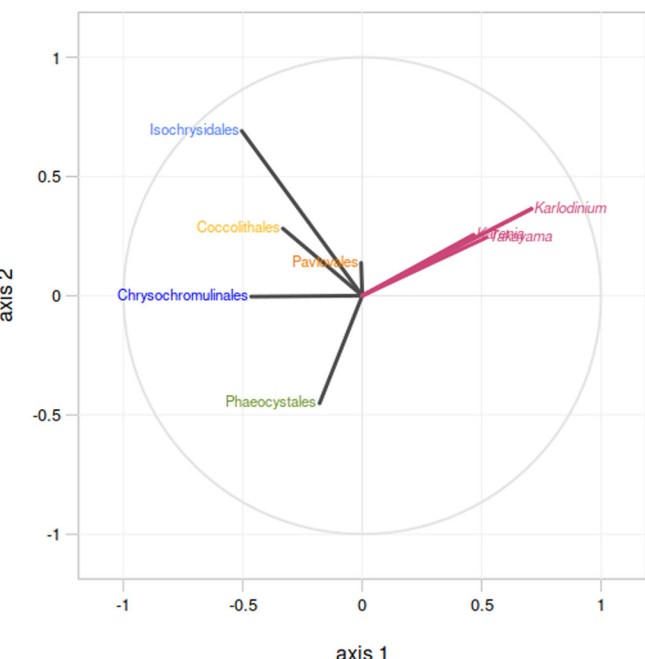

**Circle of Correlations**

**Figure 7.   Non-coincidence of kareniaceae and haptophytes in global metagenome datasets.**

Correlation circle based on the partial least square analysis of the *Tara* Oceans relative abundances (surface depth only) of the three studied kareniacean genera against those of different haptophyte families (color-coding is the same as in Fig. 1B) as observable variables; all three kareniaceans are negatively correlated to all families and most strongly to Chrysochromulinaceae.

analyses from *Tara* Oceans to explore the evolution of the fucoxanthin-containing plastids of the kareniacean dinoflagellates, a system of special evolutionary importance for understanding endosymbiosis. Our data emphasize the relaxed nature of plastid acquisition and loss across Kareniaceae and their importance as a model system for studying the fundamental mechanistics and micro-evolutionary variation in plastid evolution.

The idea that individual genes encoding plastid-targeted proteins may exhibit evolutionary affiliation with other groups than the plastid donor, typifying the "shopping bag" model (Larkum et al, 2007), is well-established in many plastid lineages. Our data show that the dinoflagellate host was the principal contributor of nucleus-encoded proteins supporting the kareniacean plastid proteome (Fig. 1A). We do not detect higher numbers of proteins of Phaeocystales origin in *Takayama* or of Prymnesiales origin in *Karlodinium armiger*, in contrast to the origins of their current plastid genomes (see below). Both of these results typify the fact that all of these plastids represent replacements of previous organelles of the same type rather than complete innovations, and were therefore able to build upon a set of pre-existing genes and adaptations. This explains the observed "momentum" of the nucleus-encoded plastid proteome in comparison to the organelle itself.

Nonetheless, based on a large-scale sorting based on tree topology and manually curated phylogenies, we propose a specific haptophyte subgroup, the Chrysochromulinaceae, as a major contributor to the plastid-late part of kareniacean plastid

  

proteomes. Combined with the frequent monophyly of genes of this origin (Figs. 2 and EV2), this strongly suggests that a common ancestor of the studied organisms either possessed a stable plastid or had a long-term symbiotic relationship (e.g., kleptoplastidic) with a haptophyte lineage related to the extant Chrysochromulinaceae.

The evolutionary patterns in the compiled metabolic pathways provide further insights into the specific roles of endosymbiotic gene transfer (EGT), LGT and host-derived proteins in the early evolution of the fucoxanthin plastid (Fig. 3). The preferential incorporation of plastid-late pathways into photosynthesis-associated and ribosomal pathways may reflect that these functions were not encoded in the kareniacean host and that the new plastid (or much less frequently LGT) was the only available source. It is also possible that there has been a direct niche competition between the peridinin and fucoxanthin plastid that may have coexisted in the same host for a period of time with possibly different selective pressure on the retention of individual imported proteins. For example, there may be a selective advantage to plastid-late photosystem and ribosomal proteins interacting with plastid-encoded proteins of the same origin.

In contrast, the retention of dinoflagellate-derived proteins in other pathways (e.g., isoprenoid, heme, and amino acid biosynthesis) implies that the kareniacean ancestor retained these functions. We note that these functions are also often associated with protist lineages that have secondarily lost photosynthesis but retain vestigial plastids (Hadariová et al, 2018; Záhonová et al, 2018; Dorrell et al, 2019). It has previously been suggested that the RSD retains a non-photosynthetic form of the peridinin plastid (Hehenberger et al, 2019), and a similar organelle might have been the donor of the plastid-early pathways. One interesting case illustrating this scenario is the plastid-early chlorophyllase, the only enzyme of this origin in the chlorophyll branch of tetrapyrrole metabolism, which might have been retained by a heterotrophic kareniacean ancestor for the purpose of phytol recycling from chlorophyll contained in its algal prey. Many of the plastid-early gene trees copy the organismal topology and may represent proteins vertically inherited from a cryptic peridinin plastid. Nonetheless, the presence (or eventual fate) of the peridinin plastid in fucoxanthin-containing lineages awaits structural confirmation via microscopy.

The apparent retention of ancestral RNA processing pathways (e.g., poly(U) tail addition and substitutional editing) in the fucoxanthin plastid is puzzling, given the predominantly plastid-late origins of plastid translation. If the Kareniacean ancestor possessed a non-photosynthetic plastid, it would be expected to lose its plastid genome and associated expression machinery as the peridinin plastid only encodes core photosystem subunits (Dorrell and Howe, 2012; Jackson et al, 2013; Cooney et al, 2022). That said, RNA editing is widespread in dinoflagellate mitochondrial and nuclear genomes (Lin et al, 2002; Liew et al, 2017), and is also present in the green plastids of *Lepidodinium* that also secondarily replaced the peridinin-containing ones (Matsuo et al, 2022). It remains to be determined if the fucoxanthin plastid RNA processing machinery is of peridinin plastid or non-plastidial host origin.

These general trends aside, the plastid metabolic map is generally divided between parts strongly enriched in plastid-late proteins and parts assembled from both sources, likely in an evolutionarily neutral, patchwork-like way (Fig. 3). This pattern does not only manifest at the level of one pathway being formed by enzymes of different evolutionary affiliations but often also at the level of one step in the pathway being carried out by homologs of different origin in different organisms. We propose that fucoxanthin plastid establishment may have been a relatively gradual process during which peridinin-like and haptophyte-like genes coexisted and competed for place, even after the divergence of individual species. This is also reflected in the partial plastid genomes assembled from *Karenia* and *Karlodinium*, which contain partially non-overlapping sets of genes that suggest independent post-endosymbiotic plastid genome reduction (Gabrielsen et al, 2011; Richardson et al, 2014; Dorrell et al, 2016). Our data establishes the importance of species-level variation as playing a substantial role in shaping the kareniacean plastid proteome.

Strikingly, our three phylogenomic analyses based on (a) plastid-unrelated conserved nuclear genes, (b) nucleus-encoded and haptophyte-like plastid genes, and (c) plastid genome-encoded genes, yield contradicting tree topologies (Figs. 5 and EV1,EV2). Kareniacean plastid genomes are split into three lineages, each associated with different haptophytes: Prymnesiales for *Karlodinium armiger*, Phaeocystales for *Takayama*, and a further unresolved but clearly distinct source for the genus *Karenia* and *Karlodinium micrum*. Remarkably, this phylogenetic diversity is in accord with differences in pigment composition between these lineages (Garcés et al, 2006; Zapata et al, 2012).

Furthermore, the phylogenetic signal from kareniacean plastid-targeted proteins, despite suggesting a common origin and evolutionary affinity (Fig. 1; Appendix Fig. S2), does not agree with the organismal topology and suggests a closer association of *Karlodinium* and *Karenia* in comparison to *Takayama* (Fig. 2). This result is further complicated by more recently described kareniacean species that do not possess a fucoxanthin plastid, i.e., RSD (Hehenberger et al, 2019), *Gertia* (Takahashi et al, 2019), and *Shimiella* (Ok et al, 2021) and their phylogenetic positions as closer relatives to *Karlodinium* and *Takayama* to the consistent exclusion of *Karenia*. While *Gertia* retains reduced but photosynthetically active peridinin plastid, RSD has been proposed to possess its relict form (Hehenberger et al, 2019).

In Fig. 8, we present three main hypothetical scenarios attempting to reconcile the seemingly contradicting evolutionary histories of kareniacean nuclei, plastids, and their nuclear-encoded protein inventories (top panels), further discussed below.

Given the shared haptophyte signal, often showing clear monophyly and phylogenetic association with Chrysochromulinaceae, in all organisms, we propose that they indeed possessed the same type of fucoxanthin plastid, already a replacement of the ancestral peridinin one, at some point of their history, which was later lost and replaced in *Karlodinium armiger* and *Takayama helix* but conserved in other *Karlodinium* and *Karenia*. Previous studies have noted the extensive capacity of both *Takayama helix* and *Karlodinium armiger* for mixotrophic metabolism (Berge et al, 2008; Berge and Hansen, 2016; Jeong et al, 2021), supporting the capacity for endosymbiotic replacement. *Karlodinium armiger* also produces significantly shorter branches than the others in plastid genome trees, which suggests a recent acquisition.

The scenario we find the most parsimonious and best reflecting the results of our phylogenetic analyses is the one where the fucoxanthin plastid was acquired by the common ancestor of

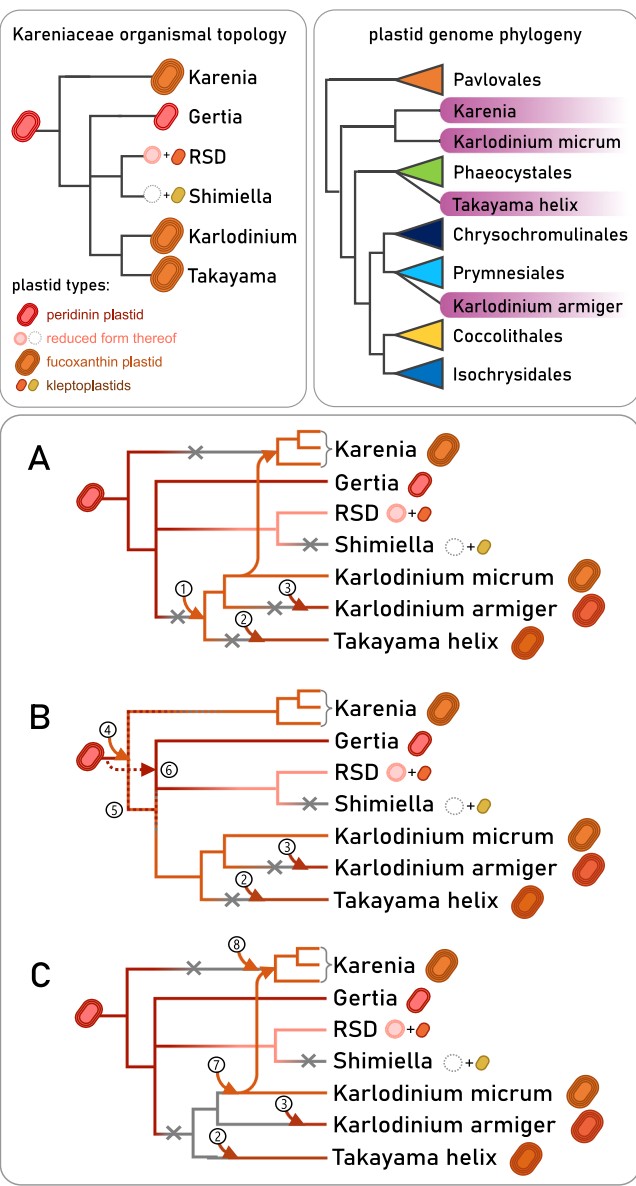

**Figure 8. Proposed scenarios of plastid evolutionary history in Kareniaceae integrating previously published organismal topology with our data based on plastid protein and plastid genome phylogeny.**

Scenario (**A**) proposes that the first fucoxanthin plastid was acquired by the common ancestor of *Karlodinium* and *Takayma* (1) and later transferred to *Karenia* and independently lost and replaced in *Takayama* and *Karlodinium armiger* (2, 3). This scenario best explains the shared Chrysochromulinaceae-like signal and tree topologies recovered by our phylogenetic analyses (Figs. 2 and EV2) but requires a higher number of plastid losses. Scenario (**B**) places the fucoxanthin plastid origin at the base of all Kareniaceae (4) and requires the peridinin plastid to either have coexisted with the fucoxanthin one for a period of time (5) or have been lost and re-acquired (6) to explain its presence in *Gertia* and RSD. It proposes less plastid losses and no additional endosymbiotic transfer between fucoxanthin species, but it explains the shared Chrysochromulinaceae-like signal only partially as it is congruent with its monophyly but not the inner relationships. Scenario (**C**) proposes an independent plastid origin for all four lineages (2,3,7,8) or for three lineages with subsequent transfer between *Karlodinium* and *Karenia* (2,3,7) or vice versa (only one direction is shown in the schematic). This scenario considers the Chrysochromulinaceae-like signal a phylogenetic artifact caused by the under-sampling of the plastid donors or sources of non-specific LGT.

*Karlodinium* and *Takayama* (Fig. 8A) and subsequently underwent an additional endosymbiotic transfer from *Karlodinium* to *Karenia*, explaining their sister position frequently recovered in our trees (Fig. 2). We note that the predator-prey relationships that may have facilitated such transfer are not uncommon even between toxin-producing dinoflagellates (Jeong et al, 2016). Conversely, the smaller size (de Salas et al, 2008) and cosmopolitan distribution of *Karlodinium* might make it a potential prey for *Karenia*, although this awaits direct observation. While it is true that we observe a small number of plastid-late genes shared between the RSD and other kareniaceae, which may be signals from a relict fucoxanthin plastid (cf., peridinin-derived signals in dinoflagellates that have completely lost a plastid; Gornik et al, 2015), this number is very small and may constitute phylogenetic noise.

In an alternative scenario, the fucoxanthin plastid was already present in the common ancestor of all three genera (Fig. 8B), i.e., the ancestor shared with non-fucoxanthin kareniaceans, all of which would have to have lost this plastid. This scenario would also have to involve either a period of coexistence of peridinin and fucoxanthin plastid or a loss and re-acquisition of the peridinin plastid to explain its presence in *Gertia*.

The third possibility considers independent acquisitions in all four lineages (Fig. 8C), with the shared phylogenetic signal representing a phylogenetic artifact stemming from the under-sampling of closely related plastid donors and sources of non-endosymbiotic LGT, or three independent acquisitions and a transfer from *Karlodinium* to *Karenia* or vice versa.

From the diversity of signal sequences targeting proteins into fucoxanthin plastids, we consider possible structural similarities that may hint at their evolutionary origin. The most striking feature of kareniacean signal peptides (both plastid-late and plastid-early proteins) is their high degree of conservation (Fig. 4A–D). At the same time, they are very dissimilar to those of peridinin dinoflagellates and haptophytes. If they were indeed inherited from one or the other, they have clearly evolved profoundly under very strong selection to the point of being more strictly conserved than their ancestors and are untraceable. The other possibility is that they arose *de novo* during fucoxanthin plastid establishment, and their highly similar sequence reflects their common and possibly relatively recent origin and spread, e.g., via gene duplications, domain swapping, or the activity of transposable elements (Burki et al, 2012). The recent assembly of a draft *Karenia brevis* nuclear genome (Nelson et al, 2021) may allow us to assess the genomic context of genes encoding plastid-targeted proteins.

Finally, the *Tara* Oceans distributions of each lineage raise questions about the broader ecological processes underpinning their plastid evolution. Despite their different distributions, all three kareniacean genera show a strong negative correlation with haptophytes. The strong negative association with Chrysochromu-linaceae and the lack of negative correlation with Pavlovales are notable in regard to their relative genetic imprints on the kareniacean plastids (highest for the former, lowest for the latter; Cf., Figs. 1B, 7). These anti-correlations might reflect competitive exclusion or active predation avoidance (Lima-Mendez et al, 2015; Vincent & Bowler, 2020) or simply different niche preferences between each group. We note that the basic metabarcoding analysis of the biogeographical distribution performed in this study is preliminary, particularly as it relies on the analysis of entire haptophyte orders. It will therefore be worthwhile in the future to

assess the distributions of other more recently developed marker genes (Penot et al, 2022; Pierella Karlusich et al, 2022). The specific ecological interactions between the progenitors of Kareniaceae, and the haptophyte ancestors of their fucoxanthin plastid, may be best inferred via ancestral niche reconstruction for each lineage.

# Methods

## Reference database preparation

An in-house protein database was used for homology searches throughout this project. This database consisted of 155 transcriptomic and genomic datasets fetched from NCBI (https://www.ncbi.nlm.nih.gov), 1KP (https://db.cngb.org/onekp/; Leebens-Mack et al, 2019), and decontaminated MMETSP (Keeling et al, 2014) as described and in Dorrell et al, 2017: 30 for eubacteria, 10 for archaea, and 115 for eukaryotes with sampling spanning most of the diversity with focus on plastid-bearing groups (100 datasets) and particularly enriched in dinoflagellate and haptophyte sequences (37 and 34 datasets, respectively). The source database can be accessed at https://osf.io/ykxes/, and an annotated list of datasets used in our study is available in Dataset EV2.

## Data acquisition and de novo transcriptome sequencing

The previously published peptide sequences for *Karenia brevis*, *Karlodinium micrum* (syn. *veneficum*), and Ross Sea Dinoflagellate (RSD) were downloaded from their respective public repositories (Data refs: Keeling et al, 2014; Ryan et al, 2014, Hehenberger et al, 2019) and decontaminated as described above. In case of *Karenia brevis*, the four available assemblies from four clones were pooled together and treated as a single dataset with redundant sequences removed randomly.

*Karenia mikimotoi* RCC1513, *Karenia papilionacea* RCC6516 and *Karlodinium* sp. RCC3446 were obtained from Roscoff Culture Collection (https://roscoff-culture-collection.org), *Takayama helix* CCMP3082 was obtained from the Bigelow National Center for Marine Algae and Microbiota (https://ncma.bigelow.org), and *Karlodinium armiger* K0668 was obtained from the Norwegian Culture Collection of Algae (https://niva-cca.no). The cultures were grown in natural seawater (Horsey Mere seal colony, North Sea) supplemented with k/2 or enriched artificial seawater on a 12 h light (50μE): 12 h dark cycle at 19 °C for 4 weeks, to late-exponential phase (1–2 million cells/ml). Cells were concentrated by centrifugation at 4000 rpm for 10 min, washed three times in sterile marine PBS (mPBS), and snap-frozen in liquid nitrogen. Total cellular RNA was extracted as previously described (Dorrell and Howe, 2012) using Trizol (Invivogen) and chloroform phase extraction, isopropanol precipitation, and resuspended in nuclease-free water (Qiagen). RNA integrity was confirmed by electrophoresis on ethidium bromide-stained agarose gels and quantified using a NanoDrop photospectrometer (Thermo Fisher). Two μg RNA was subsequently treated with 2U RNA-free DNAse (Promega) following the manufacturers' instructions, precipitated with isopropanol, resuspended in nuclease-free water, and quality-checked and quantified again.

For each species, 1 μg DNase-treated RNA was polyA-tail selected, and strand-specific libraries were prepared with TruSeq Stranded mRNA Library Prep kit (Illumina). The resulting libraries were sequenced using a Novaseq flow cell (Fasteris) with 100 bp paired reads, followed by in-house base-calling and basic Q30 enumeration (https://www.fasteris.com).

The obtained paired-end reads were further quality controlled using the RNA-QC-chain toolkit (Zhou et al, 2018) by filtering based on phred score with the threshold of 30; and GC content with the allowed range set to 42–60% based on the expected values for kareniacean dinoflagellates (Lidie et al, 2005), and by removing reads potentially representing rRNA contamination (default settings were used for this step). Filtered reads were then assembled by Trinity (Grabherr et al, 2011) with --NO_SEQTK and --trimmomatic options, and the resulting assembly was selected for the longest isoforms by the respective script from the Trinity toolkit. The quality and completeness of the transcriptomic datasets were assessed by TrinityStats and BUSCO (eukaryota_odb10, n:255) (Simão et al, 2015). The datasets were translated to proteins by TransDecoder (with default settings; https://github.com/TransDecoder/TransDecoder) and subsequently decontaminated by homology search using LAST (Kiełbasa et al, 2011) against the in-house database and removing sequences homologous to eubacterial, archaeal, or metazoan sequences with e-value of 1E-10 or lower. The translated datasets were used for further analyses alongside the previously published transcriptomes in their translated forms and are available at https://figshare.com/articles/dataset/full_protein_datasets_tar/21647771. The sequence data were deposited at NCBI under BioProject ID PRJNA788777.

## Plastid signal prediction

First, the relative effectiveness of two SignalP versions (3.0 and 5.0) were determined for discriminating plastid-targeted proteins using a model dataset (Appendix Fig. S20; Dataset EV6). SignalP 5.0 (Almagro Armenteros et al, 2019) was determined to possess greater specificity for the identification of plastid-targeted proteins (22/728 non-plastid control proteins predicted to possess signal peptides, compared to 34 for SignalP 3.0) and used for all subsequent analysis.

Both original sequences and sequences cut to the first methionine were analysed to accommodate for potential translated UTRs or spliced leaders, and the results were pooled afterward prioritizing the positive signal peptide prediction in case of different results. The obtained SignalP results were converted into a tabular format compatible with SignalP versions 3.0–4.1 for downstream applications with a custom script (Computer code EV1). For the second step predicting the rest of the bipartite targeting signal, a modified version of ASAFind (Gruber et al, 2015; Füssy et al, 2019) was used with a custom scoring matrix (Computer code EV1; Dataset EV5) that reflects the differences in sequence surrounding the signal peptide cleavage site of kareniaceans (for instance, instead of the phenylalanine motif, multiple arginine residues are typically present). The matrix was prepared based on sequence logos prepared by Seq2Logo (Thomsen and Nielsen, 2012; Appendix Fig. S18) from model datasets of *Karenia brevis* and *Karlodinium micrum* proteins determined as having exclusively plastidial, mitochondrial, nuclear, or endomembrane functions, e.g., photosynthetic core machinery proteins for plastidial functions, and identified from reciprocal BLAST against well-curated proteomic datasets from other organisms (Butterfield

et al, 2013; Klinger et al, 2013; Dorrell et al, 2017; Bannerman et al, 2018; Beauchemin and Morse, 2018), the queries and retrieved model sequences are available in Dataset EV5. The modified script and custom scoring matrix were subsequently tested and optimized for sensitivity and specificity, and in the final version represented an improvement of 56.4 and 0.5 percentage points, respectively, over unmodified ASAFind. The detection thresholds used with SignalP were also tested, with no apparent improvement in specificity/ sensitivity tradeoffs over the default settings; as well as prediction by HECTAR and a combination of PrediSI and ChloroP (Gschloessl et al, 2008; Dorrell et al, 2019; Ebenezer et al, 2019; Appendix Figs. S19, S20; Dataset EV6).

Additionally, we employed an alternative prediction combining PrediSI (Hiller et al, 2004) and ChloroP (Emanuelsson et al, 1999) with the intention to capture potential proteins with peridinin plastid-like signal domains similar to those of *Euglena* on which it was previously tested (Patron and Waller, 2007; Ebenezer et al, 2019). Proteins judged as not coding for a plastid-targeting sequences by the modified ASAFind but identified as plastid-targeted by this approach were included in the predicted plastid proteomes as well, albeit with prefix "SPTP-" denoting the presence of an alternative signal.

The presence of large numbers of highly similar sequences caused by very recent paralog duplication and/or alternative splicing is common in dinoflagellates. To reduce this complexity for phylogenetic analyses, all datasets were additionally clustered by cd-hit with an identity threshold of 0.9. This was performed after signal prediction to avoid accidentally discarding an isoform/paralog with a predictable targeting signal in favor of one without it.

## Phylogenetic analysis

Each of the predicted plastid proteomes was processed independently in order not to omit any variant signals in individual proteins. First, homologs were mined by LAST (Kiełbasa et al, 2011) search with threshold e-value 1E-10 against the in-house database plus all the fucoxanthin dinoflagellate transcriptomes. The best hit from all organisms in the in-house dataset were retained alongside the best five hits from each other fucoxanthin dinoflagellate library to account for potential paralogs in fucoxanthin dinoflagellate transcriptomes. Proteins with no hits whatsoever or with no homologs in any taxa other than dinoflagellates (Kareniaceae or other) were annotated as "lineage-specific" or "dinoflagellate-specific" respectively, and their phylogeny was not investigated further. The remaining homolog sets were aligned by MAFFT (version 7.310, Mar 17 2017; Katoh and Standley, 2013) with default settings and trimmed by TrimAl (version 1.4, Dec 17 2013; Capella-Gutierrez et al, 2009) with -gappyout parameter. The trimmed alignments were then filtered by a custom python script that discarded sequences comprising of more than 75% gaps, and then rejected alignments shorter than 100 positions or containing fewer than 10 taxa. The overall statistics of these alignments (distributions of their total lengths, numbers of sequences, and percentages of gaps) are provided in Appendix Figs. S21–S23. The satisfactory alignments were then used for maximum-likelihood tree construction by IQ-TREE (multicore version 2.0.5, May 15 2020; Minh et al, 2020) with automatic model selection and 1000 ultrafast bootstraps. For a graphical summary of this pipeline, see Appendix Fig. S1. All trees are available at https://figshare.com/articles/dataset/all-automatically-built-trees_tar/21647768; in-house scripts used in the pipeline are available among supplementary files (Computer code EV1); as well as the final predicted plastid protein datasets with their phylogenetic annotations (Dataset EV9).

TreeSorter, a freeware program written in Python (https://github.com/vanclcode/treesorter/), builds abstract tree structures from a tree file in Nexus format. It then takes arguments defining sets of taxa, and criteria based on those sets, that determine what constitutes a valid subtree. Each tree is traversed and the highest bootstrap value of an edge (bipartition) that produces a subtree containing the taxon of interest and matching the defined criteria is reported. Criteria allow for further quantification of taxon sets, such that the minimum and/or maximum number or proportion of taxa from a particular set per subtree. TreeSorter was used for large-scale analysis of the tree topologies and sorting and annotating the seed plastid proteins by their probable evolutionary origin into custom categories: haptophyte (plastid-late), dinoflagellate (plastid-early), alveolate (non-plastid). Further signals considered included potential LGTs/EGTs from green (i.e., green algal, plant, or chlorarachniophyte), brown (i.e., ochrophyte), or prokaryotic sources as sequences of these evolutionary origins were noticed in these organisms in previous studies (Nosenko et al, 2006; Waller et al, 2006); all remaining trees for which one of the above annotations could not be identified (i.e., trees with no clear phylogenetic signal) were classed as other/unresolved. In order to be assigned to one of the categories, the seed protein had to be either sister to or nested within a clade comprising exclusively members of said category and of at least 2 taxa. The bootstrap value of the highest-supported branch dividing such clade from the rest of the tree was then saved as the numerical score of this sorting result. In case the protein can be assigned to more than one category, the one with a higher bootstrap was selected, and if they had equal scores, the protein was considered unresolved. In the case of nested categories (such as dinoflagellates and alveolates), the wider category is selected. Bootstrap scores of 75 were generally considered as the lower threshold in interpreting the evolutionary origins, however under specific circumstances (e.g., when the same protein was assigned the same origin with score ≥75 in other studied species), lower scoring results are reported as well.

The predicted plastid proteomes were automatically annotated by KAAS and major metabolic pathways were reconstructed using KEGG Mapper (https://www.genome.jp/kegg/mapper/reconstruct.html). Additional functions and missing enzymes were manually identified by targeted HMMER, based on their conserved domains as detected by PfamScan (Finn et al, 2014).

## Phylogenomic analysis

First, a phylogenetic matrix for non-plastid proteins (72,162 positions) was prepared using the standard workflow of the PhyloFisher toolkit (Tice et al, 2021), with the manual step of paralog annotation by parasorter, and removal of sequences containing >66% of gaps; the matrix constructor statistics, input metadata, and the matrix itself are available as supplementary files (Dataset EV10).

Manually curated phylogenetic matrices for plastid-targeted proteins were prepared from genes that passed our criteria of (a)

tree-based evidence of plastid-late origins, (b) occurrence in at least 5 of the 7 organisms, and (c) manually verified functional annotations and plastid-associated functions: 1-acyl-sn-glycerol-3-phosphate acyltransferase, 3-hydroxyacyl-ACP dehydratase, 3-oxoacyl-ACP synthase, ACP, cysteinyl-tRNA synthetase, digalactosyldiacylglycerol synthase, glutaminyl-tRNA synthetase, heme oxygenase, chlorophyll a synthase, chlorophyllide b reductase, LHCA1, lycopene beta cyclase, magnesium chelatase subunit H, magnesium-protoporphyrin *O*-methyltransferase, protochlorophyllide reductase, PsbO, PsbP, PsbU, RP-L1, RP-L13, SecA, SufC, and SufD. The alignments, single-gene and concatenated trees are available in supplementary files (Dataset EV10).

Next, a phylogenetic matrix for plastid-encoded proteins (2404 positions) was prepared using five genes (*PsaA*, *PsbA*, *PsbC*, *PsbD*, and *RbcL*) obtained from previously published plastid genomes and sequences encoded in MMETSP transcriptomes and *de novo* sequencing of PCR products (Dorrell et al, 2017), and inspected with IQ-TREE as above. Preliminary single-gene trees for the plastid-targeted proteins were manually checked to remove paralogs, very long branches, and likely cases of LGT, if present; these trees with the removed taxa highlighted, as well as all original and trimmed datasets, and both final matrices are available as supplementary files (Dataset EV10). Finally, trees based on a single gene for 16S rRNA with up to 644 haptophyte sequences, including environmental data (Choi et al, 2017), were constructed using various alignment, trimming, and models (details in the supplementary files).

Concatenated trees were then constructed by IQ-TREE with the LG + C60 + F model for the plastid matrices and posterior mean site frequency (PMSF) model (LG + C60 + F + G with a guide tree constructed with C20) for the PhyloFisher matrix (Wang et al, 2018).

## Signal peptide analysis

Four experimental datasets for this analysis consisted of two sets of plastid-targeted plastid-early (264) and plastid-late (437) proteins with assigned KEGG annotation with uniquely plastidial function and two analogous sets selected from MMETSP transcriptomes for five peridinin dinoflagellates (358) and five haptophytes (272) (details in Dataset EV7) based on presence of (a) detectable signal peptide and (b) the same KEGG annotations to produce comparable datasets without functional bias. Signal peptide regions of these proteins were determined by SignalP 5.0 (the region of ~25 positions immediately following the predicted cleavage site was considered as putative partial transit peptide region). The regions were aligned by ClustalW, manually curated to keep their conserved part and used to create sequence logos for each dataset using WebLogo (https://weblogo.berkeley.edu/logo.cgi; Crooks et al, 2004). The overall amino acid composition and number of occurrences of three-letter sequence motifs suggested by the logos in each dataset was also determined.

## Metabarcoding distributions analysis

To explore the distribution of the three Kareniaceae genera, the V9 18S rDNA metabarcoding data from *Tara* Oceans corresponding to the genera *Karenia*, *Karlodinium* and *Takayama* were extracted from (Ibarbalz et al, 2019) (available at https://zenodo.org/record/3768510#.Ye-xqpHMJzp). Overall, a total of 69 barcodes were

extracted for *Karenia*, 76 for *Karlodinium*, and 131 for *Takayama*. Their respective distributions (at least equal to three reads) in the eukaryotic-enriched size fractions (Pierella Karlusich et al, 2019) (0.8–5 μm; 5–20 μm; 20–180 μm, and 180–2000 μm) and two depths (surface and deep-chlorophyll maximum (DCM)) were pooled together and normalized against the total reads assigned to dinoflagellates, extracted as above, within each sample to account for their respective contribution to dinoflagellate populations. Results were plotted using the R package "mapdata" v2.3.0 (http://cran.nexr.com/web/packages/mapdata/index.html). The dinoflagellate occupancy was estimated as the percentage of stations in which their respective barcodes were identified compared with the total stations investigated.

A correlation analysis was conducted on the pooled Kareniaceae size fractions and depths, including a total of 76 stations in which at least one of the organisms was retrieved, with the R package "corrplot" (Wei and Simko, 2021). Partial least square analyses were implemented with the R package "plsdepot" (https://github.com/gastonstat/plsdepot) to explore the correlations between each kareniacean genus and a diverse set of environmental variables (range-transformed median values of nutrient concentrations of iron, nitrate, silicate, phosphate; and oxygen, salinity, temperature, all extracted from the PANGAEA database; Ardyna et al, 2017; Guidi et al, 2017a, 2017b). A second set of partial least square analyses were performed to explore the correlations of each kareniacean genus to each other, and to the total relate abundance of V9 metabarcode data for five haptophyte orders (Chrysochromulinaceae, Coccolithales, Isochrysidales, Pavlovophyceae, and Phaeocystales) as described in Penot et al, 2022. Abundances were all Hellinger-transformed. Abundances and environmental data considered for the partial least square analyses were computed only for the surface layer with all four size fraction abundances pooled together.

## Plastid transcript amplification and sequencing

Nucleotide sequences corresponding to plastidial transcripts from MMETSP libraries of *Karenia brevis* and *Karlodinium micrum* were identified by tBLASTn searches using a composite sequence library of predicted plastid-encoded proteins from three haptophyte species (*Emiliania huxleyi*, *Phaeocystis antarctica*, *Pavlova lutheri*) and one kareniacean dinoflagellate (*Karenia mikimotoi*) with threshold e-value 1E-05, following previous studies (Dorrell et al, 2016; Dorrell et al, 2017) (Dataset EV8). Retrieved transcripts were first validated by BLASTx search against the entire NCBI database (accessed December 2017) to exclude bacterial, mitochondrial, chimeric, and artifactual BLAST hits, and then assembled by species and genes into contigs using the in-built sequence assembly function in Geneious vR10.0.9 (Kearse et al, 2012). Transcript sequences were trimmed to remove non-conserved nucleotides and inferred frameshifts prior to conceptual translation with a standard plastid translation table. Identified transcript sequences are provided in Dataset EV8.

cDNA was synthesized from 200 ng DNase-treated *Karenia papillionacea*, *Karlodinium armiger*, and *Takayama helix* RNA using Maxima First Strand cDNA Synthesis Kit for RT-qPCR (Thermo Fisher) following the manufacturer's instructions. cDNAs corresponding to six plastid-encoded marker genes (*psaA*, *psbA*, *psbC*, *psbD*, *rbcL*, and 16 S rDNA/ *rrnS*) sequenced in previous studies of *Karenia mikimotoi* and *Karlodinium micrum* (Dorrell

and Howe, 2012; Richardson et al, 2014) were amplified from each species using DreamTaq PCR MasterMix (Thermo Fisher), following the manufacturer's instructions. Consensus primers (typically: four forward and three reverse primers, with an additional oligo-d(A) reverse primer that anneals to transcript poly(U) tails) were designed from alignments of *Karenia mikimotoi*, *Karenia brevis*, and *Karlodinium micrum* RNA sequences as inferred above (Dorrell et al, 2016), and are provided in Dataset EV8. PCR conditions were typically: 10 min at 95 °C, followed by 35 cycles of 45 s at 95 °C, 45 s at 55 °C, and 2 min at 72 °C, with a terminal 5-min elongation at 72 °C. Different combinations of forward and reverse primer were tested for each gene, and for specific genes, the annealing phase temperature was modified on a gradient between 48 °C and 60 °C.

Products of the expected size, as inferred from agarose gel electrophoresis, were purified using a DNA column spin kit (Macherey-Nagel). In certain cases, products were cloned into pGEM-T Easy vector (Promega) and transformed into competent *E. coli* cells, with inserts amplified using standard T7 and SP6 primers and purified as before. Purified products were Sanger sequenced using each PCR primer under standard conditions (Eurofins genomics). 18S rDNA and ITS1 sequences were amplified and sequenced above, using consensus primers as provided in Gachon et al, 2013 (Dataset EV8). Novel sequences associated with this project are publicly available from GenBank under sequence IDs OQ679836-OQ679852 and OQ680141-OQ680143.

## Data availability

The datasets and computer code produced in this study are available in the following databases: • translated transcriptomic datasets: https://figshare.com/articles/dataset/full_protein_datasets_tar/21647771. • raw reads and nucleotide assemblies: NCBI SRA BioProject ID PRJNA788777 (https://www.ncbi.nlm.nih.gov/bioproject/788777). • trees produced by the automatic pipeline: https://figshare.com/articles/dataset/all-automatically-built-trees_tar/21647768. • alignments produced and used by the automatic pipeline: https://figshare.com/articles/dataset/all-automatically-generated-alignments_rar/24347032. • predicted plastid protein datasets with phylogenetic annotations: https://figshare.com/ articles/dataset/phylo-annotated_proteins_sets/24407914. • genomes and transcriptomes used for homolog mining: https://osf.io/ykxes/. • automatic tree sorting script: https://github.com/vanclcode/treesorter/.

## Peer review information

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

## Acknowledgements

RGD and ANV acknowledge support from a CNRS Momentum Fellowship awarded to RGD, 2019-2021, and an ANR JCJC ("PanArctica", ANR-21-CE02-0014) awarded to RGD, 2021-2022. RGD further acknowledges an ERC Starting Grant ("ChloroMosaic", project number 101039760), awarded 2023-2027. ANV acknowledges support from ANR ("ArchaeArf", ANR-20-CE13-0007), awarded 2020-2024. CB acknowledges funding from the European Research Council (ERC) under the European Union's Horizon 2020 research and innovation program (Diatomic; grant agreement No. 835067), the French Government "Investissements d'Avenir" programs MEMO LIFE (ANR-10-LABX-54) and PSL* Research University (ANR-1253 11-IDEX-0001-02), and from the Agence Nationale de la Recherche (BrownCut; grant agreement ANR-19-CE20-0020). Computational resources were provided by the e-INFRA CZ project (ID:90254), supported by the Ministry of Education, Youth and Sports of the Czech Republic. The authors thank Elisabeth Hehenberger for sharing translated and decontaminated RSD data and consultation regarding their analysis, Lukáš Novák for consultation regarding phylogenomics, and Emmanuel Alastra for aid in media substrate preparation. This article is a contribution #151 to *Tara* Oceans.

## Author contributions

**Anna MG Novák Vanclová**: Conceptualization; Data curation; Software; Formal analysis; Investigation; Visualization; Methodology; Writing—original draft; Writing—review and editing. **Charlotte Nef**: Data curation; Formal analysis; Investigation; Methodology; Writing—original draft. **Zoltan Fussy**: Data curation; Formal analysis; Investigation; Methodology. **Adél Vancl**: Software. **Fuhai Liu**: Data curation; Software; Methodology. **Chris Bowler**: Supervision; Funding acquisition; Validation; Writing—review and editing. **Richard G Dorrell**: Conceptualization; Data curation; Formal analysis; Supervision; Funding acquisition; Validation; Investigation; Methodology; Writing—original draft; Project administration; Writing—review and editing.

## Disclosure and competing interests statement

The authors declare no competing interests.

# Expanded View Figures

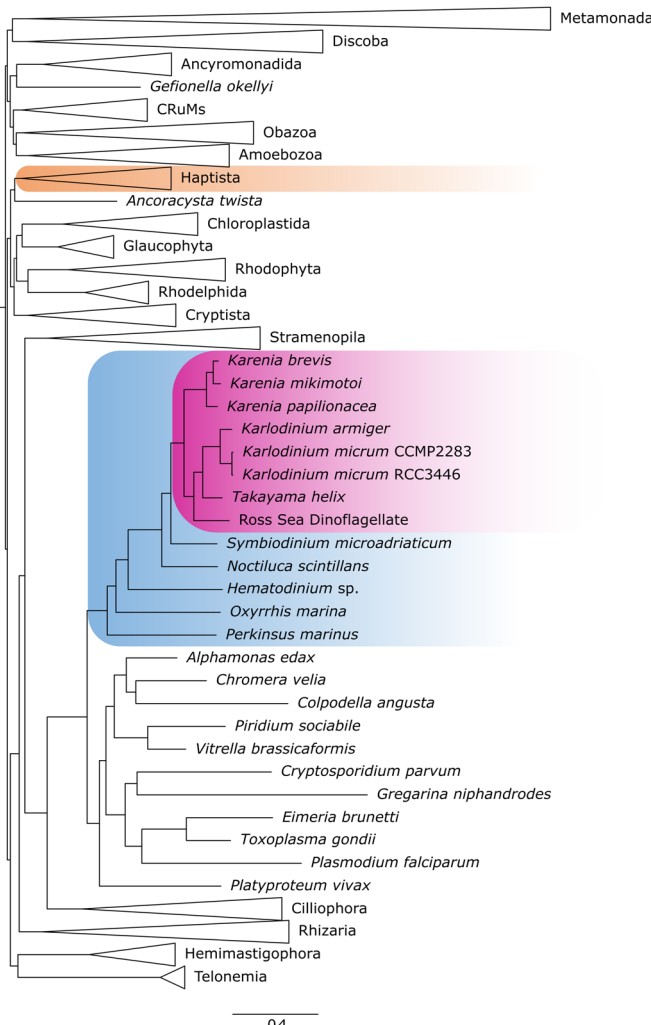

**Figure EV1. The eight kareniaceans in the pan-eukaryotic phylogenetic context as reconstructed by IQ-TREE based on a matrix of 241 genes prepared using PhyloFisher toolkit.**

The inner relationships between the studied kareniaceans are resolved with maximum support and confirm the results of previous phylogenetic analyses based on ribosomal subunits (Takahashi et al, 2019; Ok et al, 2021).

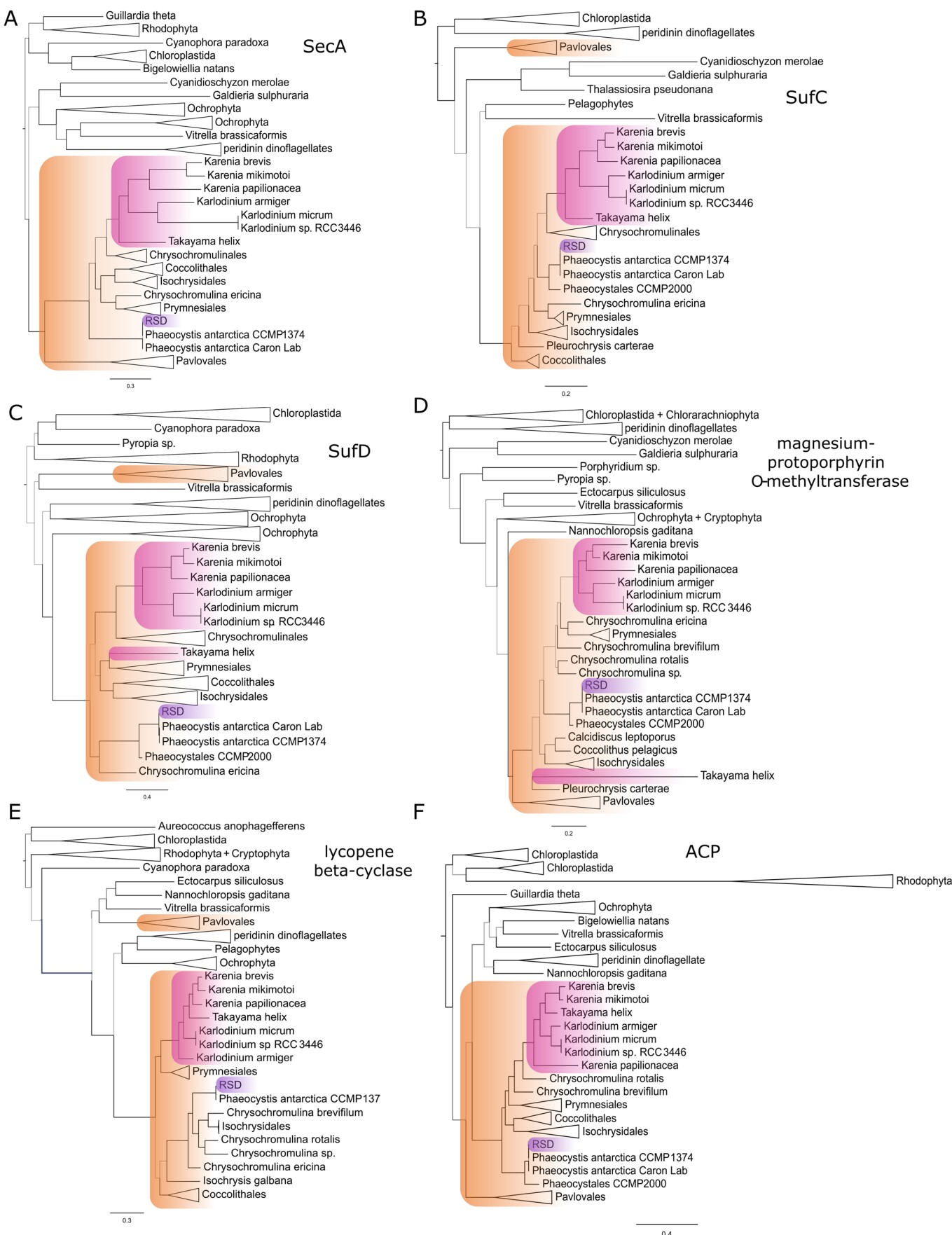

◀ **Figure EV2. Example single-gene trees for proteins of plastid-late origin, with specifically plastidial function and present in all studied organisms.**

The non-RSD kareniacean sequences typically resolve adjacent to Chrysochromulinales or a mixed Chrysochromulinales/Prymnesiales clade. Their inner topology varies and typically differs from the organismal one (Figure EV1), with *Takayama* often branching as sister to the rest (**A**, **B**) and sometimes even separately (**C**, **D**), albeit with low support, or inside the *Karenia/Karlodinium* clade but sister to *Karenia* rather than *Karlodinium* (**E**, **F**). All species of *Karenia* and *Karlodinium* are, however, consistently retrieved as monophyletic, suggesting a common chrysochromulinalean-like origin of their nucleus-encoded plastid protein inventory, which is for the most part, also shared with *Takayama*.

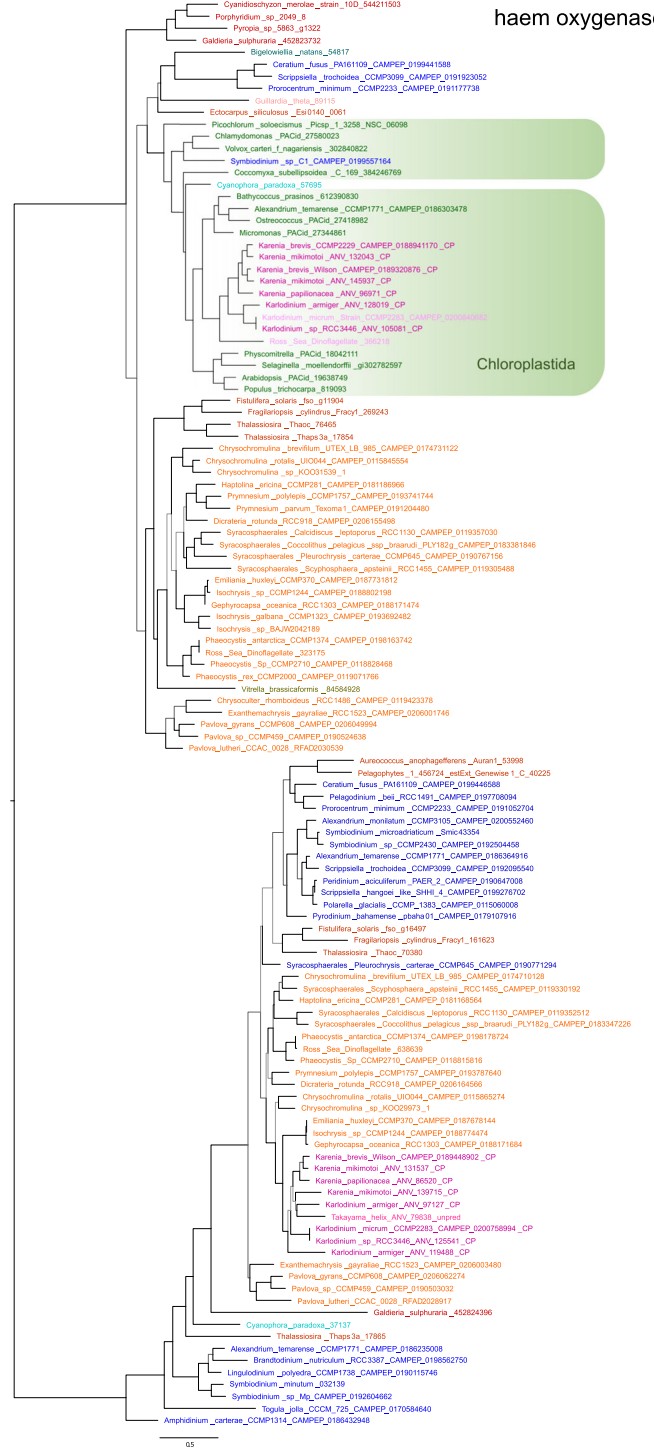

**Figure EV3.** **Automatically generated phylogenetic tree showing the green origin of one form of heme oxygenase in *Karenia*, *Karlodinium*, and the RSD (no *Takayama* homolog was retrieved); the second form is of plastid-late origin in all genera.**

Bootstrap support is expressed by the branch color (black for ≥90%, dark gray for ≥75%, and light gray for <75%). Kareniacean sequences are colored pink with proteins with predicted plastid-targeting signal in darker shades and annotated "CP" suffix; dinoflagellates are coded blue; haptophytes are coded orange; green algae (Chloroplastida) are coded green.

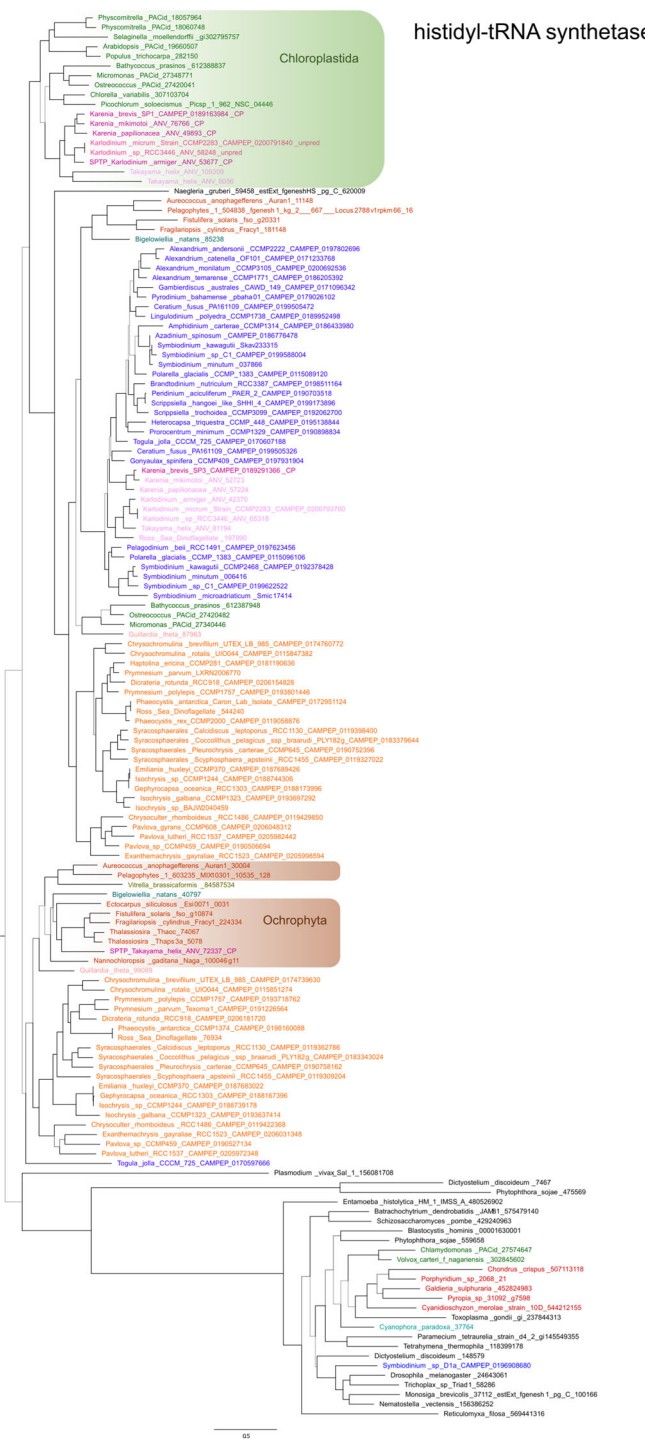

**Figure EV4.  Automatically generated phylogenetic tree showing the green origin of plastidial histidyl-tRNA synthetases of *Karenia* and *Karlodinium* and brown origin of the plastidial homolog in *Takayama*.**

A green-like homolog present in *Takayama* is not predicted as plastidial. The *Karenia brevis* SP3 homolog clustering with the dinoflagellates was likely falsely predicted as plastid-targeted due to an artificial N-terminal extension of the transcript. Bootstrap support is expressed by the branch color (black for ≥90%, dark gray for ≥75%, light gray for <75%). Kareniacean sequences are colored pink with proteins with predicted plastid-targeting signal in darker shades and annotated "CP" suffix; dinoflagellates are coded blue; haptophytes are coded orange; green algae (Chloroplastida) are coded green; and brown algae (Ochrophyta) are coded brown.

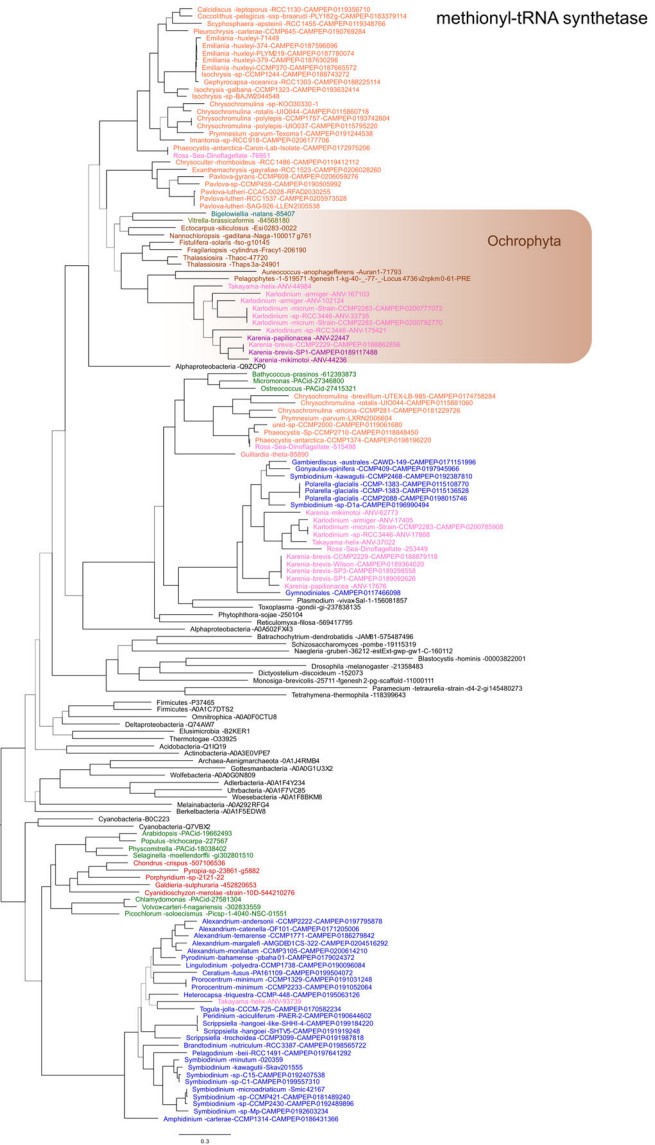

**Figure EV5. Automatically generated phylogenetic tree showing the brown origin of plastidial methionyl-tRNA synthetase in all three genera.**

Bootstrap support is expressed by the branch color (black for ≥90%, dark gray for ≥75%, and light gray for <75%). Kareniacean sequences are colored pink with proteins with predicted plastid-targeting signal in darker shades and annotated "CP" suffix; dinoflagellates are coded blue; haptophytes are coded orange; and brown algae (Ochrophyta) are coded brown.

