## [Peer Review File · EMBO Reports]

New plastids, old proteins: repeated endosymbiotic acquisitions in karenian dinoflagellates

Anna Novák Vanclová, Charlotte Nef, Zoltan Fussy, Adél Vancl, Fuhai Liu, Chris Bowler, and Richard Dorrell

Corresponding author(s): Anna Novák Vanclová (anna.novak-vanclova@ijm.fr, vanclova@gmail.com) , Richard Dorrell (richard.dorrell@sorbonne-universite.fr, richard.dorrell.algae@gmail.com)

Review Timeline:

Transfer Date:	24th Nov 23
Editorial Decision:	28th Nov 23
Revision Received:	9th Dec 23
Editorial Decision:	16th Jan 24
Revision Received:	19th Jan 24
Accepted:	6th Feb 24

Editor: Yehu Moran

Transaction Report: This manuscript was transferred to EMBO reports following peer review at Review Commons.

The logo for Review Commons, featuring the word "Review" in a large, blue, serif font with a diagonal slash through the letter 'v', and the word "COMMONS" in a smaller, blue, sans-serif font below it.

Review #1

1. Evidence, reproducibility and clarity:

Evidence, reproducibility and clarity (Required)

The manuscript investigated the composition of the plastid proteomes of seven distantly-related kareniacean dinoflagellates, including newly-sequenced members of three genera (*Karenia*, *Karlodinium*, and *Takayama*). Using a custom plastid-targeting predictor, automatic single-gene tree building and phylogenetic sorting of plastid-targeted proteins for plastid proteome construction, the authors suggest that the haptophyte order *Chrysochromulinales* is the closest living relative of the fucoxanthin plastid donor. Interestingly, the N-terminal targeting sequences of kareniacean plastid signal peptides, reveal a high sequence conservation. Moreover, ecological and mechanistic factors are suggested that may have driven the endosymbiotic acquisition of the fucoxanthin plastid. Overall, this is a comprehensive and interesting analysis.

****Other comments.****

1. For analyses of N-terminal targeting sequences, why did the authors not consider to employ Predalgo as an additional tool?
2. Given the fact that peridinin or fucoxanthin pigment binding is in the focus of the paper, a more detailed introduction of the peridinin and fucoxanthin light-harvesting systems should be given.
3. The authors state "It is also possible that there has been a direct niche competition between the peridinin and fucoxanthin plastid that may have coexisted in the same host for a period of time with possibly different selective pressure on retention of their respective proteins based on their interaction with plastid-encoded components, e.g., extrinsic photosystem subunits not assembling correctly with their intrinsic haptophyte-like counterparts." It is tempting to ask, whether peridinin light-harvesting systems have left traces in the fucoxanthin plastid, possibly due to mistargeting of peridinin light-harvesting systems into the fucoxanthin plastid? Are some photosynthetic subunits "in-between" peridinin and fucoxanthin plastids?
4. Figure 3 is difficult to understand, e.g. for PSI and PSII which subunits are shown, why has PSI "more" contribution from dinoflagellates as compared to PSII?
5. Data shown in figure 4, is there experimental evidence for signal peptide cleavage site(s). Could these data been used to predict mature plastid targeted protein sequence?
6. The authors state "Partial Least Square (PLS) analysis shows a set of environmental variables (salinity, silicate, iron) positively correlated with abundances of both

Karenia and Takayma and also haptophytes as a whole, but at the same time negatively correlated to Karlodinium (Figure S8), further illustrating that the latter genus is quite distant from the rest in its biogeographical pattern." How could this be interpreted in the light of the plastid proteomes?

2. Significance:

Significance (Required)

The current manuscript gives insights into the endosymbiotic acquisition of the fucoxanthin plastids.

3. How much time do you estimate the authors will need to complete the suggested revisions:

Estimated time to Complete Revisions (Required)

(Decision Recommendation)

Less than 1 month

Yes

Review #2

1. Evidence, reproducibility and clarity:

Evidence, reproducibility and clarity (Required)

This is a well done, detailed bioinformatic analysis of genomic and transcriptomic data from an important lineage of dinoflagellates that have undergone serial substitution of their plastid. On the whole I am enthusiastic about the paper; it presents valuable new insights, and is rigorously performed. However, I have to object to the way the term "proteome" is used in the paper; the manuscript is talking about the predicted proteome, not a measured proteome. This is something of a technical distinction, but it is an important one because the transcriptome and the proteome don't necessarily track each other, and there is little or no actual proteomic data available from dinoflagellates. We assume that transcript abundance has something to do with proteome abundance, but this is often violated. What this paper is really addressing is the potential proteome, because if a given gene is completely absent from the genome and the transcriptome we can be confident it will not be present in the proteome. The converse is not true. For this reason I feel it is important to be clear on the distinction. I would be satisfied in this regard by minor modifications, using the term "predicted proteome" in the title, and being more direct in the introduction about the distinction.

Overall the analyses are impressive. I do have to squirm a little when I see automated analyses generating alignments where the threshold is less than 75% gaps and at least 100 nucleotides aligned. I looked at the supplementary data and the figshare files and could not find the alignments themselves, so I don't know what fraction of the sequences are in that territory. Because phylogenetic analysis (as performed here) treats the alignments as an observation, and because the alignments include sequences with more than 50% gaps, it is entirely possible that some taxa, or even whole segments of the tree, are based on non-overlapping data.

Mind you, we have done similar analyses, and I don't think this invalidates the results, but it does open up the possibility of some dramatic artifacts. Consequently, I would recommend a) making the alignments available (or more obvious where to find them), and b) providing more detail on the alignments, including, if possible, to add a figure (probably in the supplementary data) that visualizes them. It is not given in the text itself, but according to the figure 2 caption there are 22 sequences thought to be "plastid late", and 241 in the pan-eukaryotic dataset. This is a scale that is feasible to put in a figure showing, for example, each aligned residue as a color and indels as grey. Such a figure is readable even when the individual residues are only a few pixels in size (less than a millimeter when printed). I also recommend describing the final alignments more fully in the text. Most of the summary statistics are presented in normalized form, and that can obscure patterns that come from poorly sampled taxa.

Better clarify on the characteristics of the alignments will make it easier to interpret the findings overall. Although this is critical to interpreting the results, gappy

alignments are not uncommon in analyses of this sort, and setting that aside the analyses presented are comprehensive and thorough. The discussion does a good job of addressing the significance of the work, and potential causes of error are addressed adequately (aside from the matter of the alignments).

2. Significance:

Significance (Required)

I find the paper to be exciting and important. These organisms are economically important, particularly as potential nuisance organisms, but also because of their role in primary productivity. They also have extremely complex evolutionary histories and similarly complex genomes. performing any bioinformatic analysis of these organisms is a substantial challenge because almost every gene exists in high copy number and with complex and often obscure patterns of homology. The manuscript brings forward these challenges, and makes a substantial step forward in elucidating the evolution of a group that is fascinating and important, but remarkably difficult to work with. I feel that it is an important analysis, and should be of interest to a broad audience.

3. How much time do you estimate the authors will need to complete the suggested revisions:

Estimated time to Complete Revisions (Required)

(Decision Recommendation)

Between 1 and 3 months

Yes

Review #3

1. Evidence, reproducibility and clarity:

Evidence, reproducibility and clarity (Required)

Summary

This manuscript entitled "Divergent and diversified proteome content across a serially acquired plastid lineage" by Novak Vanclova et al. proposes the origin and evolution of plastids in kareniacean dinoflagellates. The authors generated new transcriptome data from *Karenia mikimotoi*, *Karenia papilionacea*, *Karlodinium micrum*, *Karlodinium armiger*, and *Takayama helix*. Combining them to the previously published transcriptome data from kareniacean dinoflagellates, they constructed the pan-kareniacean transcriptome library. They surveyed plastid-targeted protein-coding transcripts in the dataset, and consequently they estimated ~14.5% of the transcriptome data were of plastid-targeted ones. Of them, 65-80% were derived from a peridinin-containing dinoflagellate ancestor while ~15% were derived from EGTs from a haptophyte endosymbiont of the current plastid origin. By using the plastid-targeted transcript dataset, they investigated 1) origins of the plastid-targeted protein-coding transcripts by single gene-trees, 2) the plastid origin and evolution by the multigene dataset of 22 conserved plastid-targeted protein-coding transcripts and of 3) plastid genome-derived transcripts, 4) plastid functions, 5) diversity of plastid-targeted signals in kareniacean dinoflagellates, and 6) the distributions of kareniacean species by using the Tara Oceans database. On the basis of their results, they proposed many hypotheses regarding kareniacean dinoflagellate evolution, such as i) the chrysochromulinales-origin of the plastids, ii) more recent acquisition of the plastid than previously thought, iii) a plastid replacement within kareniaceae evolution, iv) the strict selection of signal peptides but non-conserved transit peptides in the kareniacean plastid-targeted proteins, and v) correlated or non-correlated distribution patterns of kareniacean dinoflagellates to specific haptophyte lineages.

Although their proposals are interesting, I have many concerns to be addressed. Especially, their analyses on which the above proposals are based seem to be still preliminary and inconclusive. To support their proposals more confidently, I also suggest some additional analyses.

Major comments

1. seemingly inconsistency between the authors' claims

The most striking is inconsistency of the authors' claims proposed in this manuscript. Their proposals include a) the common ancestor of kareniaceans has not possessed a fucoxanthin plastid but the plastid has been acquired more recently, b) an ancestor of Takayama and Karlodinium has gained a fucoxanthin plastid from a (chrysochromulinales) haptophyte, c) an ancestor of *Karenia* has gained a fucoxanthin plastid from *Karlodinium*.

However, they also demonstrate a higher proportion of plastid-late proteins in *Karenia* than *Karlodinium* and *Takayama*.

If I understand correctly, "a higher proportion of plastid-late proteins in *Karenia* than *Karlodinium* and *Takayama*" would seemingly be inconsistent to and challenge two of the authors' claims: no haptophyte-derived plastid in the common ancestor of kareniacean dinoflagellates and a *Karlodinium*-to-*Karenia* plastid transfer (Fig. 7). If the *Karenia* plastid is derived from *Karlodinium*, I have no idea why haptophyte-derived plastid proteome of *Karenia* is larger than that of *Karlodinium*. After the plastid acquisition in *Karenia*, *Karenia* might have gained more genes for plastid-targeted proteins from haptophytes by LGTs. If this is true, many single gene trees would suggest different origins of plastid-targeted proteins between *Karenia* and *Karlodinium*/*Takayama*. Can we see it in the single gene analyses? I would like authors to rationalize the inconsistency in the main text.

2. Signal peptide prediction

I think the modified ASAFind would be greatly helpful for future studies on automatic prediction of plastid proteomes in kareniacean dinoflagellates. However, I found no data on selection criteria for the signal peptide prediction program SignalP5.0 used. I believe such data would be very important to interpret the previously published paper by Gruber et al. in which prediction methods for plastid-targeting sequences are compared to each other to see how sensitively and specifically they can capture the plastid proteomes.

Gruber et al. 2020. Comparison of different versions of SignalP and TargetP for diatom plastid protein predictions with ASAFind.

According to Gruber et al. (2020), signalP5.0 is not suitable for prediction of signal peptides for diatoms, in consistent with the authors' claim for kareniacean dinoflagellates. This inconsistency would be difference of the nature in signal peptides between diatoms and kareniacean dinoflagellates. Even if so, it would be useful to see quantitatively how much different their signal peptides are in terms of their suitable prediction programs.

I also have a concern about use of the combination of PrediSI and ChloroP,

combination which is suitable for the plastid proteome prediction in *Euglena gracilis*. The authors should rationalize why the method for *Euglena* plastids can be applicable without any modification to the plastid proteome prediction in kareniacean dinoflagellates. Although *Euglena* plastids are enclosed by three membranes, kareniacean plastids are by four. Therefore, from the side of molecular mechanisms in protein import, the method suitable for *Euglena* plastids is not necessarily suitable for kareniacean dinoflagellate plastids.

By using PrediSI and ChloroP, they detected additional "candidate plastid proteomes" including several proteins not detectable by SignalP5.0 and the modified ASAFind. That seems great. However, they did not seem to consider false positives since there is no mention on it. Although the additional candidates predicted by PrediSI and ChloroP included true plastid proteins of kareniacean dinoflagellates, many might not be. Nevertheless, the authors suggest 7.5 to 14.5% in *K. micrum* and *K. brevis*, respectively, are of plastid-targeted ones. I am so afraid if the proportions would be highly overestimated due to false positives by PrediSI and ChloroP.

To rationalize the use of PrediSI and ChloroP, the authors should show sensitivity and specificity by quantitative analyses with a benchmark dataset.

3. Origin and evolution of kareniacean plastids

The authors suggest the chrysochromulinales origin of the kareniacean dinoflagellate plastids and the *Karlodinium*-to-*Karenia* plastid transfer, on the basis of phylogenetic analyses using the concatenated datasets with the 22 conserved plastid-targeted proteins and with plastid-genome derived transcripts. It is very interesting that those plastid-targeted proteins in kareniacean dinoflagellates might be phylogenetically closely related to chrysochromulinales haptophyte

I have suggestions on the analyses and interpretation

As the 22 analyzed genes are nuclear-encoded plastid targeted genes, they are a quite small portion of entire plastid proteins. I am not convinced by that evolution of the small number of genes reflects evolution of fucoxanthin plastids of which proteomes are comprised of >1000 proteins.

How many genes for haptophyte-derived plastid-targeted proteins suggest the monophyly of kareniacean dinoflagellates and chrysochromulinales haptophytes should be investigated by, for example, a coalescence-based analysis such as Astral for all the detected haptophyte-derived plastid-targeted proteins including the 22 genes. This is because the monophyly could be reconstructed only by one or few, limited number of proteins even if the concatenated dataset is analyzed.

Relevant to this, plastid-targeted proteins derived from a peridinin-containing ancestor might still have phylogenetic signals of host evolution. I am interested in whether such analyses with peridinin plastid-derived plastid-targeted proteins reconstruct Takayama and *Karlodinium* as monophyletic but separate *Karenia* from them, as

suggested in the phylogenomics with non-plastid proteins.

For the phylogenetic analysis of plastid genome-derived transcripts, I might be wrong, but I could not find any information on dataset sizes (i.e., the numbers of sites) and evolutionary models for the analyses in the main text nor supplementary document. Although one may see the dataset sizes when looking at the original datasets in the supplementary files, such information is substantial and thus is to be described in the materials and methods section. I am afraid if this analysis was performed with a small dataset size. I would like to know total lengths of the concatenated sequences and especially that for Takayama. The phylogenetic position of Takayama, distantly related to the other kareniaceans, in this tree might be caused by a larger portion of gaps in the Takayama sequences than in the other kareniaceans.

Moreover, due to lack of the plastid genome sequence of Takayama, no one could confidently identify plastid genome-derived transcripts: some of those could be derived from second, nuclear copies that might be pseudogenes. Otherwise, even if they are plastid-derived, no one can evaluate whether they are transcripts after or prior to RNA editing. I am afraid if the dataset used is comprised of a mixture of edited and non-edited sequences in kareniacean sequences. Either of sequences after or prior to RNA editing, latter of which are identical with DNA sequences, should be consistently used for the phylogenetic analysis.

In any case, the plastid genomes are necessary for this analysis, and the authors can easily obtain them by DNaseq as they have the cultures.

In addition, although I might be wrong, the phylogenomic analysis for plastid-encoded transcripts might be performed with their nucleotide sequences according to the figure title and legend of Figure S4 mentioning "nucleotide phylogenetic matrix" and the file name "plastid_coded_nt_concatenation_files.tar". If so, translated amino acid sequences should be subjected to phylogenetic analysis, to avoid a well-known artifact that is caused by saturation of substitutions at the 3rd codon.

4. Duplication of an ATP synthase subunit

Duplication and relocation of ATP synthase subunit delta seems interesting. In figure S6.4.1, could you clarify why the possible extensions containing signal peptides lack the initiation methionine at N-termini? I wonder they are 5' UTRs but artifactually detected as signal peptides, if they all indeed lack Met. To evaluate this point, I recommend 5' RACE followed by transformation into a model organism as performed in previous studies by some of the authors.

5. Comparison of transit peptides

Amino acid compositions in transit peptides would vary when targeted compartments are different. In complex plastids, there are functionally distinct compartments: lumen, stroma, periplastidal compartment (PPC). Comparison should therefore be conducted separately for lumen-targeted, stroma-targeted and PPC-targeted proteins

in order to claim their transit peptides are not conserved.

6. RDS never possessed a stable fucoxanthin plastid

Although the authors cite Hehenberger et al. 2019 for that RDS never possessed a stable fucoxanthin plastid, as far as I know, that paper seems not to mention it. Could you let me know where that is mentioned in the paper? Hehenberger et al. instead proposed the retention of non-photosynthetic peridinin plastid.

Regardless of whether Hehenberger et al. mentioned or not, Novák Vanclová et al. propose that RDS never possessed a stable fucoxanthin plastid because, if I understand correctly, they detected no or few haptophyte-derived RDS genes for plastid-targeted proteins of which origins are shared with those of *Karlodinium*, *Karenia*, and *Takayama*. What about the possibility that the last common ancestor of kareniacean dinoflagellates possessed a fucoxanthin plastid in addition to peridinin plastid followed by almost complete losses of those haptophyte-derived genes after loss of a fucoxanthin plastid in evolution leading to RSD? Free living eukaryotes were appeared to have lost a plastid in recent studies and they have only a few or no genes showing evidence of a plastid previously retained. We cannot rule out that an ancestor of kareniacean dinoflagellates possessed both of peridinin and fucoxanthin plastids, as the authors mention in the main text, and either plastid was inherited to each lineage by differential losses. Accordingly, I would say Fig. 7 is a too much strong proposal as alternative hypotheses are still present. They should be introduced equally.

7. rRNA copy numbers in dinoflagellates

It is known that the rRNA gene copy number varies among populations or strains in dinoflagellates; some possess several dozens of times as many rRNA gene copies as others (Galluzzi et al. 2010). Is it informative to see the ocean wide rRNA gene amplicon data for the kareniacean dinoflagellates? The numbers of rRNA gene-derived reads would not necessarily reflect the cell abundance of dinoflagellates.

Galluzzi et al. 2010. Analysis of rRNA gene content in the Mediterranean dinoflagellate *Alexandrium catenella* and *Alexandrium taylori*: implications for the quantitative real-time PCR-based monitoring methods. *J Appl Phycol* 22:1-9

****Minor points****

1. the dataset size for the 241 protein-based host phylogeny should also be described in the main text.
2. The authors mention in Discussion "Thus, our results illuminate the mechanistic of a fundamental process that may under pin vast tracts of chloroplast evolution". If I understand correctly, I think this is based on "shopping bag model" when considering plastid replacements in dinoflagellates. It is helpful to add more details to clarify why the authors would like to claim so. "Chloroplast" should be replaced with

"plastid".

3. Supplementary document S6.6

I found the term nitrogen fixation, but should this be replaced with "nitrogen assimilation"?

4. Figure S5

For those LGTs, all the trees should be shown in supplementary text as they are only 11 or 12 trees. Especially, please add the chlorophyllide b reductase and chlorophyllase in the figure.

5. References

I am not picky about a format of the reference list, but I think it should be consistent throughout the list. I recommend adding journals, volumes, and pages precisely for cited papers. I found lack of them at least in Novak Vanclova et al. and Pierella Karlusich et al.

6. Figures

In figure 3, I strongly recommend adding RDS data, while distinguishing them by another color if they are derived from different origins from those of *Karenia*, *Karlodinium*, and *Takayama*. This would make the authors claim clearer that there are few haptophyte-derived genes for plastid targeted proteins of which origins are shared with those of the other karenian dinoflagellates.

In figures S5.1-2 showing LGTs, I found two paralogs of karenian dinoflagellates. What does "CP" mean? If "CP" means ChloroPlast-targeted, both paralogs of *K. brevis* in HARS and those of *K. micrum* are of plastid-targeted in TARS and they do not have cytosolic ones. I am afraid if these cases are caused by false positives of detection for plastid-targeted proteins by PredSI and ChloroP.

Similarly, in figure S5.4, I found two distant paralogs of heme oxygenase in the tree and the taxon names for both types in karenian dinoflagellates include "CP." Are both targeted to the plastids or of false positives?

2. Significance:

Significance (Required)

General assessment: provide a summary of the strengths and limitations of the study. What are the strongest and most important aspects? What aspects of the study should be improved or could be developed?

This study by Novak Vanclova et al. provide new transcriptome datasets from multiple species in karenian dinoflagellates including harmful and toxic species. Their transcriptome datasets would help understand their biology, evolution, and ecology. The authors also provide a program that predicts plastid proteomes in those dinoflagellates, which would be useful for future studies to focus on karenian

dinoflagellate plastids, after further refinement. The most important aspect of this study is that many plastid-targeted proteins might be derived from a particular haptophyte lineage, although it is still not sure whether they are derived from LGTs or EGTs. Phylogenetic analyses performed in this study should be improved by adding some plastid genomes, in order to gain more conclusive results. In addition to methods, interpretation of the current results and proposals on plastid evolution should be toned-down.

Advance: compare the study to the closest related results in the literature or highlight results reported for the first time to your knowledge; does the study extend the knowledge in the field and in which way? Describe the nature of the advance and the resulting insights (for example: conceptual, technical, clinical, mechanistic, functional,...).

Although there are technical issues, this study improves our conceptual understanding the plastid proteome evolution in Kareniacean dinoflagellates. The plastid proteomes are comprised of proteins with more various origins in those dinoflagellates, suggesting more complex plastid proteome evolution than previously thought.

Audience: describe the type of audience ("specialized", "broad", "basic research", "translational/clinical", etc...) that will be interested or influenced by this research; how will this research be used by others; will it be of interest beyond the specific field?

This study seems to be "basic research".

Please define your field of expertise with a few keywords to help the authors contextualize your point of view. Indicate if there are any parts of the paper that you do not have sufficient expertise to evaluate.

algal evolution, eukaryotic evolution, mitochondrial metabolisms, plastid metabolisms, phylogenomics

3. How much time do you estimate the authors will need to complete the suggested revisions:

Estimated time to Complete Revisions (Required)

(Decision Recommendation)

Between 1 and 3 months

No

Full Revision

Manuscript number: RC-2022-01782

Corresponding author(s): Anna, Novák Vanclová; Richard, Dorrell

1. General Statements

Dear editor,

We appreciate the time and effort that you and the reviewers dedicated to evaluating our manuscript and are grateful for their feedback. We have prepared a new version of the manuscript, integrating additional analyses that have yielded important new results and dramatically increased the significance of our study. The Results and Discussion sections of the manuscript have been heavily rewritten, incorporating multiple new figures, tables, and files.

Much of the novel content within our manuscript relates to an insightful suggestion from Reviewer 3 to consider the evolutionary origins of the genomes, as well as nucleus-encoded proteomes of different kareniacean plastids. By performing a new phylogenetic analysis (Fig. 5) incorporating sequenced plastid marker genes from each of our studied species, we show with robust support that the plastids of *Takayama helix* and *Karlodinium armiger* come from different haptophyte lineages, respectively in the orders Phaeocystales and Prymnesiales, to the deeply-positioned haptophyte-derived plastids in *Karenia* and *Karlodinium micrum*. Taking into account the previously-characterised kleptoplasts of the kareniacean Ross Sea Dinoflagellate (Hehenberger et al., *PNAS* 2019) these data indicate the independent acquisition of haptophyte plastids at least four times within the kareniaceae, sitting alongside established cases of repeated and independent plastid acquisitions in other dinoflagellate orders (Yamada et al., *MBE* 2017; Sarai et al., *PNAS* 2019). We propose multiple replacements and serial transfers of the fucoxanthin plastid lineage across these kareniaceae, summarised in a revised Fig. 8.

In our initial paper submission, we demonstrated that much of the nucleus-encoded and plastid-targeted kareniacean proteome is shared across species despite important lineage- and pathway-specific variation (Figs. 1 and 3). The discordant but complementary origins of the fucoxanthin plastid underline that Kareniaceae are highly chimeric organisms whose nuclear and plastid genomes, and plastid proteomes undertook different evolutionary trajectories even at micro-evolutionary scales. We further explore the consequence of this variability at cellular and ecosystem levels (Figs. 6-7). We believe this revised manuscript presents a landmark addition to the complex history of the ever-changing dinoflagellate plastids, and provides us with unprecedented insights into the different genetic consequences of post-endosymbiotic organelle evolution, and will benefit from publication in a widely-read and interdisciplinary journal within the *Review Commons* portfolio, such as *PLoS Biology*, *EMBO J*, *Mol Syst Biol*, or *eLife*.

We thank you in advance for your appraisal and look forward to hearing from you shortly.

Anna M.G. Novák Vanclová, and Richard G. Dorrell

Please see below for the detailed description of the changes made in response to the reviewers' comments (in blue, for better legibility).

Reviewer #1 (Evidence, reproducibility and clarity (Required)):

The manuscript investigated the composition of the plastid proteomes of seven distantly-related karenian dinoflagellates, including newly-sequenced members of three genera (*Karenia*, *Karlodinium*, and *Takayama*). Using a custom plastid-targeting predictor, automatic single-gene tree building and phylogenetic sorting of plastid-targeted proteins for plastid proteome construction, the authors suggest that the haptophyte order Chrysochromulinales is the closest living relative of the fucoxanthin plastid donor. Interestingly, the N-terminal targeting sequences of karenian plastid signal peptides, reveal a high sequence conservation. Moreover, ecological and mechanistic factors are suggested that may have driven the endosymbiotic acquisition of the fucoxanthin plastid. Overall, this is a comprehensive and interesting analysis.

Other comments.

1. For analyses of N-terminal targeting sequences, why did the authors not consider to employ Predalgo as an additional tool?

Author response: We thank the reviewer for their suggestion. To our understanding, PredAlgo is a targeting predictor trained on primary green algae, which have two-membrane bound plastids and purely hydrophilic N-terminal plastid targeting sequences. It thus would be expected to perform poorly for the prediction of N-terminal targeting sequences in complex plastids such as those of the Kareniaceae bound by three or more membranes, who are located within endomembrane-derived compartments and which utilise plastid-targeting sequences based on an N-terminal hydrophobic signal peptide for ER import.

We considered the application of PredAlgo for the identification of downstream hydrophilic transit peptide regions in Karenian presequences, but note that the specific residue positioned after the signal peptidase cleavage site is typically a much better predictor than transit peptide hydrophobicity for identifying plastid-targeting sequences (Gruber et al., *Plant J* 2015, and citing references). We found that other targeting prediction tools based primarily on hydrophobicity (e.g., HECTAR) performed poorly in identifying probable plastid-targeting sequences in our control Karenian dataset, and therefore chose to prioritise a modified version of ASAFind that takes into account the residue context of Karenian signal peptidase cleavage site for our targeting predictor, which works with high sensitivity and specificity on our control dataset. We summarise these observations in Fig. S15.

2. Given the fact that peridinin or fucoxanthin pigment binding is in the focus of the paper, a more detailed introduction of the peridinin and fucoxanthin light-harvesting systems should be given.

Author response: A brief introduction to the pigment-binding proteins in dinoflagellates was added, "These include a unique carotenoid pigment... massively paralogized and synthesized as polyproteins" (lines 86-89).

3. The authors state "It is also possible that there has been a direct niche competition between the peridinin and fucoxanthin plastid that may have coexisted in the same host for a period of time with possibly different selective pressure on retention of their respective proteins based on their interaction with plastid-encoded components, e.g., extrinsic photosystem subunits not assembling correctly with their intrinsic haptophyte-like counterparts." It is tempting to ask, whether peridinin light-harvesting systems have left traces in the fucoxanthin plastid, possibly due to mistargeting of peridinin light-harvesting systems into the fucoxanthin plastid? Are some photosynthetic subunits "in-between" peridinin and fucoxanthin plastids?

Author response: We did not identify any other peridinin-like photosystem subunits than the ones visualized in the map schematic (i.e., ferredoxin/PetF in both *Karenia* and *Karlodinium* and Psad of *Karlodinium micrum*) and discussed in the supplementary text. PetF is the only consistently retained peridinin-like photosystem protein, likely due to the fact that it is not strictly linked to photosynthesis: it is expressed in plant leucoplasts, and plastid-encoded in some non-photosynthetic chrysophytes. We have added a sentence in Supporting Text 6.4 that "we detect no possible homologues of peridinin-chlorophyll binding proteins (PCP) in any karenian transcriptome" (line 91).

4. Figure 3 is difficult to understand, e.g. for PSI and PSII which subunits are shown, why has PSI "more" contribution from dinoflagellates as compared to PSII?

Author response: The photosystem subunits are ordered numerically in the schematic, and detailed information on each protein and the corresponding sequences with their origin are included in the supplementary table S3. A single subunit of photosystem I (PsaD) was determined to be of plastid-early (peridinin-like) origin in *Karlodinium* (while the same protein is plastid-encoded in *Karenia* and undetermined in *Takayama*). We believe this may be simply due to an evolutionarily neutral differential loss / non-adaptive retention of photosynthesis-related proteins in a secondarily non-photosynthetic host before the acquisition of a replacement plastid. We note that there are only two (incomplete) karenian plastid genomes available so we cannot rule out the possibility of this subunit being plastid-encoded in *Karlodinium* as well (which would mean that both plastid-late and plastid-early homologs co-occur in this genus).

Fig. 3 is necessarily complex due to the size and multiplicity of the dataset considered. To facilitate reader navigation, we have added the following text to the figure legend (lines 1128-1140) text "Plastid proteins are arranged by major metabolic pathway or biological process, with each protein shown as rosettes ... Proteins of plastid-late (haptophyte) origin, such as are concentrated in photosystem and ribosomal processes, are coloured red; and proteins of plastid-early (dinoflagellate) origin, such as are concentrated in carbon and amino acid metabolism are coloured blue. ... In certain cases (shown as rosettes with multiple colours), homologues from different species have different evolutionary origins, e.g. *Karenia* possessing plastid-late and *Karlodinium/Takayama* plastid-early".

5. Data shown in figure 4, is there experimental evidence for signal peptide cleavage site(s). Could these data be used to predict mature plastid targeted protein sequence?

Author response: We were able to determine the conserved motives in signal peptide, including its cleavage site (GRR) which we exploited in the design of kareniaceae-specific matrix for ASAFind. We show these residues in Fig. 4. We note that these motifs were identified based on homology to known signal processing peptidase recognition sites, as opposed to experimentally determined protein N-termini.

Consistent with previous studies (e.g. Yokoyama et al., *J Phycol* 2011) we see limited evidence for consensus plastid transit peptide cleavage motifs in kareniacean presequences, and do not discuss this further as a result.

6. The authors state "Partial Least Square (PLS) analysis shows a set of environmental variables (salinity, silicate, iron) positively correlated with abundances of both *Karenia* and *Takayma* and also haptophytes as a whole, but at the same time negatively correlated to *Karlodinium* (Figure S8), further illustrating that the latter genus is quite distant from the rest in its biogeographical pattern." How could this be interpreted in the light of the plastid proteomes

Author response: We believe that this may be due to the more cosmopolitan distribution of *Karlodinium*, and possibly also a result of bias stemming from our strategy of grouping the organisms at the genus level (as not enough data was available at species level) which may obscure the potential outlier status of only some species/ subpopulations. This is particularly true for the haptophytes, where in the absence of specific ancestry for individual kareniacean plastids we are only able to consider distributions at the levels of entire orders. We now acknowledge this in the Discussion: "specific ecological interactions between the progenitors ... via ancestral niche reconstruction for each lineage" (lines 473-475).

Please note, that the results might have changed slightly from the previous version due to the recalculation following additional normalization of the data (see below).

Reviewer #1 (Significance (Required)):

The current manuscript gives insights into the endosymbiotic acquisition of the fucoxanthin plastids.

Reviewer #2 (Evidence, reproducibility and clarity (Required)):

This is a well done, detailed bioinformatic analysis of genomic and transcriptomic data from an important lineage of dinoflagellates that have undergone serial substitution of their plastid. On the whole I am enthusiastic about the paper; it presents valuable new insights, and is rigorously performed. However, I have to object to the way the term "proteome" is used in the paper; the manuscript is talking about the predicted proteome, not a measured proteome. This is something of a technical distinction, but it is an important one because the transcriptome and the proteome don't necessarily track each other, and there is little or no actual proteomic data available from dinoflagellates. We assume that transcript abundance has something to do with proteome abundance, but this is often violated. What this paper is really addressing is the potential proteome, because if a given gene is completely absent from the genome and the transcriptome we can be confident it will not be present in the proteome. The converse is not true. For this reason I feel it is important to be clear on the distinction. I would be satisfied in this regard by minor

Full Revision

modifications, using the term "predicted proteome" in the title, and being more direct in the introduction about the distinction.

Author response: We agree that the usage of the word proteome for *in silico* predictions is not entirely correct, and have used the term "predicted proteome" where possible in the text to clarify this.

We have also, as described in our response to Reviewer 1 above, included a statement in the Discussion that our largely bioinformatic results will be transformed by an experimentally realised kareniacean plastid proteome, which we nonetheless feel goes beyond the scope of our manuscript.

Overall the analyses are impressive. I do have to squirm a little when I see automated analyses generating alignments where the threshold is less than 75% gaps and at least 100 nucleotides aligned. I looked at the supplementary data and the figshare files and could not find the alignments themselves, so I don't know what fraction of the sequences are in that territory. Because phylogenetic analysis (as performed here) treats the alignments as an observation, and because the alignments include sequences with more than 50% gaps, it is entirely possible that some taxa, or even whole segments of the tree, are based on non-overlapping data.

Author response: We thank the reviewer for their comment and have added in three new supplementary figures (S16-S18) providing statistics on alignment size, length, and average gap percentage distribution. We report that most of the alignments contained relatively little gaps: 90% of the alignments contained between 1.1 and 24.5% of gaps with median value of 6.6%.

Mind you, we have done similar analyses, and I don't think this invalidates the results, but it does open up the possibility of some dramatic artifacts. Consequently, I would recommend a) making the alignments available (or more obvious where to find them), and b) providing more detail on the alignments, including, if possible, to add a figure (probably in the supplementary data) that visualizes them. It is not given in the text itself, but according to the figure 2 caption there are 22 sequences thought to be "plastid late", and 241 in the pan-eukaryotic dataset. This is a scale that is feasible to put in a figure showing, for example, each aligned residue as a color and indels as grey. Such a figure is readable even when the individual residues are only a few pixels in size (less than a millimeter when printed). I also recommend describing the final alignments more fully in the text. Most of the summary statistics are presented in normalized form, and that can obscure patterns that come from poorly sampled taxa. Better clarity on the characteristics of the alignments will make it easier to interpret the findings overall. Although this is critical to interpreting the results, gappy alignments are not uncommon in analyses of this sort, and setting that aside the analyses presented are comprehensive and thorough. The discussion does a good job of addressing the significance of the work, and potential causes of error are addressed adequately (aside from the matter of the alignments).

Author response: We thank the reviewer for their comment and have provided alignments for all single-gene trees, in a dedicated online supporting repository (https://figshare.com/articles/dataset/all-automatically-generated-alignments_rar/24347032). The datasets and alignments used for PhyloFisher and plastid-encoded gene trees are included directly in the supplementary files (phylofisher_files.tar, plastid_genome_phylogeny_files.tar and plastid_protein_phylogeny_files.tar).

We have additionally included three new supporting figures (S16-S18) showing the distributions of lengths, gaps and homologues in each single-gene tree. These data project largely completion of individual alignments, with only 5% containing > 20% gapped positions (see Fig. S18), for example. We have additionally clarified in the Methods that “The trimmed alignments were then filtered by a custom python script that discarded sequences comprising of more than 75% gaps and then rejected alignments shorter than 100 positions or containing fewer than 10 taxa.” (lines 571-573).

For the two concatenated trees presented, we have clarified in the Methods the alignment lengths (PhyloFisher: 72, 162 positions; plastid genes: 2,404 positions), and that we removed sequences containing >66% of gaps from the final alignment. Reflecting on the congruency assumptions required to concatenated alignments, we have chosen to replace the plastid-late concatenated tree (which may group proteins with multiple phylogenetic signals) with a new main text figure 2 providing an overview of the plastid signals we observe across the entire dataset (see comments below to Reviewer 3).

Reviewer #2 (Significance (Required)):

I find the paper to be exciting and important. These organisms are economically important, particularly as potential nuisance organisms, but also because of their role in primary productivity. They also have extremely complex evolutionary histories and similarly complex genomes. performing any bioinformatic analysis of these organisms is a substantial challenge because almost every gene exists in high copy number and with complex and often obscure patterns of homology. The manuscript brings forward these challenges, and makes a substantial step forward in elucidating the evolution of a group that is fascinating and important, but remarkably difficult to work with. I feel that it is an important analysis, and should be of interest to a broad audience.

Reviewer #3 (Evidence, reproducibility and clarity (Required)):

Summary

This manuscript entitled "Divergent and diversified proteome content across a serially acquired plastid lineage" by Novak Vanclova et al. proposes the origin and evolution of plastids in kareniacean dinoflagellates. The authors generated new transcriptome data from *Karenia mikimotoi*, *Karenia papilionacea*, *Karlorodinium micrum*, *Karlorodinium armiger*, and *Takayama helix*. Combining them to the previously published transcriptome data from kareniacean dinoflagellates, they constructed the pan-kareniacean transcriptome library. They surveyed plastid-targeted protein-coding transcripts in the dataset, and consequently they estimated ~14.5% of the transcriptome data were of plastid-targeted ones. Of them, 65-80% were derived from a peridinin-containing dinoflagellate ancestor while ~15% were derived from EGTs from a haptophyte endosymbiont of the current plastid origin. By using the plastid-targeted transcript dataset, they investigated 1) origins of the plastid-targeted protein-coding transcripts by single gene-trees, 2) the plastid origin and evolution by the multigene dataset of 22 conserved plastid-targeted protein-coding transcripts and of 3) plastid genome-derived transcripts, 4) plastid functions, 5) diversity of plastid-targeted signals in kareniacean dinoflagellates, and 6) the distributions of kareniacean species by using the Tara Oceans database. On the basis of their results, they proposed many hypotheses regarding kareniacean dinoflagellate evolution, such as i) the chrysochromulinales-origin of the plastids, ii) more recent acquisition of the plastid than previously thought, iii) a plastid replacement within

kareniaeeae evolution, iv) the strict selection of signal peptides but non-conserved transit peptides in the kareniaeeae plastid-targeted proteins, and v) correlated or non-correlated distribution patterns of kareniaeeae dinoflagellates to specific haptophyte lineages.

Although their proposals are interesting, I have many concerns to be addressed. Especially, their analyses on which the above proposals are based seem to be still preliminary and inconclusive. To support their proposals more confidently, I also suggest some additional analyses.

Major comments

1. seemingly inconsistency between the authors' claims

The most striking is inconsistency of the authors' claims proposed in this manuscript. Their proposals include a) the common ancestor of kareniaeeae has not possessed a fucoxanthin plastid but the plastid has been acquired more recently, b) an ancestor of Takayama and Karlodinium has gained a fucoxanthin plastid from a (chrysochlorulinales) haptophyte, c) an ancestor of Karenia has gained a fucoxanthin plastid from Karlodinium.

However, they also demonstrate a higher proportion of plastid-late proteins in Karenia than Karlodinium and Takayama.

If I understand correctly, "a higher proportion of plastid-late proteins in Karenia than Karlodinium and Takayama" would seemingly be inconsistent to and challenge two of the authors' claims: no haptophyte-derived plastid in the common ancestor of kareniaeeae dinoflagellates and a Karlodinium-to-Karenia plastid transfer (Fig. 7). If the Karenia plastid is derived from Karlodinium, I have no idea why haptophyte-derived plastid proteome of Karenia is larger than that of Karlodinium. After the plastid acquisition in Karenia, Karenia might have gained more genes for plastid-targeted proteins from haptophytes by LGTs. If this is true, many single gene trees would suggest different origins of plastid-targeted proteins between Karenia and Karlodinium/Takayama. Can we see it in the single gene analyses? I would like authors to rationalize the inconsistency in the main text.

Author response: We agree with the reviewer that the evolutionary origins and dynamics of the kareniaeeae plastid proteome are complex, and thank them for their suggestion.

First, to take into account the different evolutionary scenarios that could explain the present-day distribution of the kareniaeeae plastids, including the new plastid genome sequences identified in response to the reviewer's suggestions, we have made a revised version of Fig. 8 evaluating three different hypotheses (see below). Nonetheless, we feel that the *Karlodinium-to-Karenia* model we propose is plausible, based on the following observations:

- We identify 1,418 plastid protein gene trees in which at least two of the three studied genera (*Karenia*, *Karlodinium*, *Takayama*), and 748 in which all three resolve as monophyletic, and with a haptophyte sister-group (i.e., a common plastid-late origin; Fig. S2). This points to a common haptophyte ancestry in all three groups, as opposed to independent endosymbiotic consumptions of free-living haptophytes in *Karenia* and *Karlodinium micrum*.
- We see no such shared signal with the RSD, which shares only 42 proteins with at least two other kareniaeeae genera (Fig. S4). Thus, and consistent with previous studies (Hehenberger et al., *PNAS* 2019) we cannot invoke an ancestral presence of a fucoxanthin plastid shared with the

RSD in the last common karenian ancestor. This discrepancy thus likely points to a serial transfer of the karenian plastid from either *Karodinium* into *Karenia* or vice versa (Fig. 8).

- Concerning the direction of this transfer, among 1,059 gene trees of plastid-late origin found in both *Takayama*, *Karenia* and *Karodinium*, 873 place *Takayama* as basal to a monophyletic clade of *Karenia* and *Karodinium*, i.e. support a specific plastid transfer between the latter two genera. The most parsimonious explanation for this is the origin of the fucoxanthin plastid in the common *Takayama*/*Karodinium* ancestor, which was subsequently transferred into *Karenia*.

It is true that *Karenia* contains both a greater absolute proportion of predicted plastid-targeted proteins (Fig. 1) and greater number of unique KO number annotations (Table S4) of plastid-late origin than either *Karodinium* or *Takayama*. That said, this signal may be influenced by multiple other factors beyond how old the given endosymbiosis is (i.e., longer coexistence implies more EGT). For example, the number of plastid-late gene in a host genome may depend on the frequency of duplication of plastid-late genes and the receptiveness of the host nuclear genome to incoming horizontally derived genes. It may further be influenced by the presence and relative selective advantage or disadvantage of competing genes of host nuclear origin (i.e. plastid-early genes) that may be differentially selected over plastid-late genes, which might vary between *Karenia* and *Karodinium* due to differential retention of the ancestral peridinin-type plastid in each lineage.

We have elaborated on this point in the Discussion, noting that there may have been “a direct niche competition between the peridinin and fucoxanthin plastid ... with possibly different selective pressure on retention of individual imported proteins” (lines 370-372), “relatively recent origin and spread throughout the karenian genome, e.g., via gene duplications” (line 459), and finally that precedent for divergent evolutionary trajectories in different Karenianae exists from the *Karenia* and *Karodinium* plastid genomes that “contain partially non-overlapping sets of genes that suggest independent post-endosymbiotic plastid genome reduction” (lines 403-404). Nonetheless, we acknowledge that the evolutionary model we propose is not definitive, and that alternative explanations may find more favour with increased genome data.

2. Signal peptide prediction

I think the modified ASAFind would be greatly helpful for future studies on automatic prediction of plastid proteomes in karenian dinoflagellates. However, I found no data on selection criteria for the signal peptide prediction program SignalP5.0 used. I believe such data would be very important to interpret the previously published paper by Gruber et al. in which prediction methods for plastid-targeting sequences are compared to each other to see how sensitively and specifically they can capture the plastid proteomes.

Gruber et al. 2020. Comparison of different versions of SignalP and TargetP for diatom plastid protein predictions with ASAFind.

According to Gruber et al. (2020), signalP5.0 is not suitable for prediction of signal peptides for diatoms, in consistent with the authors' claim for karenian dinoflagellates. This inconsistency would be difference of the nature in signal peptides between diatoms and karenian dinoflagellates. Even if so, it would be useful to see quantitatively how much different their signal peptides are in terms of their suitable prediction programs.

Author response: In our preliminary benchmarking using only the previously published transcriptomes (see additional sheet in Supplementary tables), SignalP 5.0 performed substantially better in terms of specificity than SignalP 3.0 (i.e., 22 versus 34/ 728 retrieved positive hits of proteins with uniquely non-plastidial functions), with comparable sensitivity in the correct prediction of positive control proteins. Given the size of our dataset, and the substantial risk of false positive detection in the highly expanded and redundant dinoflagellate transcriptomes we have used, we feel that the greater specificity of SignalP 5.0 is important to integrate in our model selection. We have clarified this position in the Methods, stating “First, the relative effectiveness of two SignalP versions ... SignalP 5.0 was used for all subsequent analysis.” (lines 525-529).

I also have a concern about use of the combination of PrediSI and ChloroP, combination which is suitable for the plastid proteome prediction in *Euglena gracilis*. The authors should rationalize why the method for *Euglena* plastids can be applicable without any modification to the plastid proteome prediction in kareniacean dinoflagellates. Although *Euglena* plastids are enclosed by three membranes, kareniacean plastids are by four. Therefore, from the side of molecular mechanisms in protein import, the method suitable for *Euglena* plastids is not necessarily suitable for kareniacean dinoflagellate plastids.

By using PrediSI and ChloroP, they detected additional "candidate plastid proteomes" including several proteins not detectable by SignalP5.0 and the modified ASAFind. That seems great. However, they did not seem to consider false positives since there is no mention on it. Although the additional candidates predicted by PrediSI and ChloroP included true plastid proteins of kareniacean dinoflagellates, many might not be. Nevertheless, the authors suggest 7.5 to 14.5% in *K. micrum* and *K. brevis*, respectively, are of plastid-targeted ones. I am so afraid if the proportions would be highly overestimated due to false positives by PrediSI and ChloroP.

To rationalize the use of PrediSI and ChloroP, the authors should show sensitivity and specificity by quantitative analyses with a benchmark dataset.

Author response: We thank the author for this comment. The reasoning behind using the parallel PrediSI+ChloroP strategy was the previously reported similarity of the plastid signal structure between euglenids and peridinin dinoflagellates (c.f., Lukes et al., *PNAS*, 2009) and the previous observation that some kareniaceae possess plastid-targeting sequences resembling those of peridinin dinoflagellates (c.f., Hehenberger et al., *PNAS*, 2019). Per the reviewers' suggestion, we present a modified sensitivity/ specificity testing PrediSI+ChloroP, alongside other alternative targeting predictors in Figure S15. While the PrediSI+ChloroP sensitivity is very low, its specificity is comparable with the modified ASAFind, and in this regard outperforms other targeting predictor tools, thus rationalising the use of both targeting prediction tools together.

3. Origin and evolution of kareniacean plastids

The authors suggest the chrysochromulinales origin of the kareniacean dinoflagellate plastids and the Karlodinium-to-Karenia plastid transfer, on the basis of phylogenetic analyses using the concatenated datasets with the 22 conserved plastid-targeted proteins and with plastid-genome derived transcripts. It is very interesting that those plastid-targeted proteins in kareniacean dinoflagellates might be phylogenetically closely related to chrysochromulinales haptophyte

I have suggestions on the analyses and interpretation

As the 22 analyzed genes are nuclear-encoded plastid targeted genes, they are a quite small portion of entire plastid proteins. I am not convinced by that evolution of the small number of genes reflects evolution of fucoxanthin plastids of which proteomes are comprised of >1000 proteins.

How many genes for haptophyte-derived plastid-targeted proteins suggest the monophyly of kareniacean dinoflagellates and chrysochromulinales haptophytes should be investigated by, for example, a coalescence-based analysis such as Astral for all the detected haptophyte-derived plastid-targeted proteins including the 22 genes. This is because the monophyly could be reconstructed only by one or few, limited number of proteins even if the concatenated dataset is analyzed.

Relevant to this, plastid-targeted proteins derived from a peridinin-containing ancestor might still have phylogenetic signals of host evolution. I am interested in whether such analyses with peridinin plastid-derived plastid-targeted proteins reconstruct Takayama and Karlodinium as monophyletic but separate Karenia from them, as suggested in the phylogenomics with non-plastid proteins.

Author response: We agree with the reviewer concerning the problematic nature of concatenations with small numbers of genes, particularly if the underlying gene trees are not phylogenetically congruent to one another, and have chosen to replace the concatenation with a more global evaluation of the different plastid protein origins across our entire dataset. Using automated sorting approaches, we have evaluated the support for our evolutionary model across hundreds of gene trees. We feel that this approach supercedes coalescence-based techniques, as it enables us to treat each gene topology as an independent event, and to consider multiplicity in the origin of the kareniacean plastid proteome. We present these data in a new Fig. 2 and S2.

As stated above, these data strongly support monophyly of all three Kareniacean genera. Concerning the potential Chrysochromulinales plastid signal in our dataset, we have reanalysed our data and quantify a substantial number of trees (220/ 1,418 of plastid-late origin) that specifically place multiple kareniacean genera within the Chrysochromulinales. This figure is more than twice the number (91) that place the kareniaceae with the next most occurrent haptophyte group in our dataset, Isochrysidales. We nonetheless have chosen to no longer present this as a cryptic plastid endosymbiosis, in the absence of clear examples of extant kareniaceae still possessing this plastid, saying purely in the Discussion that “a common ancestor of the studied organisms either possessed a stable plastid or had a long-term symbiotic relationship (e.g., kleptoplastidic) with a haptophyte lineage related to the extant Chrysochromulinales” (lines 363-365).

Concerning the phylogenetic placement of each kareniacean genus, the majority of our plastid-late trees specifically recover the monophyly of *Karenia* and *Karlodinium*. Remarkably, we find that *Takayama* and *Karlodinium* only resolve together in 69/ 1,039 plastid-late gene trees in which all three genera are represented, strongly refuting a vertical origin of the haptophyte-derived components of their plastid proteome. This is not due to the Phaeocystales origin of the current *Takayama* plastid genome, which is found in only 21 of our plastid protein trees. Nonetheless, as the reviewer suggests, the opposite trend (1,505/ 2,804 gene trees grouping *Takayama* and *Karlodinium* as monophyletic) was observed amongst plastid-early gene trees, which might reflect a cryptic peridinin plastid shared between these groups. We expand on these results in the Discussion, stating “Many of the plastid-early gene trees copy the organismal topology ...this awaits structural confirmation via microscopy” (lines 383-386).

Finally, to enable reviewer comprehension of the relationships shown, we have presented some exemplar topologies of some of the trees previously displayed in the concatenation, provided in a new Fig. S5.2.

For the phylogenetic analysis of plastid genome-derived transcripts, I might be wrong, but I could not find any information on dataset sizes (i.e., the numbers of sites) and evolutionary models for the analyses in the main text nor supplementary document. Although one may see the dataset sizes when looking at the original datasets in the supplementary files, such information is substantial and thus is to be described in the materials and methods section. I am afraid if this analysis was performed with a small dataset size. I would like to know total lengths of the concatenated sequences and especially that for Takayama. The phylogenetic position of Takayama, distantly related to the other kareniaceans, in this tree might be caused by a larger portion of gaps in the Takayama sequences than in the other kareniaceans.

Author response: As noted in our response to Reviewer 2, we have included three new supplementary figures (S16-S18) with statistics on alignment size, length, and average gap percentage distribution. The average and median values of these three measurements do not differ significantly when calculated separately for different organisms. We have clarified in the Methods that the concatenated alignments retained (PhyloFisher, and plastid-encoded genes) were “constructed by IQ-TREE with the LG+C60+F model for the plastid matrices and posterior mean site frequency (PMSF) model (LG+C60+F+G with a guide tree constructed with C20) for PhyloFisher matrix” (lines 630-632).

Moreover, due to lack of the plastid genome sequence of Takayama, no one could confidently identify plastid genome-derived transcripts: some of those could be derived from second, nuclear copies that might be pseudogenes. Otherwise, even if they are plastid-derived, no one can evaluate whether they are transcripts after or prior to RNA editing. I am afraid if the dataset used is comprised of a mixture of edited and non-edited sequences in kareniacean sequences. Either of sequences after or prior to RNA editing, latter of which are identical with DNA sequences, should be consistently used for the phylogenetic analysis.

In any case, the plastid genomes are necessary for this analysis, and the authors can easily obtain them by DNaseq as they have the cultures.

Author response: We thank the Reviewer for their insightful response. We agree that understanding the evolution of kareniacean plastid genomes are crucial to understanding their evolutionary history.

We have accordingly, as described above, integrated a new main text Fig. 5 building a concatenated tree of plastid marker genes (*psbA*, *psch*, *psbD*, *psaA*, *rbcL*, and 16S rDNA) historically and commonly used to assess the evolutionary origins of fucoxanthin plastids (e.g., Takishita et al., *Phycol Res* 1999; Dorrell and Howe, *PNAS* 2012). These sequences were amplified cryopreserved stocks of total RNA and specific primers, amplified by RT-PCR. We have chosen here to use RNA sequences, to account for the presence of plastid RNA editing, which has been shown to play an important role in maintaining sequence identity between kareniacean plastids and haptophyte relatives despite a high DNA mutation rate in the former (Jackson et al., *MBE* 2013; Klinger et al., *GBE* 2018), rather than DNA sequences for this analysis.

Additionally, we would like to note that while plastid genomes are generally relatively simple to sequence and assemble, this is not the case in Kareniaceae. The existing plastid genome assemblies are partially incomplete and suggest more complex and possibly unstable structures (e.g., involving at least some minicircles in *Karlodinium micrum*, Espelund et al., *PLoS One* 2012; Richardson et al., *MBE* 2014). Fragmentation of the *Karlodinium* plastid genome makes complete plastome assembly in Kareniaceae complex (Espelund et al. 2012, Richardson et al. 2014). This strongly invites a separate project focused on kareniacean plastid genomes but is vastly out of scope of this study.

As described above, we have obtained striking new results which we are happy to report in the revised manuscript and which suggest even more, so far unnoticed, plastid replacements in the kareniacean lineage. In light of these findings, parts of the Results and Discussion sections have been extensively rewritten, and the schematic models presented in Fig. 8 has been updated to account for the distinct evolutionary origins of the *Karlodinium armiger* and *Takayama helix* plastids.

In addition, although I might be wrong, the phylogenomic analysis for plastid-encoded transcripts might be performed with their nucleotide sequences according to the figure title and legend of Figure S4 mentioning "nucleotide phylogenetic matrix" and the file name "plastid_coded_nt_concatenation_files.tar". If so, translated amino acid sequences should be subjected to phylogenetic analysis, to avoid a well-known artifact that is caused by saturation of substitutions at the 3rd codon.

Author response: With the exception of our 16S rDNA trees (in supporting data), all of our trees were generated with conceptual amino acid translations using a standard codon translation table, in accordance with previous studies (e.g., Klinger et al. *GBE* 2018). We have revised the file and figure names accordingly.

4. Duplication of an ATP synthase subunit

Duplication and relocation of ATP synthase subunit delta seems interesting. In figure S6.4.1, could you clarify why the possible extensions containing signal peptides lack the initiation methionine at N-termini? I wonder they are 5' UTRs but artifactually detected as signal peptides, if they all indeed lack Met. To evaluate this point, I recommend 5' RACE followed by transformation into a model organism as performed in previous studies by some of the authors.

Author response: We reinvestigated these sequences more thoroughly using raw nucleotide data and conclude that the evidence for their retargeting to plastids is very weak and the reported extensions more likely represent untranslated regions some of which were falsely predicted as signal peptides. This section was removed from the new version of the manuscript, although we have noted in Supplementary Text 6.4 that: "A targeted HMMER search for possible distant homologs revealed that the distantly related functional analog of this protein in mitochondrial F-type ATP synthase (ATP5D, K02134) is duplicated in all species except *Takayama*. The additional copies, however, do not possess a detectable plastid-targeting signal and the specific functions of this duplicated subunit remain to be determined" (lines 107-111).

5. Comparison of transit peptides

Amino acid compositions in transit peptides would vary when targeted compartments are different. In

complex plastids, there are functionally distinct compartments: lumen, stroma, periplastidal compartment (PPC). Comparison should therefore be conducted separately for lumen-targeted, stroma-targeted and PPC-targeted proteins in order to claim their transit peptides are not conserved.

Author response: We acknowledge that this question was not explored in our analysis. We therefore re-analyzed our datasets taking the inferred sub-plastidial (thylakoid vs other, based on function) localization of the proteins into account. Our results showed no notable differences between these subsets and are reported in supplementary figure S10.

6. RDS never possessed a stable fucoxanthin plastid

Although the authors cite Hehenberger et al. 2019 for that RDS never possessed a stable fucoxanthin plastid, as far as I know, that paper seems not to mention it. Could you let me know where that is mentioned in the paper? Hehenberger et al. instead proposed the retention of non-photosynthetic peridinin plastid.

Author response: We have modified the Results text, noting that we only identify 42 plastid-late proteins shared between RSD and other Kareniaceae, and in the Discussion that these data provide only limited support for a shared fucoxanthin plastid. We further clarify in the Introduction that “In some cases, the co-existence of a new organelle or endosymbiont with a remnant of the ancestral plastid has been proposed” (lines 106-108) and “It has previously been suggested that the RSD retains a non-photosynthetic form of peridinin plastid” (lines 378-379) with regard to the Hehenberger paper.

Regardless of whether Hehenberger et al. mentioned or not, Novák Vanclová et al. propose that RDS never possessed a stable fucoxanthin plastid because, if I understand correctly, they detected no or few haptophyte-derived RDS genes for plastid-targeted proteins of which origins are shared with those of *Karlodinium*, *Karenia*, and *Takayama*. What about the possibility that the last common ancestor of kareniacean dinoflagellates possessed a fucoxanthin plastid in addition to peridinin plastid followed by almost complete losses of those haptophyte-derived genes after loss of a fucoxanthin plastid in evolution leading to RSD? Free living eukaryotes were appeared to have lost a plastid in recent studies and they have only a few or no genes showing evidence of a plastid previously retained. We cannot rule out that an ancestor of kareniacean dinoflagellates possessed both of peridinin and fucoxanthin plastids, as the authors mention in the main text, and either plastid was inherited to each lineage by differential losses. Accordingly, I would say Fig. 7 is a too much strong proposal as alternative hypotheses are still present. They should be introduced equally.

Author response: We thank the reviewer for this comment. As discussed above, we evaluate the possibility of a cryptic peridinin plastid shared in different kareniaceae, which is suggested at a genetic level by our data but awaits structural confirmation.

We agree that alternative hypotheses may be invoked for the origins of the current kareniacean plastids, and have modified our Fig. 8 to present three alternative possibilities: serial transfer, independent acquisition, and coexistence of an ancestral peridinin and fucoxanthin plastid, as the reviewer suggests. The presence of an ancestral fucoxanthin plastid that was subsequently replaced in *Takayama* and *Karlodinium armiger* is strongly suggested by the monophyly of the plastid-late signal across all kareniacean species studied, except RSD. We nonetheless feel that the frequent monophyletic placement of the *Karenia* and *Karlodinium micrum* plastids to the exclusion of

Takayama in our plastid-late gene trees strongly argues against a vertical inheritance of this plastid from the common kareniacean ancestor, and more likely reflects a serial transfer between the *Karenia* and *Karlodinium* / *Takayama* branches. We have evaluated the evidence for and against each hypothesis in the Discussion and in the Fig. 8 legend.

7. rRNA copy numbers in dinoflagellates

It is known that the rRNA gene copy number varies among populations or strains in dinoflagellates; some possess several dozens of times as many rRNA gene copies as others (Galluzzi et al. 2010). Is it informative to see the ocean wide rRNA gene amplicon data for the kareniacean dinoflagellates? The numbers of rRNA gene-derived reads would not necessarily reflect the cell abundance of dinoflagellates.

Galluzzi et al. 2010. Analysis of rRNA gene content in the Mediterranean dinoflagellate *Alexandrium catenella* and *Alexandrium taylori*: implications for the quantitative real-time PCR-based monitoring methods. *J Appl Phycol* 22:1-9

Author response: We thank the reviewer for raising this point. The exploration of Kareniaceae distribution was intended primarily to investigate their respective ecological relevance in terms of niche diversity, in particular compared with the well-known cosmopolitan patterns of haptophytes, rather than comparing their abundance patterns. We feel that our approach, treating each Kareniacean genus independently, is sufficient for this, but have now clarified in the Results that the different abundances observed “may be biased by the different ribosomal DNA copy numbers in different genera” (lines 330-331) and have cited the reference the reviewer has kindly supplied.

We further note in the Discussion that “It will therefore be worthwhile in the future to assess the distributions of other more recently developed marker genes (Penot et al., 2022; Pierella Karlusich et al., 2023)” (lines 371-372).

Minor points

1. the dataset size for the 241 protein-based host phylogeny should also be described in the main text.

Author response: The information (72,162 positions, 241 genes, removal of sequences with >66% gaps) has been included in the Materials and Methods.

2. The authors mention in Discussion “Thus, our results illuminate the mechanistics of a fundamental process that may under pin vast tracts of chloroplast evolution”.

If I understand correctly, I think this is based on “shopping bag model” when considering plastid replacements in dinoflagellates. It is helpful to add more details to clarify why the authors would like to claim so. “Chloroplast” should be replaced with “plastid”.

Author response: We agree that the term plastid is more appropriate in this context, and have used it globally throughout the manuscript. We have mentioned once in the Introduction “primary plastids, i.e. chloroplasts” to orient the non-specialist reader.

We have elaborated on our definition of the Shopping Bag model, and the specific importance of the Kareniaceae, in the Discussion: “The idea that individual genes encoding plastid-targeted proteins may exhibit evolutionary affiliation with other groups than the plastid donor, typifying the “shopping bag” model (Larkum et al., 2007), is well-established in many plastid lineages” (lines 350-352).

Nonetheless, we feel that our data are in many ways different to those previously observed in other plastid lineages. This may reflect that the karenian plastid has undergone one, and potentially multiple, recent replacement events. Nonetheless, the predominant contribution of the host to the plastid proteome is striking, which we elaborate in the Discussion: “Our data show that the dinoflagellate host was the principal contributor of nucleus-encoded proteins supporting the karenian plastid proteome” (lines 352-353).

3. Supplementary document S6.6

I found the term nitrogen fixation, but should this be replaced with "nitrogen assimilation"?

Author response: We have corrected the text as requested.

4. Figure S5

For those LGTs, all the trees should be shown in supplementary text as they are only 11 or 12 trees. Especially, please add the chlorophyllide b reductase and chlorophyllase in the figure.

Author response: Trees for all laterally transferred genes mentioned in the text have been provided among supplementary figures (S7.1-10).

5. References

I am not picky about a format of the reference list, but I think it should be consistent throughout the list. I recommend adding journals, volumes, and pages precisely for cited papers. I found lack of them at least in Novak Vanclova et al. and Pierella Karlusich et al.

Author response: We corrected the incomplete citations and will perform a complete reformatting of the references to comply with the requirements of a concrete affiliate journal.

6. Figures

In figure 3, I strongly recommend adding RDS data, while distinguishing them by another color if they are derived from different origins from those of *Karenia*, *Karlodinium*, and *Takayama*. This would make the authors claim clearer that there are few haptophyte-derived genes for plastid targeted proteins of which origins are shared with those of the other karenian dinoflagellates.

Author response: We believe the comparison to RSD is not among the main stories of our study and adding this dimension to the already complex discussion and metabolic map schematic would compromise the overall clarity. This point is already noted by Reviewer 1 (above). However, this question may indeed be asked by some readers, therefore we decided to include the results for RSD as an additional column in the supplementary table S3 and as an additional graphical element in the supplementary version of the map schematic (figure S8). Per the reviewer's comments above, we have further stated the number of plastid-late trees shared (42) between the RSD and other karenian in the Results text.

In figures S5.1-2 showing LGTs, I found two paralogs of karenian dinoflagellates. What does "CP" mean? If "CP" means ChloroPlast-targeted, both paralogs of *K. brevis* in HARS and those of *K. micrum* are of plastid-targeted in TARS and they do not have cytosolic ones. I am afraid if these cases are caused by false positives of detection for plastid-targeted proteins by PredSI and ChloroP. Similarly, in figure S5.4, I found two distant paralogs of heme oxygenase in the tree and the taxon names for both types in karenian include "CP." Are both targeted to the plastids or of false positives?

Author response: The annotation with “CP” and darker colour denotes proteins that were predicted as plastid-targeted by our pipeline. We have clarified in supporting text 6.8 that we investigated our aminoacyl-tRNA synthetases for possible dual targeting to both plastid and mitochondria but found no evidence for it.

We have searched the *K. brevis* SP3 HARS sequence (CAMPEP-0189291366) by CD-search and note that the conserved domain (underlined) starts at residue 24 after the first predicted methionine (**bold**), which is inconsistent with the probable length a plastid-targeting sequence, and we have noted in the figure legend that this is likely to represent a false positive.

> CAMPEP_0189291366_Karenia-brevis-SP3-20130916

```
SWLVLLAFALTPGPVVAVSATILRGLLVGLQRPCAAALRLSCCAATRALPLPGASELGSRFAAAAASSARMGKEGKKKEDGK
KKKDETKTEKLIGLEPPSGTRDFFPAEMRQORYIFNKFRETANLYGFQEYDAPVLEHQELYIRKQGEEITDQMYSFDDKEGAKV
TLRPEMPTLARMLVNLNRVETGEMAAQLPLKWFSSIPQCWFETTQRGRKREHYQWNMDIVGVTSIYAEAEALLSAICNFFESV
GITSKDVGLRVNSRKVLNAVTKLAGVPDDRFAETCVIIDKLDKIGAEAVKTEMREKIGLPEEVGERIVKATGAKSLEEFADLAG
VGQNNPEVLELKHLELAEDYGYGDWLFIDASVVRGLGYTGVVFEFDRAGVLRAICGGGRYDRLLTKFGSPKEIPCVMGFGF
GDCVIAELLKEKGVTPSLPEHIDFVVAAFNSEMMGKAMNAARRLRLGGKSVDFTEPGKKVGKAFNYADRVGADMVAFIAPD
EWAKGLVRIKALRMGQDVPDDQKQKDVPLEDLANVDSYFGLAPAAAAPVMSAAPAASTVKSTAPALAVPAAAKASAPKAAAP
SGTGADVEAFLVDHPYVGGFRPCARDRTLFDLRLTSGRPSTPALGRWYDHIDSFFPAVVRASWC
```

The green HARS sequences (including that of *Karenia brevis* SP1) in contrast typically have conserved domains starting after residues 50-60, and are likely to be genuinely plastid-targeted. Reflecting that the automated prediction approach used within our dataset may contain other such false positive results (c.f., Fig. S18), we have chosen for tree-sorting and pathway reconstruction analyses to only consider genes in which we can identify plastid-targeted homologues of the same inferred phylogenetic origin in at least two distinct Kareniacean genera (Figs. 2, 3).

For the *Karlotinium micrum* TARS sequence we have identified a second TARS sequence (CAMPEP_0200847158) that is of apparent dinoflagellate origin and lacks a credible targeting sequence, and have updated the tree accordingly.

In the case of heme oxygenases, we are convinced that (at least) two paralogs of distinct origins are indeed plastid targeted. The presence of multiple copies of this enzyme has been noticed in other organisms including some plants (e.g., Dammeyer and Frankenberg-Dinkel, Photochemical & Photobiological Sciences, 2008) and may be reflective of functional specialization or regulation / expression under different conditions. We have discussed this in the supporting text 6.1: “Two evolutionarily distinct versions of the biliverdin-producing haem oxygenase seem to be present ...the specific metabolic functions of the green- and haptophyte-like haem oxygenases in the fucoxanthin plastid await experimental characterisation.” (lines 52-58).

Reviewer #3 (Significance (Required)):

Significance

General assessment: provide a summary of the strengths and limitations of the study. What are the strongest and most important aspects? What aspects of the study should be improved or could be developed?

This study by Novak Vanclova et al. provide new transcriptome datasets from multiple species in kareniacean dinoflagellates including harmful and toxic species. Their transcriptome datasets would help understand their biology, evolution, and ecology. The authors also provide a program that predicts plastid

Full Revision

proteomes in those dinoflagellates, which would be useful for future studies to focus on kareniacean dinoflagellate plastids, after further refinement. The most important aspect of this study is that many plastid-targeted proteins might be derived from a particular haptophyte lineage, although it is still not sure whether they are derived from LGTs or EGTs. Phylogenetic analyses performed in this study should be improved by adding some plastid genomes, in order to gain more conclusive results. In addition to methods, interpretation of the current results and proposals on plastid evolution should be toned-down.

Advance: compare the study to the closest related results in the literature or highlight results reported for the first time to your knowledge; does the study extend the knowledge in the field and in which way? Describe the nature of the advance and the resulting insights (for example: conceptual, technical, clinical, mechanistic, functional,...).

Although there are technical issues, this study improves our conceptual understanding the plastid proteome evolution in Kareniacean dinoflagellates. The plastid proteomes are comprised of proteins with more various origins in those dinoflagellates, suggesting more complex plastid proteome evolution than previously thought.

Audience: describe the type of audience ("specialized", "broad", "basic research", "translational/clinical", etc...) that will be interested or influenced by this research; how will this research be used by others; will it be of interest beyond the specific field?

This study seems to be "basic research".

Please define your field of expertise with a few keywords to help the authors contextualize your point of view. Indicate if there are any parts of the paper that you do not have sufficient expertise to evaluate.

algal evolution, eukaryotic evolution, mitochondrial metabolisms, plastid metabolisms, phylogenomics

Decision Letter from The EMBO Journal

Date: 20th Nov 23 05:47:08

Last Sent: 20th Nov 23 05:47:08

Triggered By: Yehu Moran

From: y.moran@embojournal.org

To: vanclova@gmail.com

CC: yehu.moran@mail.huji.ac.il

BCC: office@reviewcommons.org

Subject: Decision on Manuscript EMBOJ-2023-116118 | [RC-2022-01782] [D-REF]

Message: 20th Nov 2023

RE: Manuscript EMBOJ-2023-116118, New plastids, old proteins: repeated endosymbiotic acquisitions in fucoxanthin-containing dinoflagellates

Dear Dr. Novák Vanclová and Dr. Dorrell,

Thank you again for submitting your work to The EMBO Journal. After discussing your manuscript and the report of the three referees from Review Commons with the editors of both EMBO Journal and EMBO Reports we reached the conclusion that while your manuscript describes very interesting findings its breadth is insufficient for meeting the requirements of EMBO Journal. However, it does fit EMBO Reports quite well and we would be happy to accept it in principle to EMBO Reports with some minor editing, mandatory formatting and styling requirements. I hope you will choose to accept our offer.

Best regards,
Yehu Moran
Editor

As a service to authors, EMBO provides authors with the possibility to transfer a manuscript that one journal cannot offer to publish to another EMBO publication. The full manuscript and if applicable, reviewers reports are automatically sent to the receiving journal to allow for fast handling and a prompt decision on your manuscript. For more details of this service, and to transfer your manuscript to another EMBO title please click on *Link Unavailable*

Yours sincerely,

Yehu Moran
Editor
The EMBO Journal

=====

Rev_Com_number: RC-2022-01782
New_manu_number: EMBOJ-2023-116118
Corr_author: Novák Vanclová
Title: New plastids, old proteins: repeated endosymbiotic acquisitions in fucoxanthin-containing dinoflagellates

Dear Dr. Novák Vanclová

Thank you for the submission of your manuscript to EMBO Reports. Your manuscript, EMBOR-2023-58545-T, still has many minor edits that I would like you to incorporate before we can proceed with the official acceptance of your manuscript.

2) individual production quality figure files as .eps, .tif, .jpg (one file per figure).

Please download our Figure Preparation Guidelines (figure preparation pdf) from our Author Guidelines pages <https://www.embopress.org/page/journal/14693178/authorguide> for more info on how to prepare your figures.

4) a complete author checklist, which you can download from our author guidelines (). Please insert information in the checklist that is also reflected in the manuscript. The completed author checklist will also be part of the RPF.

5) Please note that all corresponding authors are required to supply an ORCID ID for their name upon submission of a revised manuscript (). Please find instructions on how to link your ORCID ID to your account in our manuscript tracking system in our Author guidelines

()

6) We replaced Supplementary Information with Expanded View (EV) Figures and Tables that are collapsible/expandable online. A maximum of 5 EV Figures can be typeset. EV Figures should be cited as 'Figure EV1, Figure EV2' etc... in the text and their respective legends should be included in the main text after the legends of regular figures.

7) Before submitting your revision, primary datasets (and computer code, where appropriate) produced in this study need to be deposited in an appropriate public database (see < <https://www.embopress.org/page/journal/14693178/authorguide#dataavailability>>).

- Dataset #1

- Dataset #2>

The accession numbers and database should be listed in a formal "Data Availability " section (placed after Materials & Method) that follows the model below (see also < <https://www.embopress.org/page/journal/14693178/authorguide#dataavailability>>).

Please note that the Data Availability Section is restricted to new primary data that are part of this study.

Data availability

- RNA-Seq data: Gene Expression Omnibus GSE46843 (<https://www.ncbi.nlm.nih.gov/geo/query/acc.cgi?acc=GSE46843>)

- [data type]: [name of the resource] [accession number/identifier/doi] ([URL or identifiers.org/DATABASE:ACCESSION])

7) Please note that a Data Availability section at the end of Materials and Methods is now mandatory. In case you have no data that requires deposition in a public database, please state so instead of refereeing to the database.

See also < <https://www.embopress.org/page/journal/14693178/authorguide#dataavailability>>. Please note that the Data

Availability Section is restricted to new primary data that are part of this study.

Additional information on source data and instruction on how to label the files are available .

10) Figure legends and data quantification:

- the name of the statistical test used to generate error bars and P values,
 - the number (n) of independent experiments (please specify technical or biological replicates) underlying each data point,
 - the nature of the bars and error bars (s.d., s.e.m.)
- If the data are obtained from n {less than or equal to} 5, show the individual data points in addition to the SD or SEM.
- If the data are obtained from n {less than or equal to} 2, use scatter blots showing the individual data points.

11) Our journal encourages inclusion of *data citations in the reference list* to directly cite datasets that were re-used and obtained from public databases. Data citations in the article text are distinct from normal bibliographical citations and should directly link to the database records from which the data can be accessed. In the main text, data citations are formatted as follows: "Data ref: Smith et al, 2001" or "Data ref: NCBI Sequence Read Archive PRJNA342805, 2017". In the Reference list, data citations must be labeled with "[DATASET]". A data reference must provide the database name, accession number/identifiers and a resolvable link to the landing page from which the data can be accessed at the end of the reference. Further instructions are available at .

12) As part of the EMBO publication's Transparent Editorial Process, EMBO Reports publishes online a Review Process File to accompany accepted manuscripts. This File will be published in conjunction with your paper and will include the referee reports, your point-by-point response and all pertinent correspondence relating to the manuscript.

More points that should be addressed in the re-formatting:

- All figures need to be in portrait orientation. Fig 1, Fig 3, Fig. S8 are in landscape and need to be changed
 - Supplementary information needs to be changed to an Appendix. This is a pdf that includes all figures and their legends. The nomenclature is Appendix Figure S#. It needs a table of content with page numbers.
 - The table legends must be removed from the document.
 - The Supplementary tables S1 - S20 need to be uploaded as individual Datasets. The nomenclature is Dataset EV#. The legend is provided in a separate sheet.
- Given that there are so many datasets, I suggest to combine related datasets into one .xls file as separate sheets. e.g., Table S1 and Appendix should go into one Dataset file with two sheets. Also any other related datasets can be combined into one Dataset file with several sheets, e.g., the Read abundances of barcodes Table S5 - Table S8.
- The preliminary benchmarking seems not to be a dataset and should be added where appropriate in the manuscript.
 - The phylogenetic datasets can be zipped together with the legend README.txt file
 - Scripts need to be called Computer Code EV# and are uploaded as Datasets.
 - All results and discussions need to be part of the main manuscript and not in the Appendix. If you consider this information as highly specialized, please contact us to discuss.
 - All funding information listed in the Acknowledgements must be entered in our online submission system as well.
 - References: et al is used after the 10th author name. DOIs must be removed.

Kind regards,

Yehu Moran
Academic Editor
EMBO Reports

Rev_Com_number: N/a

New_manu_number: EMBOR-2023-58545-T

Corr_author: Novák Vanclová

Title: New plastids, old proteins: repeated endosymbiotic acquisitions in fucoxanthin-containing dinoflagellates

Corresponding authors: Anna, Novák Vanclová; Richard, Dorrell
Journal Submitted to: EMBO Reports
Manuscript number: EMBOR-2023-58545-T

Dear editor,

we are thankful for the effort you dedicated to evaluating our manuscript and for your assistance in its finalization. We have followed the EMBO Reports author guidelines and the direct requests stated in our email communication and would like to present our finalized submission.

Following is a brief summary of the changes we made since our initial submission to EMBO Reports via the Review Commons platform.

- We have filled in the Author checklist (provided as a separate xlsx file).
- We have renamed and reorganized the supporting datasets in as per the EMBO press author guidelines and your suggestions, most notably in case of the supplementary Tables (Datasets EV1-8), sequence and other bioinformatic datasets (Datasets EV9-10), and computer code (Computer code EV1). The legends are provided either on a dedicated sheet (in case of the tables) or in standalone README files added to the compressed archives.
- We have selected five figures previously included in the “Supplementary figures and text” to be made part of the Extended view (Figures EV1-5), with their figure legends included at the end of the main manuscript file.
- We have edited the citation style in the main text, added dataset citations, and Data availability statement.
- We have added the subchapter of results regarding detailed reconstruction of the plastid metabolic pathways, previously included in the “Supplementary figures and text”, to the main text in slightly more concise form, along with several additional references. This major change in the main text has been highlighted in blue.
- We have shortened the abstract to fit the 175-word limit.
- We have reworked the remainder of the “Supplementary figures and text” into “Appendix”.
- We provide Figures and Extended view figures in appropriate file formats.

The response letter including the reviewers’ comments and our replies and detailed description of the changes we made in response, previously uploaded through Review Commons, has undergone minor changes warranted by the renaming and reordering of supporting materials and addition of the abovementioned subchapter, and is attached further down.

Sincerely,

Anna M. G. Novák Vanclová & Richard G. Dorrell

Manuscript number: RC-2022-01782

Corresponding author(s): Anna, Novák Vanclová; Richard, Dorrell

1. General Statements

Dear editor,

We appreciate the time and effort that you and the reviewers dedicated to evaluating our manuscript and are grateful for their feedback. We have prepared a new version of the manuscript, integrating additional analyses that have yielded important new results and dramatically increased the significance of our study. The Results and Discussion sections of the manuscript have been heavily rewritten, incorporating multiple new figures, tables, and files.

Much of the novel content within our manuscript relates to an insightful suggestion from Reviewer 3 to consider the evolutionary origins of the genomes, as well as nucleus-encoded proteomes of different kareniacean plastids. By performing a new phylogenetic analysis (Fig. 5) incorporating sequenced plastid marker genes from each of our studied species, we show with robust support that the plastids of *Takayama helix* and *Karlodinium armiger* come from different haptophyte lineages, respectively in the orders Phaeocystales and Prymnesiales, to the deeply-positioned haptophyte-derived plastids in *Karenia* and *Karlodinium micrum*. Taking into account the previously-characterised kleptoplasts of the kareniacean Ross Sea Dinoflagellate (Hehenberger et al., *PNAS* 2019) these data indicate the independent acquisition of haptophyte plastids at least four times within the kareniaceae, sitting alongside established cases of repeated and independent plastid acquisitions in other dinoflagellate orders (Yamada et al., *MBE* 2017; Sarai et al., *PNAS* 2019). We propose multiple replacements and serial transfers of the fucoxanthin plastid lineage across these kareniaceae, summarised in a revised Fig. 8.

In our initial paper submission, we demonstrated that much of the nucleus-encoded and plastid-targeted kareniacean proteome is shared across species despite important lineage- and pathway-specific variation (Figs. 1 and 3). The discordant but complementary origins of the fucoxanthin plastid underline that Kareniaceae are highly chimeric organisms whose nuclear and plastid genomes, and plastid proteomes undertook different evolutionary trajectories even at micro-evolutionary scales. We further explore the consequence of this variability at cellular and ecosystem levels (Figs. 6-7). We believe this revised manuscript presents a landmark addition to the complex history of the ever-changing dinoflagellate plastids, and provides us with unprecedented insights into the different genetic consequences of post-endosymbiotic organelle evolution, and will benefit from publication in a widely-read and interdisciplinary journal within the *Review Commons* portfolio, such as *PLoS Biology*, *EMBO J*, *Mol Syst Biol*, or *eLife*.

We thank you in advance for your appraisal and look forward to hearing from you shortly.

Anna M.G. Novák Vanclová, and Richard G. Dorrell

Please see below for the detailed description of the changes made in response to the reviewers' comments (in blue, for better legibility).

Reviewer #1 (Evidence, reproducibility and clarity (Required)):

The manuscript investigated the composition of the plastid proteomes of seven distantly-related kareniacean dinoflagellates, including newly-sequenced members of three genera (Karenia, Karlodinium, and Takayama). Using a custom plastid-targeting predictor, automatic single-gene tree building and phylogenetic sorting of plastid-targeted proteins for plastid proteome construction, the authors suggest that the haptophyte order Chrysochromulinales is the closest living relative of the fucoxanthin plastid donor. Interestingly, the N-terminal targeting sequences of kareniacean plastid signal peptides, reveal a high sequence conservation. Moreover, ecological and mechanistic factors are suggested that may have driven the endosymbiotic acquisition of the fucoxanthin plastid. Overall, this is a comprehensive and interesting analysis.

Other comments.

1. For analyses of N-terminal targeting sequences, why did the authors not consider to employ Predalgo as an additional tool?

Author response: We thank the reviewer for their suggestion. To our understanding, PredAlgo is a targeting predictor trained on primary green algae, which have two-membrane bound plastids and purely hydrophilic N-terminal plastid targeting sequences. It thus would be expected to perform poorly for the prediction of N-terminal targeting sequences in complex plastids such as those of the Kareniaceae bound by three or more membranes, who are located within endomembrane-derived compartments and which utilise plastid-targeting sequences based on an N-terminal hydrophobic signal peptide for ER import.

We considered the application of PredAlgo for the identification of downstream hydrophilic transit peptide regions in Kareniacean presequences, but note that the specific residue positioned after the signal peptidase cleavage site is typically a much better predictor than transit peptide hydrophobicity for identifying plastid-targeting sequences (Gruber et al., *Plant J* 2015, and citing references). We found that other targeting prediction tools based primarily on hydrophobicity (e.g., HECTAR) performed poorly in identifying probable plastid-targeting sequences in our control Kareniacean dataset, and therefore chose to prioritise a modified version of ASAFind that takes into account the residue context of Kareniacean signal peptidase cleavage site for our targeting predictor, which works with high sensitivity and specificity on our control dataset. We summarise these observations in Fig. S13.

2. Given the fact that peridinin or fucoxanthin pigment binding is in the focus of the paper, a more detailed introduction of the peridinin and fucoxanthin light-harvesting systems should be given.

Author response: A brief introduction to the pigment-binding proteins in dinoflagellates was added, "These include a unique carotenoid pigment... massively paralogized and synthesized as polyproteins" (lines 71-74).

3. The authors state "It is also possible that there has been a direct niche competition between the peridinin and fucoxanthin plastid that may have coexisted in the same host for a period of time with possibly different selective pressure on retention of their respective proteins based on their interaction with plastid-encoded components, e.g., extrinsic photosystem subunits not assembling correctly with their intrinsic haptophyte-like counterparts." It is tempting to ask, whether peridinin light-harvesting systems have left traces in the fucoxanthin plastid, possibly due to mistargeting of peridinin light-harvesting systems into the fucoxanthin plastid? Are some photosynthetic subunits "in-between" peridinin and fucoxanthin plastids?

Author response: We did not identify any other peridinin-like photosystem subunits than the ones visualized in the map schematic (i.e., ferredoxin/PetF in both *Karenia* and *Karlodinium* and Psad of *Karlodinium micrum*) and discussed in the supplementary text. PetF is the only consistently retained peridinin-like photosystem protein, likely due to the fact that it is not strictly linked to photosynthesis: it is expressed in plant leucoplasts, and plastid-encoded in some non-photosynthetic chrysophytes. We have added a sentence "we detect no possible homologues of peridinin-chlorophyll binding proteins (PCP) in any kareniacean transcriptome" (line 284-285).

4. Figure 3 is difficult to understand, e.g. for PSI and PSII which subunits are shown, why has PSI "more" contribution from dinoflagellates as compared to PSII?

Author response: The photosystem subunits are ordered numerically in the schematic, and detailed information on each protein and the corresponding sequences with their origin are included in Dataset EV3. A single subunit of photosystem I (PsaD) was determined to be of plastid-early (peridinin-like) origin in *Karlodinium* (while the same protein is plastid-encoded in *Karenia* and undetermined in *Takayama*). We believe this may be simply due to an evolutionarily neutral differential loss / non-adaptive retention of photosynthesis-related proteins in a secondarily non-photosynthetic host before the acquisition of a replacement plastid. We note that there are only two (incomplete) kareniacean plastid genomes available so we cannot rule out the possibility of this subunit being plastid-encoded in *Karlodinium* as well (which would mean that both plastid-late and plastid-early homologs co-occur in this genus).

Fig. 3 is necessarily complex due to the size and multiplicity of the dataset considered. To facilitate reader navigation, we have added the following text to the figure legend (lines 1175-1187) text "Plastid proteins are arranged by major metabolic pathway or biological process, with each protein shown as rosettes ... Proteins of plastid-late (haptophyte) origin, such as are concentrated in photosystem and ribosomal processes, are coloured red; and proteins of plastid-early (dinoflagellate) origin, such as are concentrated in carbon and amino acid metabolism are coloured blue. ... In certain cases (shown as rosettes with multiple colours), homologues from different species have different evolutionary origins, e.g. *Karenia* possessing plastid-late and *Karlodinium/Takayama* plastid-early".

5. Data shown in figure 4, is there experimental evidence for signal peptide cleavage site(s). Could these data be used to predict mature plastid targeted protein sequence?

Author response: We were able to determine the conserved motives in signal peptide, including its cleavage site (GRR) which we exploited in the design of kareniaceae-specific matrix for ASAFind. We show these residues in Fig. 4. We note that these motifs were identified based on homology to known signal processing peptidase recognition sites, as opposed to experimentally determined protein N-termini.

Consistent with previous studies (e.g. Yokoyama et al., *J Phycol* 2011) we see limited evidence for consensus plastid transit peptide cleavage motifs in kareniacean presequences, and do not discuss this further as a result.

6. The authors state "Partial Least Square (PLS) analysis shows a set of environmental variables (salinity, silicate, iron) positively correlated with abundances of both *Karenia* and *Takayma* and also haptophytes as a whole, but at the same time negatively correlated to *Karlodinium* (Figure S8), further illustrating that the latter genus is quite distant from the rest in its biogeographical pattern." How could this be interpreted in the light of the plastid proteomes

Author response: We believe that this may be due to the more cosmopolitan distribution of *Karlodinium*, and possibly also a result of bias stemming from our strategy of grouping the organisms at the genus level (as not enough data was available at species level) which may obscure the potential outlier status of only some species/ subpopulations. This is particularly true for the haptophytes, where in the absence of specific ancestry for individual kareniacean plastids we are only able to consider distributions at the levels of entire orders. We now acknowledge this in the Discussion: "specific ecological interactions between the progenitors ... via ancestral niche reconstruction for each lineage" (lines 528-530).

Please note, that the results might have changed slightly from the previous version due to the recalculation following additional normalization of the data (see below).

Reviewer #1 (Significance (Required)):

The current manuscript gives insights into the endosymbiotic acquisition of the fucoxanthin plastids.

Reviewer #2 (Evidence, reproducibility and clarity (Required)):

This is a well done, detailed bioinformatic analysis of genomic and transcriptomic data from an important lineage of dinoflagellates that have undergone serial substitution of their plastid. On the whole I am enthusiastic about the paper; it presents valuable new insights, and is rigorously performed. However, I have to object to the way the term "proteome" is used in the paper; the manuscript is talking about the predicted proteome, not a measured proteome. This is something of a technical distinction, but it is an important one because the transcriptome and the proteome don't necessarily track each other, and there is little or no actual proteomic data available from dinoflagellates. We assume that transcript abundance has something to do with proteome abundance, but this is often violated. What this paper is really addressing is the potential proteome, because if a given gene is completely absent from the genome and the transcriptome we can be confident it will not be present in the proteome. The converse is not true. For this reason I feel it is important to be clear on the distinction. I would be satisfied in this regard by minor modifications, using the term "predicted proteome" in the title, and being more direct in the introduction about the distinction.

Author response: We agree that the usage of the word proteome for *in silico* predictions is not entirely correct, and have used the term “predicted proteome” where possible in the text to clarify this.

We have also, as described in our response to Reviewer 1 above, included a statement in the Discussion that our largely bioinformatic results will be transformed by an experimentally realised kareniacean plastid proteome, which we nonetheless feel goes beyond the scope of our manuscript.

Overall the analyses are impressive. I do have to squirm a little when I see automated analyses generating alignments where the threshold is less than 75% gaps and at least 100 nucleotides aligned. I looked at the supplementary data and the figshare files and could not find the alignments themselves, so I don't know what fraction of the sequences are in that territory. Because phylogenetic analysis (as performed here) treats the alignments as an observation, and because the alignments include sequences with more than 50% gaps, it is entirely possible that some taxa, or even whole segments of the tree, are based on non-overlapping data.

Author response: We thank the reviewer for their comment and have added in three new supplementary figures (S16-S18) providing statistics on alignment size, length, and average gap percentage distribution. We report that most of the alignments contained relatively little gaps: 90% of the alignments contained between 1.1 and 24.5% of gaps with median value of 6.6%.

Mind you, we have done similar analyses, and I don't think this invalidates the results, but it does open up the possibility of some dramatic artifacts. Consequently, I would recommend a) making the alignments available (or more obvious where to find them), and b) providing more detail on the alignments, including, if possible, to add a figure (probably in the supplementary data) that visualizes them. It is not given in the text itself, but according to the figure 2 caption there are 22 sequences thought to be "plastid late", and 241 in the pan-eukaryotic dataset. This is a scale that is feasible to put in a figure showing, for example, each aligned residue as a color and indels as grey. Such a figure is readable even when the individual residues are only a few pixels in size (less than a millimeter when printed). I also recommend describing the final alignments more fully in the text. Most of the summary statistics are presented in normalized form, and that can obscure patterns that come from poorly sampled taxa. Better clarity on the characteristics of the alignments will make it easier to interpret the findings overall. Although this is critical to interpreting the results, gappy alignments are not uncommon in analyses of this sort, and setting that aside the analyses presented are comprehensive and thorough. The discussion does a good job of addressing the significance of the work, and potential causes of error are addressed adequately (aside from the matter of the alignments).

Author response: We thank the reviewer for their comment and have provided alignments for all single-gene trees, in a dedicated online supporting repository (https://figshare.com/articles/dataset/all-automatically-generated-alignments_rar/24347032). The datasets and alignments used for PhyloFisher and plastid-encoded gene trees are included directly in the supplementary files (Dataset EV10).

We have additionally included three new supporting figures (S14-S16) showing the distributions of lengths, gaps and homologues in each single-gene tree. These data project largely completion of individual alignments, with only 5% containing > 20% gapped positions (see Fig. S16), for example. We have additionally clarified in the Methods that “The trimmed alignments were then

filtered by a custom python script that discarded sequences comprising of more than 75% gaps and then rejected alignments shorter than 100 positions or containing fewer than 10 taxa.” (lines 627-629).

For the two concatenated trees presented, we have clarified in the Methods the alignment lengths (PhyloFisher: 72, 162 positions; plastid genes: 2,404 positions), and that we removed sequences containing >66% of gaps from the final alignment. Reflecting on the congruency assumptions required to concatenated alignments, we have chosen to replace the plastid-late concatenated tree (which may group proteins with multiple phylogenetic signals) with a new main text figure 2 providing an overview of the plastid signals we observe across the entire dataset (see comments below to Reviewer 3).

Reviewer #2 (Significance (Required)):

I find the paper to be exciting and important. These organisms are economically important, particularly as potential nuisance organisms, but also because of their role in primary productivity. They also have extremely complex evolutionary histories and similarly complex genomes. performing any bioinformatic analysis of these organisms is a substantial challenge because almost every gene exists in high copy number and with complex and often obscure patterns of homology. The manuscript brings forward these challenges, and makes a substantial step forward in elucidating the evolution of a group that is fascinating and important, but remarkably difficult to work with. I feel that it is an important analysis, and should be of interest to a broad audience.

Reviewer #3 (Evidence, reproducibility and clarity (Required)):

Summary

This manuscript entitled "Divergent and diversified proteome content across a serially acquired plastid lineage" by Novak Vanclova et al. proposes the origin and evolution of plastids in kareniacean dinoflagellates. The authors generated new transcriptome data from *Karenia mikimotoi*, *Karenia papilionacea*, *Karlodinium micrum*, *Karlodinium armiger*, and *Takayama helix*. Combining them to the previously published transcriptome data from kareniacean dinoflagellates, they constructed the pan-kareniacean transcriptome library. They surveyed plastid-targeted protein-coding transcripts in the dataset, and consequently they estimated ~14.5% of the transcriptome data were of plastid-targeted ones. Of them, 65-80% were derived from a peridinin-containing dinoflagellate ancestor while ~15% were derived from EGTs from a haptophyte endosymbiont of the current plastid origin. By using the plastid-targeted transcript dataset, they investigated 1) origins of the plastid-targeted protein-coding transcripts by single gene-trees, 2) the plastid origin and evolution by the multigene dataset of 22 conserved plastid-targeted protein-coding transcripts and of 3) plastid genome-derived transcripts, 4) plastid functions, 5) diversity of plastid-targeted signals in kareniacean dinoflagellates, and 6) the distributions of kareniacean species by using the Tara Oceans database. On the basis of their results, they proposed many hypotheses regarding kareniacean dinoflagellate evolution, such as i) the chrysochromulinales-origin of the plastids, ii) more recent acquisition of the plastid than previously thought, iii) a plastid replacement within kareniaceae evolution, iv) the strict selection of signal peptides but non-conserved transit peptides in the kareniacean plastid-targeted proteins, and v) correlated or non-correlated distribution patterns of kareniacean dinoflagellates to specific haptophyte lineages.

Although their proposals are interesting, I have many concerns to be addressed. Especially, their analyses on which the above proposals are based seem to be still preliminary and inconclusive. To support their proposals more confidently, I also suggest some additional analyses.

Major comments

1. seemingly inconsistency between the authors' claims

The most striking is inconsistency of the authors' claims proposed in this manuscript. Their proposals include a) the common ancestor of kareniaceans has not possessed a fucoxanthin plastid but the plastid has been acquired more recently, b) an ancestor of Takayama and Karlodinium has gained a fucoxanthin plastid from a (chrysochlomulinales) haptophyte, c) an ancestor of *Karenia* has gained a fucoxanthin plastid from Karlodinium.

However, they also demonstrate a higher proportion of plastid-late proteins in *Karenia* than Karlodinium and Takayama.

If I understand correctly, "a higher proportion of plastid-late proteins in *Karenia* than Karlodinium and Takayama" would seemingly be inconsistent to and challenge two of the authors' claims: no haptophyte-derived plastid in the common ancestor of kareniacean dinoflagellates and a Karlodinium-to-*Karenia* plastid transfer (Fig. 7). If the *Karenia* plastid is derived from Karlodinium, I have no idea why haptophyte-derived plastid proteome of *Karenia* is larger than that of Karlodinium. After the plastid acquisition in *Karenia*, *Karenia* might have gained more genes for plastid-targeted proteins from haptophytes by LGTs. If this is true, many single gene trees would suggest different origins of plastid-targeted proteins between *Karenia* and Karlodinium/Takayama. Can we see it in the single gene analyses? I would like authors to rationalize the inconsistency in the main text.

Author response: We agree with the reviewer that the evolutionary origins and dynamics of the kareniacean plastid proteome are complex, and thank them for their suggestion.

First, to take into account the different evolutionary scenarios that could explain the present-day distribution of the kareniacean plastids, including the new plastid genome sequences identified in response to the reviewer's suggestions, we have made a revised version of Fig. 8 evaluating three different hypotheses (see below). Nonetheless, we feel that the *Karlodinium*-to-*Karenia* model we propose is plausible, based on the following observations:

- We identify 1,418 plastid protein gene trees in which at least two of the three studied genera (*Karenia*, *Karlodinium*, *Takayama*), and 748 in which all three resolve as monophyletic, and with a haptophyte sister-group (i.e., a common plastid-late origin; Fig. S2). This points to a common haptophyte ancestry in all three groups, as opposed to independent endosymbiotic consumptions of free-living haptophytes in *Karenia* and *Karlodinium micrum*.
- We see no such shared signal with the RSD, which shares only 42 proteins with at least two other kareniacean genera (Fig. S4). Thus, and consistent with previous studies (Hehenberger et al., *PNAS* 2019) we cannot invoke an ancestral presence of a fucoxanthin plastid shared with the RSD in the last common kareniacean ancestor. This discrepancy thus likely points to a serial transfer of the kareniacean plastid from either *Karlodinium* into *Karenia* or vice versa (Fig. 8).

- Concerning the direction of this transfer, among 1,059 gene trees of plastid-late origin found in both *Takayama*, *Karenia* and *Karlodinium*, 873 place *Takayama* as basal to a monophyletic clade of *Karenia* and *Karlodinium*, i.e. support a specific plastid transfer between the latter two genera. The most parsimonious explanation for this is the origin of the fucoxanthin plastid in the common *Takayama*/*Karlodinium* ancestor, which was subsequently transferred into *Karenia*.

It is true that *Karenia* contains both a greater absolute proportion of predicted plastid-targeted proteins (Fig. 1) and greater number of unique KO number annotations (Dataset EV3) of plastid-late origin than either *Karlodinium* or *Takayama*. That said, this signal may be influenced by multiple other factors beyond how old the given endosymbiosis is (i.e., longer coexistence implies more EGT). For example, the number of plastid-late gene in a host genome may depend on the frequency of duplication of plastid-late genes and the receptiveness of the host nuclear genome to incoming horizontally derived genes. It may further be influenced by the presence and relative selective advantage or disadvantage of competing genes of host nuclear origin (i.e. plastid-early genes) that may be differentially selected over plastid-late genes, which might vary between *Karenia* and *Karlodinium* due to differential retention of the ancestral peridinin-type plastid in each lineage.

We have elaborated on this point in the Discussion, noting that there may have been “a direct niche competition between the peridinin and fucoxanthin plastid ... with possibly different selective pressure on retention of individual imported proteins” (lines 425-427), “relatively recent origin and spread throughout the kareniacean genome, e.g., via gene duplications” (line 514), and finally that precedent for divergent evolutionary trajectories in different Kareniaceae exists from the *Karenia* and *Karlodinium* plastid genomes that “contain partially non-overlapping sets of genes that suggest independent post-endosymbiotic plastid genome reduction” (lines 458-459). Nonetheless, we acknowledge that the evolutionary model we propose is not definitive, and that alternative explanations may find more favour with increased genome data.

2. Signal peptide prediction

I think the modified ASAFind would be greatly helpful for future studies on automatic prediction of plastid proteomes in kareniacean dinoflagellates. However, I found no data on selection criteria for the signal peptide prediction program SignalP5.0 used. I believe such data would be very important to interpret the previously published paper by Gruber et al. in which prediction methods for plastid-targeting sequences are compared to each other to see how sensitively and specifically they can capture the plastid proteomes.

Gruber et al. 2020. Comparison of different versions of SignalP and TargetP for diatom plastid protein predictions with ASAFind.

According to Gruber et al. (2020), signalP5.0 is not suitable for prediction of signal peptides for diatoms, in consistent with the authors' claim for kareniacean dinoflagellates. This inconsistency would be difference of the nature in signal peptides between diatoms and kareniacean dinoflagellates. Even if so, it would be useful to see quantitatively how much different their signal peptides are in terms of their suitable prediction programs.

Author response: In our preliminary benchmarking using only the previously published transcriptomes (see Dataset EV6), SignalP 5.0 performed substantially better in terms of specificity

than SignalP 3.0 (i.e., 22 versus 34/ 728 retrieved positive hits of proteins with uniquely non-plastidial functions), with comparable sensitivity in the correct prediction of positive control proteins. Given the size of our dataset, and the substantial risk of false positive detection in the highly expanded and redundant dinoflagellate transcriptomes we have used, we feel that the greater specificity of SignalP 5.0 is important to integrate in our model selection. We have clarified this position in the Methods, stating “First, the relative effectiveness of two SignalP versions ... SignalP 5.0 was used for all subsequent analysis.” (lines 581-585).

I also have a concern about use of the combination of PrediSI and ChloroP, combination which is suitable for the plastid proteome prediction in *Euglena gracilis*. The authors should rationalize why the method for *Euglena* plastids can be applicable without any modification to the plastid proteome prediction in kareniacean dinoflagellates. Although *Euglena* plastids are enclosed by three membranes, kareniacean plastids are by four. Therefore, from the side of molecular mechanisms in protein import, the method suitable for *Euglena* plastids is not necessarily suitable for kareniacean dinoflagellate plastids.

By using PrediSI and ChloroP, they detected additional "candidate plastid proteomes" including several proteins not detectable by SignalP5.0 and the modified ASAFind. That seems great. However, they did not seem to consider false positives since there is no mention on it. Although the additional candidates predicted by PrediSI and ChloroP included true plastid proteins of kareniacean dinoflagellates, many might not be. Nevertheless, the authors suggest 7.5 to 14.5% in *K. micrum* and *K. brevis*, respectively, are of plastid-targeted ones. I am so afraid if the proportions would be highly overestimated due to false positives by PrediSI and ChloroP.

To rationalize the use of PrediSI and ChloroP, the authors should show sensitivity and specificity by quantitative analyses with a benchmark dataset.

Author response: We thank the author for this comment. The reasoning behind using the parallel PrediSI+ChloroP strategy was the previously reported similarity of the plastid signal structure between euglenids and peridinin dinoflagellates (c.f., Lukes et al., *PNAS*, 2009) and the previous observation that some kareniaceae possess plastid-targeting sequences resembling those of peridinin dinoflagellates (c.f., Hehenberger et al., *PNAS*, 2019). Per the reviewers' suggestion, we present a modified sensitivity/ specificity testing PrediSI+ChloroP, alongside other alternative targeting predictors in Figure S13. While the PrediSI+ChloroP sensitivity is very low, its specificity is comparable with the modified ASAFind, and in this regard outperforms other targeting predictor tools, thus rationalising the use of both targeting prediction tools together.

3. Origin and evolution of kareniacean plastids

The authors suggest the chrysochromulinales origin of the kareniacean dinoflagellate plastids and the Karlodinium-to-Karenia plastid transfer, on the basis of phylogenetic analyses using the concatenated datasets with the 22 conserved plastid-targeted proteins and with plastid-genome derived transcripts. It is very interesting that those plastid-targeted proteins in kareniacean dinoflagellates might be phylogenetically closely related to chrysochromulinales haptophyte

I have suggestions on the analyses and interpretation

As the 22 analyzed genes are nuclear-encoded plastid targeted genes, they are a quite small portion of entire plastid proteins. I am not convinced by that evolution of the small number of genes reflects evolution of fucoxanthin plastids of which proteomes are comprised of >1000 proteins.

How many genes for haptophyte-derived plastid-targeted proteins suggest the monophyly of kareniacean dinoflagellates and chrysochromulinales haptophytes should be investigated by, for example, a coalescence-based analysis such as Astral for all the detected haptophyte-derived plastid-targeted proteins including the 22 genes. This is because the monophyly could be reconstructed only by one or few, limited number of proteins even if the concatenated dataset is analyzed.

Relevant to this, plastid-targeted proteins derived from a peridinin-containing ancestor might still have phylogenetic signals of host evolution. I am interested in whether such analyses with peridinin plastid-derived plastid-targeted proteins reconstruct Takayama and Karlodinium as monophyletic but separate Karenia from them, as suggested in the phylogenomics with non-plastid proteins.

Author response: We agree with the reviewer concerning the problematic nature of concatenations with small numbers of genes, particularly if the underlying gene trees are not phylogenetically congruent to one another, and have chosen to replace the concatenation with a more global evaluation of the different plastid protein origins across our entire dataset. Using automated sorting approaches, we have evaluated the support for our evolutionary model across hundreds of gene trees. We feel that this approach supercedes coalescence-based techniques, as it enables us to treat each gene topology as an independent event, and to consider multiplicity in the origin of the kareniacean plastid proteome. We present these data in a new Fig. 2 and S2.

As stated above, these data strongly support monophyly of all three Kareniacean genera. Concerning the potential Chrysochromulinalean plastid signal in our dataset, we have reanalysed our data and quantify a substantial number of trees (220/ 1,418 of plastid-late origin) that specifically place multiple kareniacean genera within the Chrysochromulinales. This figure is more than twice the number (91) that place the kareniaceae with the next most occurrent haptophyte group in our dataset, Isochrysidales. We nonetheless have chosen to no longer present this as a cryptic plastid endosymbiosis, in the absence of clear examples of extant kareniaceae still possessing this plastid, saying purely in the Discussion that “a common ancestor of the studied organisms either possessed a stable plastid or had a long-term symbiotic relationship (e.g., kleptoplastidic) with a haptophyte lineage related to the extant Chrysochromulinaceae” (lines 418-420).

Concerning the phylogenetic placement of each kareniacean genus, the majority of our plastid-late trees specifically recover the monophyly of *Karenia* and *Karlodinium*. Remarkably, we find that *Takayama* and *Karlodinium* only resolve together in 69/ 1,039 plastid-late gene trees in which all three genera are represented, strongly refuting a vertical origin of the haptophyte-derived components of their plastid proteome. This is not due to the Phaeocystales origin of the current *Takayama* plastid genome, which is found in only 21 of our plastid protein trees. Nonetheless, as the reviewer suggests, the opposite trend (1,505/ 2,804 gene trees grouping *Takayama* and *Karlodinium* as monophyletic) was observed amongst plastid-early gene trees, which might reflect a cryptic peridinin plastid shared between these groups. We expand on these results in the Discussion, stating “Many of the plastid-early gene trees copy the organismal topology ...this awaits structural confirmation via microscopy” (lines 440-441).

Finally, to enable reviewer comprehension of the relationships shown, we have presented some exemplar topologies of some of the trees previously displayed in the concatenation, provided in a new Fig. EV2.

For the phylogenetic analysis of plastid genome-derived transcripts, I might be wrong, but I could not find any information on dataset sizes (i.e., the numbers of sites) and evolutionary models for the analyses in the main text nor supplementary document. Although one may see the dataset sizes when looking at the original datasets in the supplementary files, such information is substantial and thus is to be described in the materials and methods section. I am afraid if this analysis was performed with a small dataset size. I would like to know total lengths of the concatenated sequences and especially that for Takayama. The phylogenetic position of Takayama, distantly related to the other kareniaceans, in this tree might be caused by a larger portion of gaps in the Takayama sequences than in the other kareniaceans.

Author response: As noted in our response to Reviewer 2, we have included three new supplementary figures (S14-S16) with statistics on alignment size, length, and average gap percentage distribution. The average and median values of these three measurements do not differ significantly when calculated separately for different organisms. We have clarified in the Methods that the concatenated alignments retained (PhyloFisher, and plastid-encoded genes) were “constructed by IQ-TREE with the LG+C60+F model for the plastid matrices and posterior mean site frequency (PMSF) model (LG+C60+F+G with a guide tree constructed with C20) for PhyloFisher matrix” (lines 686-688).

Moreover, due to lack of the plastid genome sequence of Takayama, no one could confidently identify plastid genome-derived transcripts: some of those could be derived from second, nuclear copies that might be pseudogenes. Otherwise, even if they are plastid-derived, no one can evaluate whether they are transcripts after or prior to RNA editing. I am afraid if the dataset used is comprised of a mixture of edited and non-edited sequences in kareniacean sequences. Either of sequences after or prior to RNA editing, latter of which are identical with DNA sequences, should be consistently used for the phylogenetic analysis.

In any case, the plastid genomes are necessary for this analysis, and the authors can easily obtain them by DNaseq as they have the cultures.

Author response: We thank the Reviewer for their insightful response. We agree that understanding the evolution of kareniacean plastid genomes are crucial to understanding their evolutionary history.

We have accordingly, as described above, integrated a new main text Fig. 5 building a concatenated tree of plastid marker genes (*psbA*, *psch*, *psbD*, *psaA*, *rbcL*, and 16S rDNA) historically and commonly used to assess the evolutionary origins of fucoxanthin plastids (e.g., Takishita et al., *Phycol Res* 1999; Dorrell and Howe, *PNAS* 2012). These sequences were amplified cryopreserved stocks of total RNA and specific primers, amplified by RT-PCR. We have chosen here to use RNA sequences, to account for the presence of plastid RNA editing, which has been shown to play an important role in maintaining sequence identity between kareniacean plastids and haptophyte relatives despite a high DNA mutation rate in the former (Jackson et al., *MBE* 2013; Klinger et al., *GBE* 2018), rather than DNA sequences for this analysis.

Additionally, we would like to note that while plastid genomes are generally relatively simple to sequence and assemble, this is not the case in Kareniaceae. The existing plastid genome assemblies are partially incomplete and suggest more complex and possibly unstable structures (e.g., involving at least some minicircles in *Karlodinium micrum*, Espelund et al., *PLoS One* 2012; Richardson et al., *MBE* 2014). Fragmentation of the *Karlodinium* plastid genome makes complete plastome assembly in Kareniaceae complex (Espelund et al. 2012, Richardson et al. 2014). This strongly invites a separate project focused on karenian plastid genomes but is vastly out of scope of this study.

As described above, we have obtained striking new results which we are happy to report in the revised manuscript and which suggest even more, so far unnoticed, plastid replacements in the karenian lineage. In light of these findings, parts of the Results and Discussion sections have been extensively rewritten, and the schematic models presented in Fig. 8 has been updated to account for the distinct evolutionary origins of the *Karlodinium armiger* and *Takayama helix* plastids.

In addition, although I might be wrong, the phylogenomic analysis for plastid-encoded transcripts might be performed with their nucleotide sequences according to the figure title and legend of Figure S4 mentioning "nucleotide phylogenetic matrix" and the file name "plastid_coded_nt_concatenation_files.tar". If so, translated amino acid sequences should be subjected to phylogenetic analysis, to avoid a well-known artifact that is caused by saturation of substitutions at the 3rd codon.

Author response: With the exception of our 16S rDNA trees (in supporting data), all of our trees were generated with conceptual amino acid translations using a standard codon translation table, in accordance with previous studies (e.g., Klinger et al. *GBE* 2018). We have revised the file and figure names accordingly.

4. Duplication of an ATP synthase subunit

Duplication and relocation of ATP synthase subunit delta seems interesting. In figure S6.4.1, could you clarify why the possible extensions containing signal peptides lack the initiation methionine at N-termini? I wonder they are 5' UTRs but artifactually detected as signal peptides, if they all indeed lack Met. To evaluate this point, I recommend 5' RACE followed by transformation into a model organism as performed in previous studies by some of the authors.

Author response: We reinvestigated these sequences more thoroughly using raw nucleotide data and conclude that the evidence for their retargeting to plastids is very weak and the reported extensions more likely represent untranslated regions some of which were falsely predicted as signal peptides. This section was removed from the new version of the manuscript, although we have noted in the text that: "Interestingly, an unrelated and structurally dissimilar but functionally analogous subunit of mitochondrial ATPase (ATP5D, K02134) seems to be duplicated but it is unclear whether these two phenomena could be linked." (lines 299-301).

5. Comparison of transit peptides

Amino acid compositions in transit peptides would vary when targeted compartments are different. In complex plastids, there are functionally distinct compartments: lumen, stroma, periplastidal compartment

(PPC). Comparison should therefore be conducted separately for lumen-targeted, stroma-targeted and PPC-targeted proteins in order to claim their transit peptides are not conserved.

Author response: We acknowledge that this question was not explored in our analysis. We therefore re-analyzed our datasets taking the inferred sub-plastidial (thylakoid vs other, based on function) localization of the proteins into account. Our results showed no notable differences between these subsets and are reported in supplementary figure S8.

6. RDS never possessed a stable fucoxanthin plastid

Although the authors cite Hehenberger et al. 2019 for that RDS never possessed a stable fucoxanthin plastid, as far as I know, that paper seems not to mention it. Could you let me know where that is mentioned in the paper? Hehenberger et al. instead proposed the retention of non-photosynthetic peridinin plastid.

Author response: We have modified the Results text, noting that we only identify 42 plastid-late proteins shared between RSD and other Kareniaceae, and in the Discussion that these data provide only limited support for a shared fucoxanthin plastid. We further clarify in the Introduction that “In some cases, the co-existence of a new organelle or endosymbiont with a remnant of the ancestral plastid has been proposed” (lines 92-93) and “It has previously been suggested that the RSD retains a non-photosynthetic form of peridinin plastid” (lines 433-434) with regard to the Hehenberger paper.

Regardless of whether Hehenberger et al. mentioned or not, Novák Vanclová et al. propose that RDS never possessed a stable fucoxanthin plastid because, if I understand correctly, they detected no or few haptophyte-derived RDS genes for plastid-targeted proteins of which origins are shared with those of *Karlodinium*, *Karenia*, and *Takayama*. What about the possibility that the last common ancestor of kareniacean dinoflagellates possessed a fucoxanthin plastid in addition to peridinin plastid followed by almost complete losses of those haptophyte-derived genes after loss of a fucoxanthin plastid in evolution leading to RSD? Free living eukaryotes were appeared to have lost a plastid in recent studies and they have only a few or no genes showing evidence of a plastid previously retained. We cannot rule out that an ancestor of kareniacean dinoflagellates possessed both of peridinin and fucoxanthin plastids, as the authors mention in the main text, and either plastid was inherited to each lineage by differential losses. Accordingly, I would say Fig. 7 is a too much strong proposal as alternative hypotheses are still present. They should be introduced equally.

Author response: We thank the reviewer for this comment. As discussed above, we evaluate the possibility of a cryptic peridinin plastid shared in different kareniaceae, which is suggested at a genetic level by our data but awaits structural confirmation.

We agree that alternative hypotheses may be invoked for the origins of the current kareniacean plastids, and have modified our Fig. 8 to present three alternative possibilities: serial transfer, independent acquisition, and coexistence of an ancestral peridinin and fucoxanthin plastid, as the reviewer suggests. The presence of an ancestral fucoxanthin plastid that was subsequently replaced in *Takayama* and *Karlodinium armiger* is strongly suggested by the monophyly of the plastid-late signal across all kareniacean species studied, except RSD. We nonetheless feel that the frequent monophyletic placement of the *Karenia* and *Karlodinium micrum* plastids to the exclusion of *Takayama* in our plastid-late gene trees strongly argues against a vertical inheritance of this plastid

from the common kareniacean ancestor, and more likely reflects a serial transfer between the *Karenia* and *Karlodinium* / *Takayama* branches. We have evaluated the evidence for and against each hypothesis in the Discussion and in the Fig. 8 legend.

7. rRNA copy numbers in dinoflagellates

It is known that the rRNA gene copy number varies among populations or strains in dinoflagellates; some possess several dozens of times as many rRNA gene copies as others (Galluzzi et al. 2010). Is it informative to see the ocean wide rRNA gene amplicon data for the kareniacean dinoflagellates? The numbers of rRNA gene-derived reads would not necessarily reflect the cell abundance of dinoflagellates.

Galluzzi et al. 2010. Analysis of rRNA gene content in the Mediterranean dinoflagellate *Alexandrium catenella* and *Alexandrium taylori*: implications for the quantitative real-time PCR-based monitoring methods. *J Appl Phycol* 22:1-9

Author response: We thank the reviewer for raising this point. The exploration of Kareniaceae distribution was intended primarily to investigate their respective ecological relevance in terms of niche diversity, in particular compared with the well-known cosmopolitan patterns of haptophytes, rather than comparing their abundance patterns. We feel that our approach, treating each Kareniacean genus independently, is sufficient for this, but have now clarified in the Results that the different abundances observed “may be biased by the different ribosomal DNA copy numbers in different genera” (lines 385-386) and have cited the reference the reviewer has kindly supplied.

We further note in the Discussion that “It will therefore be worthwhile in the future to assess the distributions of other more recently developed marker genes (Penot et al., 2022; Pierella Karlusich et al., 2023)” (lines 526-530).

Minor points

1. the dataset size for the 241 protein-based host phylogeny should also be described in the main text.

Author response: The information (72,162 positions, 241 genes, removal of sequences with >66% gaps) has been included in the Materials and Methods.

2. The authors mention in Discussion “Thus, our results illuminate the mechanistics of a fundamental process that may under pin vast tracts of chloroplast evolution”.

If I understand correctly, I think this is based on “shopping bag model” when considering plastid replacements in dinoflagellates. It is helpful to add more details to clarify why the authors would like to claim so. “Chloroplast” should be replaced with “plastid”.

Author response: We agree that the term plastid is more appropriate in this context, and have used it globally throughout the manuscript. We have mentioned once in the Introduction “primary plastids, i.e. chloroplasts” to orient the non-specialist reader.

We have elaborated on our definition of the Shopping Bag model, and the specific importance of the Kareniaceae, in the Discussion: “The idea that individual genes encoding plastid-targeted proteins may exhibit evolutionary affiliation with other groups than the plastid donor, typifying the “shopping bag” model (Larkum et al., 2007), is well-established in many plastid lineages” (lines 405-407).

Nonetheless, we feel that our data are in many ways different to those previously observed in other plastid lineages. This may reflect that the karenian plastid has undergone one, and potentially multiple, recent replacement events. Nonetheless, the predominant contribution of the host to the plastid proteome is striking, which we elaborate in the Discussion: “Our data show that the dinoflagellate host was the principal contributor of nucleus-encoded proteins supporting the karenian plastid proteome” (lines 407-408).

3. Supplementary document S6.6

I found the term nitrogen fixation, but should this be replaced with "nitrogen assimilation"?

Author response: We have corrected the text as requested.

4. Figure S5

For those LGTs, all the trees should be shown in supplementary text as they are only 11 or 12 trees. Especially, please add the chlorophyllide b reductase and chlorophyllase in the figure.

Author response: Trees for all laterally transferred genes mentioned in the text have been provided among either extended view figures (Figures EV3-5) or supplementary figures (S5.1-7).

5. References

I am not picky about a format of the reference list, but I think it should be consistent throughout the list. I recommend adding journals, volumes, and pages precisely for cited papers. I found lack of them at least in Novak Vanclova et al. and Pierella Karlusich et al.

Author response: We corrected the incomplete citations and will perform a complete reformatting of the references to comply with the requirements of a concrete affiliate journal.

6. Figures

In figure 3, I strongly recommend adding RDS data, while distinguishing them by another color if they are derived from different origins from those of *Karenia*, *Karlodinium*, and *Takayama*. This would make the authors claim clearer that there are few haptophyte-derived genes for plastid targeted proteins of which origins are shared with those of the other karenian dinoflagellates.

Author response: We believe the comparison to RSD is not among the main stories of our study and adding this dimension to the already complex discussion and metabolic map schematic would compromise the overall clarity. This point is already noted by Reviewer 1 (above). However, this question may indeed be asked by some readers, therefore we decided to include the results for RSD as an additional column in Dataset EV3 and as an additional graphical element in the supplementary version of the map schematic (Figure S6). Per the reviewer's comments above, we have further stated the number of plastid-late trees shared (42) between the RSD and other karenian plastid lineages in the Results text.

In figures S5.1-2 showing LGTs, I found two paralogs of karenian dinoflagellates. What does "CP" mean? If "CP" means ChloroPlast-targeted, both paralogs of *K. brevis* in HARS and those of *K. micrum* are of plastid-targeted in TARS and they do not have cytosolic ones. I am afraid if these cases are caused by false positives of detection for plastid-targeted proteins by PredSI and ChloroP. Similarly, in figure S5.4, I found two distant paralogs of heme oxygenase in the tree and the taxon names for both types in karenian plastid lineages include "CP." Are both targeted to the plastids or of false positives?

Author response: The annotation with “CP” and darker colour denotes proteins that were predicted as plastid-targeted by our pipeline. We have clarified in supporting text 6.8 that we investigated our aminoacyl-tRNA synthetases for possible dual targeting to both plastid and mitochondria but found no evidence for it.

We have searched the *K. brevis* SP3 HARS sequence (CAMPEP-0189291366) by CD-search and note that the conserved domain (underlined) starts at residue 24 after the first predicted methionine (**bold**), which is inconsistent with the probable length a plastid-targeting sequence, and we have noted in the figure legend that this is likely to represent a false positive.

> CAMPEP_0189291366_Karenia-brevis-SP3-20130916

```
SWLVLLAFALTPGPVVAVSATILRGLLVGLQRPCAAALRLSCCAATRALPLPGASELGSRFAAAAASSARMGKEGKKKEDGK  
KKKDETKTEKLIGLEPPSGTRDFPAEMRQORYIFNKFRETANLYGFQEYDAPVLEHQELYIRKQGEEITDQMYSFDDKEGAKV  
TLRPEMPTTLARMVLNLMRVETGEMAAQLPLKWFSSIPQCWFETTQRGRKREHYQWNMDIVGVTSIYAEAEALLSAICNFFESV  
GITSKDVGLRVNSRKVLNAVTKLAGVPPDRFAETCVIIDKLDKIGAEAVKTEMREKIGLPEEVGERIVKATGAKSLEEFADLAG  
VGQNNPEVLELKHLELAEDYGYGDWLIFFDASVVRGLGYTGVVFEFGDRAGVLRRAICGGGRYDRLLTKFGSPKEIPC VGFGF  
GDCVIAELLKEKGVTPSLPEHIDFVVA AFNSEMMGKAMNAARRLRGGSVDIFTEPGKKVGKAFNYADRVGADMVAFIAPD  
EWAKGLVRIKALRMGQDVPDDQKQKDVPLEDLANVDSYFGLAPAAAAPVMSAAPAASTVKSTAPALAVPAAAASAPKAAAP  
SGTGADVEAFLVDHPYVGGFRPCARDRTLFDLRLTSGRSTPALGRWYDHDHSFPAVVRASWC
```

The green HARS sequences (including that of *Karenia brevis* SP1) in contrast typically have conserved domains starting after residues 50-60, and are likely to be genuinely plastid-targeted. Reflecting that the automated prediction approach used within our dataset may contain other such false positive results, we have chosen for tree-sorting and pathway reconstruction analyses to only consider genes in which we can identify plastid-targeted homologues of the same inferred phylogenetic origin in at least two distinct Kareniacean genera (Figs. 2, 3).

For the *Karlodinium micrum* TARS sequence we have identified a second TARS sequence (CAMPEP_0200847158) that is of apparent dinoflagellate origin and lacks a credible targeting sequence, and have updated the tree accordingly.

In the case of heme oxygenases, we are convinced that (at least) two paralogs of distinct origins are indeed plastid targeted. The presence of multiple copies of this enzyme has been noticed in other organisms including some plants (e.g., Dammeyer and Frankenberg-Dinkel, Photochemical & Photobiological Sciences, 2008) and may be reflective of functional specialization or regulation / expression under different conditions. We have added a brief discussion on this to the text: “Two evolutionarily distinct versions of the biliverdin-producing haem oxygenase seem to be present in the plastid of representatives of both *Karenia* and *Karlodinium*: one of plastid-late and one of green-like origin (each in one or two copies, Figure EV3). This enzyme is often present in multiple copies that have been proposed to play slightly different roles in other reactions of bilin metabolism, especially in photosynthetic organisms where these molecules serve as chromophores of phytochromes.” (lines 257-263).

Reviewer #3 (Significance (Required)):

Significance

General assessment: provide a summary of the strengths and limitations of the study. What are the strongest and most important aspects? What aspects of the study should be improved or could be developed?

This study by Novak Vanclova et al. provide new transcriptome datasets from multiple species in kareniacean dinoflagellates including harmful and toxic species. Their transcriptome datasets would help understand their biology, evolution, and ecology. The authors also provide a program that predicts plastid proteomes in those dinoflagellates, which would be useful for future studies to focus on kareniacean dinoflagellate plastids, after further refinement. The most important aspect of this study is that many plastid-targeted proteins might be derived from a particular haptophyte lineage, although it is still not sure whether they are derived from LGTs or EGTs. Phylogenetic analyses performed in this study should be improved by adding some plastid genomes, in order to gain more conclusive results. In addition to methods, interpretation of the current results and proposals on plastid evolution should be toned-down.

Advance: compare the study to the closest related results in the literature or highlight results reported for the first time to your knowledge; does the study extend the knowledge in the field and in which way? Describe the nature of the advance and the resulting insights (for example: conceptual, technical, clinical, mechanistic, functional,...).

Although there are technical issues, this study improves our conceptual understanding the plastid proteome evolution in Kareniacean dinoflagellates. The plastid proteomes are comprised of proteins with more various origins in those dinoflagellates, suggesting more complex plastid proteome evolution than previously thought.

Audience: describe the type of audience ("specialized", "broad", "basic research", "translational/clinical", etc...) that will be interested or influenced by this research; how will this research be used by others; will it be of interest beyond the specific field?

This study seems to be "basic research".

Please define your field of expertise with a few keywords to help the authors contextualize your point of view. Indicate if there are any parts of the paper that you do not have sufficient expertise to evaluate.

algal evolution, eukaryotic evolution, mitochondrial metabolisms, plastid metabolisms, phylogenomics

Manuscript number: EMBOR-2023-58545V2

Title: New plastids, old proteins: repeated endosymbiotic acquisitions in fucoxanthin dinoflagellates

Author(s): Anna Novák Vanclová, Charlotte Nef, Zoltan Fussy, Adél Vancl, Fuhai Liu, Chris Bowler, and Richard Dorrell

Dear Dr. Novák Vanclová and Dr. Dorrell,

Thank you for the submission of your revised manuscript to EMBO Reports. With the help of our senior editor, Martina Rembold, as well as our editorial assistants, I went over your manuscript and I still have a few minor comments about formatting that need to be addressed before final acceptance.

Once you have made these minor revisions, please use the following link to submit your corrected manuscript:

Link Not Available

If all remaining corrections have been attended to, you will then receive an official decision letter from the journal accepting your manuscript for publication in the next available issue of EMBO Reports. This letter will also include details of the further steps you need to take for the prompt inclusion of your manuscript in our next available issue.

Comments:

- As a standard procedure, we edit the title and abstract of manuscripts to make them more accessible to a general readership. Please find the edited versions near my signature below and let me know if you do NOT agree with any of the changes. Please note your title was 106 characters, but our strict policy is that titles cannot be longer than 100 characters (including spaces). Please make sure to include these changes in the new manuscript file you upload to the system.
- You provided 6 keywords, but we allow only up to 5. Please delete one of them.
- Please update the 'Conflict of interest' paragraph to our new 'Disclosure and competing interests statement'. For more information see <https://www.embopress.org/page/journal/14693178/authorguide#conflictsofinterest>
- The email address provided for correspondence with Dr. Novák Vanclová is a gmail account and not institutional. While we understand the advantage of such addresses for early career researchers, we would strongly prefer an institutional address, if possible.
- Figure 8: it would be helpful if you can refer in the text to scenarios A, B and C as you annotate them in the figure itself.
- Please add page numbers to the table of contents (Appendix) on the title page; the nomenclature is wrong, it should be Appendix Figure S1, etc. the callouts in the manuscript text need to be updated accordingly; also Figure S4_1, Figure S4_2 should be avoided and corrected to Appendix Figure S4, Appendix Figure S5, etc.
- The figure legends for figure panels EV2e-f are not provided in the manuscript. This needs to be rectified.
- Please place Table 1 between main and Expanded View figure legends.
- EMBO Reports papers are accompanied online by A) a short (1-2 sentences) summary of the findings and their significance, B) 2-3 bullet points highlighting key results and C) a synopsis image that is 550x300-600 pixels large (width x height) in PNG for JPG format. You can either show a model or key data in the synopsis image. Please note that the size is rather small and that text needs to be readable at the final size. Please send us this information along with the revised manuscript.
- On a different note, I would like to alert you that EMBO Press offers a new format for a video-synopsis of work published with us, which essentially is a short, author-generated film explaining the core findings in hand drawings, and, as we believe, can be very useful to increase visibility of the work. This has proven to offer a nice opportunity for exposure i.p. for the first author(s) of the study. Please see the following link for representative examples and their integration into the article web page:
https://www.embopress.org/video_synopses
<https://www.embopress.org/doi/full/10.15252/emboj.2019103932>

With kind regards,

Yehu Moran

Suggested title (77 characters including spaces) :
Repeated endosymbiotic acquisitions in fucoxanthin-containing dinoflagellates

Suggested abstract (175 words)

Dinoflagellates are a diverse group of ecologically significant micro-eukaryotes that can serve as model system for plastid symbiogenesis due to their proneness to plastid loss and replacement via serial endosymbiosis. Kareniaceae harbor fucoxanthin-pigmented plastids instead of the ancestral peridinin-pigmented ones and support them with a diverse range of nucleus-encoded plastid-targeted proteins originating from the haptophyte endosymbiont, dinoflagellate host, and/or LGT. Here, we present predicted plastid proteomes from seven distantly related kareniaceans in three genera (Karenia, Karlodinium, and Takayama) and analyze their evolutionary patterns using automated tree building and sorting methods. We project relatively limited (~10%) proportion of haptophyte signal pointing towards a shared origin in the family Chrysochromulinaceae. Our data establish significant variations in the functional distributions of these signals, emphasizing the importance of micro-evolutionary processes in shaping the chimeric proteomes. Analysis of plastid genome sequences recontextualizes these results by a striking finding the extant kareniacean plastids are in fact not all of the same origin, as two of the studied species (Karlodinium armiger, Takayama helix) possess plastids from different haptophyte orders than the rest.

Yours sincerely,

Yehu Moran
Editor
EMBO Reports

The authors have addressed all minor editorial requests.

Dr. Anna Novák Vanclová
Institute Jacques Monod
15 Rue Hélène Brion
Paris 75013
France

Dear Dr. Novák Vanclová,

I am very pleased to accept your manuscript for publication in the next available issue of EMBO Reports. Thank you for your contribution to our journal.

Yours sincerely,

Yehu Moran
Academic Editor
EMBO Reports

Rev_Com_number: N/a
New_manu_number: EMBOR-2023-58545V3
Corr_author: Novák Vanclová
Title: New plastids, old proteins: repeated endosymbiotic acquisitions in kareniacean dinoflagellates